# Single-cell transcriptomics uncovers EGFR signaling-mediated gastric progenitor cell differentiation in stomach homeostasis

Hitomi Takada [1,5], Yohei Sasagawa [2,3,5], Mika Yoshimura[2], Kaori Tanaka[2], Yoshimi Iwayama[2,3], Tetsutaro Hayashi [2], Ayako Isomura-Matoba [2], Itoshi Nikaido [2,3,4] ✉ & Akira Kurisaki [1] ✉

Defects in gastric progenitor cell differentiation are associated with various gastric disorders, including atrophic gastritis, intestinal metaplasia, and gastric cancer. However, the mechanisms underlying the multilineage differentiation of gastric progenitor cells during healthy homeostasis remain poorly understood. Here, using a single-cell RNA sequencing method, Quartz-Seq2, we analyzed the gene expression dynamics of progenitor cell differentiation toward pit cell, neck cell, and parietal cell lineages in healthy adult mouse corpus tissues. Enrichment analysis of pseudotime-dependent genes and a gastric organoid assay revealed that EGFR-ERK signaling promotes pit cell differentiation, whereas NF-κB signaling maintains gastric progenitor cells in an undifferentiated state. In addition, pharmacological inhibition of EGFR in vivo resulted in a decreased number of pit cells. Although activation of EGFR signaling in gastric progenitor cells has been suggested as one of the major inducers of gastric cancers, our findings unexpectedly identified that EGFR signaling exerts a differentiation-promoting function, not a mitogenic function, in normal gastric homeostasis.

A major part of the glandular stomach in mice and humans is the gastric corpus, which secretes acid from parietal cells, proteases from chief cells, and mucus from pit and neck cells. These functional epithelial cells are considered to be continuously replenished by a population of stem/progenitor cells in gastric units[1]. Defects in gastric epithelial cell differentiation are therefore associated with various gastric disorders. For example, chronic infection with *Helicobacter pylori* induces pseudopyloric metaplasia, wherein the number of proliferating progenitor cells and premature cells expressing both neck and chief cell markers were increased, and the number of parietal cells was decreased[1,2]. Additionally, Ménétrier's disease is characterized by

hyperplasia of surface pit cells and a decrease in the number of chief and parietal cells[3,4]. The Zollinger–Ellison syndrome, which is caused by hypergastrinemia, results in hyperplasia of parietal cells and chief cells[5,6]. However, little is known about the mechanisms underlying these dysregulated cell fate decisions during disease progression.

Ultrastructural evidence suggests that in healthy corpus tissues, the highly proliferative isthmus progenitor cells in the upper-middle portion of gastric units differentiate into multiple epithelial cell lineages, including pit cells, neck cells, parietal cells, and endocrine cells[7–9]. Chief cells were previously believed to be differentiated from neck cells[10,11]. However, recent studies also suggested the self-renewal

[1]Laboratory of Stem Cell Technologies, Graduate School of Science and Technology, Nara Institute of Science and Technology, Takayama-cho, Ikoma, Nara, Japan. [2]Laboratory for Bioinformatics Research, RIKEN Center for Biosystems Dynamics Research, Wako, Saitama, Japan. [3]Department of Functional Genome Informatics, Biological Data Science, Medical Research Institute, Tokyo Medical and Dental University, Bunkyo, Tokyo, Japan. [4]Master's/Doctoral Program in Life Science Innovation (Bioinformatics), Degree Programs in Systems and Information Engineering, Graduate School of Science and Technology, University of Tsukuba, Tsukuba, Ibaraki, Japan. [5]These authors contributed equally: Hitomi Takada, Yohei Sasagawa. ✉e-mail: itoshi.nikaido@riken.jp; akikuri@bs.naist.jp

ability of chief cells as stem cells to maintain themselves[12–14]. Although tight coordination of self-renewal and differentiation of the progenitor cells is a key determinant for tissue integrity, these molecular mechanisms remain poorly understood.

Recent studies using mouse genetic models have suggested the involvement of several regulatory pathways associated with gastric epithelial cell differentiation. For example, the overexpression of NOGGIN in parietal cells reduced the number of parietal cells, expanded the proliferating cells, and increased the number of transitional cells expressing both neck and chief cell markers[15]. The parietal cell-specific deletion of SHH expanded surface pit cells, increased the number of proliferating cells, and delayed the differentiation of neck cells into chief cells[16,17]. Activation of NOTCH signaling in parietal cells and their progenitor cells maintained the progenitor cells in an undifferentiated state[18]. TGFα overexpressing transgenic mice exhibited hyperplasia of surface pit cells and depletion of parietal cells and chief cells[19,20]. Although overexpression or knockout of these signaling molecules induces some phenotypic changes directly in the intended cells, these defects, in turn, cause secondary alterations in neighboring cells via epithelial–epithelial and mesenchymal–epithelial interactions[21–23]. Therefore, specific signaling pathways directly regulating the differentiation of particular cell types during homeostasis remain to be clarified.

In this study, to identify the signaling pathways directly regulating epithelial cell differentiation in corpus units, we employed Quartz-Seq2[24], which is the most precise single-cell RNA-seq method among 13 single-cell RNA sequencing protocols according to a recent benchmarking study[25]. By improving the poly-A tagging efficiency during the whole transcript amplification process, Quartz-Seq2 achieved a high gene detection rate with an effective conversion of initial reads to unique molecular identifier (UMI) counts. Pseudotemporal ordering of corpus epithelial cells and enrichment analysis of pseudotime-dependent genes identified that TGFα-EGFR-ERK signaling promotes pit cell differentiation and TNFSF12-NF-κB signaling maintains gastric epithelial cells in the undifferentiated state. These signaling pathways were further validated using an in vitro gastric organoid assay and in vivo mouse experimental models.

## Results

### Single-cell transcriptome analysis of gastric units in adult mice
To profile the gene expression of gastric tissue cells, corpus and antrum gastric units isolated from 10-week-old C57BL/6J wild-type mice were dissociated into single cells, sorted using a cell sorter (Fig. 1a, b and Supplementary Fig. 1a), and analyzed using Quartz-Seq2[24] (dataset 1). After the removal of low-quality cells (Supplementary Fig. 2a, b; see "Methods"), the transcriptomes of 491 cells were processed for further analysis. We identified 15 distinct clusters using unsupervised clustering and annotated them based on the expression of known marker genes (Supplementary Fig. 3a, b). All clusters consisted of cells from both biological replicates (Supplementary Fig. 2b). In this dataset, pit cells were separated into four clusters, corpus mature pit cells (cluster 8), corpus pre-pit cells (cluster 2), antrum mature pit cells (cluster 1), and antrum pre-pit cells (cluster 6). We identified a mature pit cell-specific marker, *Aqp3* (Supplementary Fig. 3b). Immunofluorescence staining confirmed that the mature pit cell marker AQP3 was restricted to the most luminal end of the pit cells where mature pit cells are located, whereas the pan pit cell marker GKN2 was broadly expressed in all pit cells located above MKI67+ isthmus progenitor cells (Fig. 1g, h). We also identified the corpus pit cell-specific markers *Sctr* and *Basp1* (Supplementary Fig. 3b, c) and the antrum pit cell-specific markers *Krt7* and *Krt20* (Supplementary Fig. 3b, d).

To further investigate the major corpus epithelial cells in detail, corpus gastric cells were isolated with an improved protocol (see "Methods") and analyzed using Quartz-Seq2 (dataset 2; Fig. 1b,

Supplementary Figs. 1b, c and 2c, d). Unsupervised clustering of 3993 cells using the Seurat algorithm identified 20 distinct cell clusters (Fig. 1c), which were annotated based on the expression of known marker genes (Fig. 1d and Supplementary Fig. 4) and cluster-specific genes (Supplementary Fig. 5). Two pit cell clusters, *Aqp3*+ mature pit cells (cluster 13) and pre-pit cells (cluster 4), expressed corpus pit cell-specific markers *Basp1* and *Sctr* but not antrum pit cell markers *Krt7* and *Krt20* (Supplementary Fig. 4b). Five isthmus progenitor cell clusters were separated based on cell cycle phase (Fig. 1e). Among them, clusters 5 and 6 expressed a proliferation-associated transcription factor *Foxm1* that stimulates cell cycle progression (Fig. 1d). Many of these isthmus progenitor cells express significant levels of pit cell marker *Muc5ac* or neck cell marker *Muc6* (Fig. 1f), suggesting that these isthmus cells are transcriptionally primed toward the pit or neck cell lineage. Although some of the previously identified isthmus progenitor cell-specific markers (*Stmn1* and *Iqgap*)[12,26] and other markers (*Tcf19* and *Hmgb2*) were identified, pan-stem cell markers, such as *Sox2*, *Prom1*, and *Bmi1*, did not exhibit isthmus cell-specific expression (Fig. 1d and Supplementary Fig. 4b). Previous studies have shown that TCF19 increases cell cycle progression in the gastric cancer cell line MKN-45[27] and that HMGB2 is a downstream effector of CENPU-mediated cell proliferation in AGS cells[28], which is consistent with the highly proliferative state of isthmus progenitor cells. Intriguingly, among *Muc6*+ neck cell clusters, cluster 12 expressed *Muc6* and a chief cell marker, *Cblif* (*Gif*) (Fig. 1d), suggesting the presence of intermediate-state cells between neck cells and chief cells under healthy homeostasis of corpus tissues.

### Transcription factors and regulatory pathways involved in differentiation from isthmus progenitor cells to pit cells
Dataset 2 was further analyzed using FateID algorithm, a method for the quantification of cell fate bias[29]. One of the chief cell markers, *Pgc*, was expressed in most gastric epithelial cells at mRNA level, as previously reported[12,30], although the expression level gradually increased from mature pit cells to chief cells (Fig. 1d). Interestingly, pseudotemporal ordering of pit, isthmus, and neck cells presented a continuum of differentiation states where the gradation of *Pgc* expression was observed (Supplementary Fig. 8a), while parietal cells were found as isolated clusters where the *Pgc* gradation was also observed from pre-parietal to parietal cells (Fig. 2a, Supplementary Fig. 8a). Isthmus progenitor cells differentiating toward pit cell (yellow) or neck cell (blue) lineage (Supplementary Fig. 8b), which were identified using the FateID algorithm, were basically merged with isthmus progenitor cells with high pit cell feature or high neck cell feature identified in Fig. 1f, respectively, supporting the reliability of our FateID analysis.

Pseudotemporal expression profiles from isthmus progenitor cells to pit cells (t13) identified 47 gene co-expression nodes (Fig. 2b). Hierarchical clustering of the nodes identified two major groups (Fig. 2c). Group A genes with increasing expression from isthmus to pit cells contained various pit cell markers such as *Gkn2*, *Gkn1*, and *Muc5ac*, whereas group B genes highly expressed in isthmus progenitor cells contained known isthmus progenitor cell markers such as *Mki67*, *Pcna*, and *Stmn1* (Fig. 2d). Pit cell-related group A contained several transcription factors (TFs) such as *Foxa3*, *Pparg*, and *Tcf23*, in addition to the previously reported pit cell differentiation-related TF *Klf4*[31]. In contrast, the isthmus progenitor cell-related group B contained *Dnmt1*, *Cebpb*, *Foxm1* (Fig. 2d), and *Hmga1* (Supplementary Data 1 and 6), which are associated with proliferative gastric cancer[32–35].

Enrichment analysis of group A genes using g:Profiler[36] and Enrichr[37] identified putative regulatory pathways associated with pit cell differentiation, including integrin, EGFR, and MAPK signaling pathways, as well as actin cytoskeleton-related GO terms (Fig. 2e). The pathways enriched in group B genes were related to cell cycle and proliferation (e.g., mRNA processing and ribosome biogenesis), consistent with the highly proliferative state of isthmus progenitor cells.

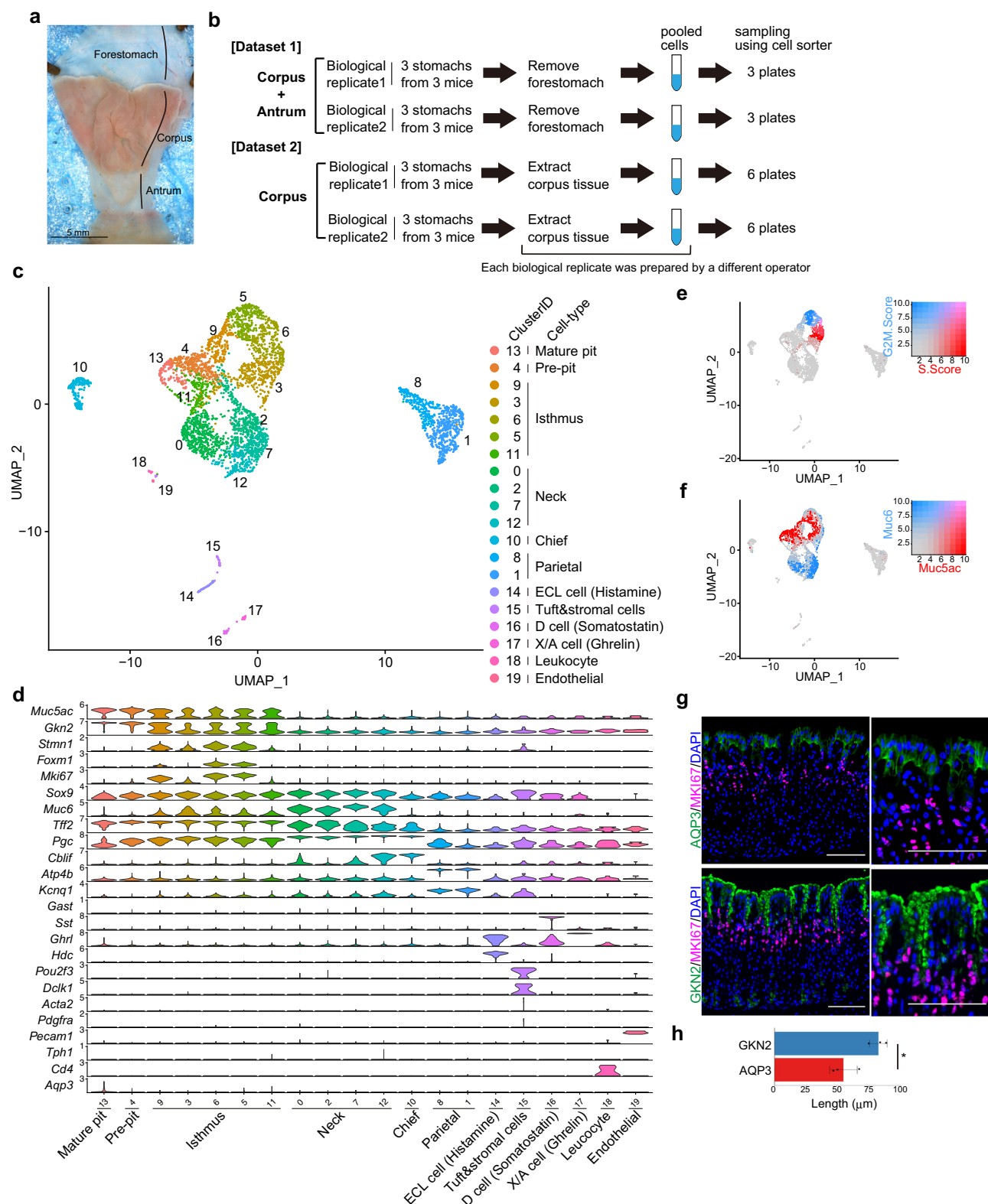

Additionally, TNF-α/NF-kB signaling and proteasome degradation pathways were enriched in genes of group B (Fig. 2f). Most enriched pathways and pseudotime-specific TFs of dataset 1 were similar to those of dataset 2 (Supplementary Fig. 9 and Supplementary Data 2 and 7). Notably, EGFR signaling and TNF-α/NF-kB signaling were enriched in the pit cell-related group A and isthmus progenitor cell-related group B, respectively (Supplementary Fig. 9e, f).

## Gene expression dynamics from isthmus progenitor cells to neck cell and parietal cell lineages

With respect to the differentiation trajectory from isthmus progenitor cells to neck cells, FateID analysis identified 65 nodes (Supplementary Fig. 10a), which were grouped using hierarchical clustering (Supplementary Fig. 10b). Neck cell differentiation-related group C contained TFs such as *Hes1*, *Pbx1*, and *Spdef*, whereas isthmus progenitor cell-

**Fig. 1 | Single-cell analysis of adult mouse corpus gastric units. a** Luminal surface of adult mouse stomach tissue. Scale bar, 5 mm. **b** Schematic of the experimental design. Cells prepared from the corpus and antrum were analyzed in dataset 1, and cells prepared from the corpus were analyzed in dataset 2. In each dataset, two biological replicates from three mice were prepared by two different operators. Each biological replicate was sorted into three or six plates in datasets 1 and 2, respectively. **c** UMAP visualization of the cells isolated from adult mouse corpus gastric units (dataset 2). Cells are colored according to the clustering results. **d** Violin plots showing the expression of known marker genes in each cluster. **e, f** UMAP of gastric cells colored by the score of cell cycle-state (S-phase and G2/M-phase) (**e**) and the expression levels for *Muc5ac* and *Muc6* (**f**). **g** Immunofluorescence staining of adult mouse corpus gastric units with a mature pit cell marker AQP3 (green), a pan pit cell marker GKN2 (green), an isthmus progenitor cell marker MKI67 (red), and DAPI (blue). Scale bar, 100 μm. **h** Bar graph showing the lengths of GKN2+ and AQP3+ regions along each gastric unit in (**g**). Each data point represents the mean value of at least 10 gastric units. Data are presented as mean values ± standard deviation (SD; *n* = 3 mice). Statistical information: significance was calculated by two-tailed Student's *t*-tests. *p = 0.0230. Source data are provided as a Source Data file.

related group D again contained gastric tumor-related TFs such as *Dnmt1* and *Foxm1* (Supplementary Fig. 10c and Supplementary Data 3 and 6). Enrichment analysis identified lysosome-related GO terms in group C (Supplementary Fig. 10d) and TNF-α/NF-kB and proteasome degradation pathways in group D (Supplementary Fig. 10e). These enriched pathways were reproducibly observed in dataset 1 (Supplementary Fig. 11d, e, Supplementary Data 4).

FateID analysis from isthmus progenitor cells to parietal cells identified 58 nodes (Supplementary Fig. 12a, b). Group E displaying immediate upregulation in expression upon parietal cell differentiation contained parietal cell markers, *Atp4b* and *Kcnq1* (Supplementary Fig. 12c, Supplementary Data 5), whereas Group F displaying immediate downregulation in expression contained the cell proliferation markers *Mki67* and *Pcna*. TNF-α/NF-kB, EGFR, and Delta-Notch signaling pathways were enriched in Group F (Supplementary Fig. 12d). Downregulation of Delta-Notch signaling in parietal cell differentiation is consistent with the previous study showing the role of NOTCH signaling in antagonizing parietal cell differentiation[18]. Although no significantly enriched pathway and GO (adjusted *p*-value < 0.05) was identified in group E, in another parietal cell-related group G showing transient upregulation in expression upon differentiation (Supplementary Fig. 12c), pathways related to mitochondrial electron transport and oxidative phosphorylation were identified (Supplementary Fig. 12e), which is consistent with the previously reported increased mitochondrial biogenesis during parietal cell differentiation[38]. Group G contained the previously described TFs (*Esrrg* and *Esrra*)[39] associated with parietal cell function, as well as other TFs (e.g., *Ybx2* and *Mitf*) (Supplementary Fig. 12c and Supplementary Data 6). In dataset 1, the gene expression dynamics were not analyzed due to the few isthmus progenitor cells in the parietal cell differentiation trajectory (Supplementary Fig. 13a, b).

## Distinct expression patterns of ligands and receptors between pit, neck, and parietal cell lineages

Non-major gastric cells (the cells other than pit, isthmus, neck, parietal, and chief cells) were analyzed with the combined datasets 1 and 2. Eleven clusters, including stromal cells, endocrine cells, tuft cells, endothelial cells, and leukocytes, were identified (Supplementary Fig. 14a). G cells were identified only in dataset 1, which contains antrum, consistent with the antrum-specific localization of G cells. Other cell types were present in both datasets (Supplementary Fig. 14b). We proceeded with the downstream analysis using the data excluding G cells. Immunofluorescence staining confirmed the presence of two types of stromal cells (PDGFRα+ and ACTA2+ cells), PECAM+ endothelial cells, and CD4+ leukocytes in corpus gastric glands (Supplementary Fig. 14d). Notably, ACTA2+ stromal cells were primarily located in the middle section, whereas PDGFRα+ stromal cells were primarily located in the upper and lower sections of the corpus gastric glands (Supplementary Fig. 14e).

Using the major gastric cell dataset (dataset 2, Fig. 1c) and non-major gastric cell dataset (combined dataset 1 and 2, Supplementary Fig. 14a), we analyzed intercellular communications using CellChat analysis[40], which predicts outgoing (i.e., the signals sent out from the cells) and incoming (i.e., the signals received by the cells) signaling

pathways across all cell types (Fig. 3a and Supplementary Fig. 15). The data suggested high EGF signaling input mainly in pit cells (blue), high TNF signaling input in isthmus cells (blue), and low NOTCH signaling input in parietal cells (white), which is consistent with the results of enrichment analysis of pseudotime-dependent genes (Fig. 2e, f and Supplementary Figs. 9e, f, 12d). CellChat suggested that *Hbegf-Egfr/Erbb2*, *Tgfa-Egfr/Erbb2*, and *Btc-Egfr/Erbb2* were the major contributors for EGF signaling, and *Tnfsf12/Tnfrsf12a* and *Tnf/Tnfrsf1a* were the dominant contributors for TNF signaling among the known ligand-receptor pairs (Fig. 3b). Inferred cell–cell interaction showed that EGF signaling input in epithelial cells is regulated in both autocrine and paracrine (epithelial–epithelial) manners, whereas TNF signaling input in epithelial cells is mainly regulated in paracrine (stromal-epithelial) manner (Supplementary Fig. 16a, b). To validate CellChat prediction, we analyzed EGF and TNF ligand-receptor expression patterns across cell types. Immunofluorescence staining showed that EGFR is specifically expressed in mature and pre-pit cells, with the highest expression at the luminal surface of the glands (Fig. 3c), suggesting that EGFR signaling is mainly activated in pit cells. Single-cell RNA-seq data showed that EGF ligands are preferentially expressed in several epithelial cell types, which is consistent with a high EGF signaling output in pit cells as predicted by CellChat (Fig. 3a), and that the predominant EGF ligands expressed in pit cells were *Tgfa* and *Btc* (Fig. 3d). With respect to TNF signaling, the major TNF superfamily ligand in the stomach is *Tnfsf12*, which is expressed in stromal cells and endothelial cells (Fig. 3d) and its cognitive receptor *Tnfrsf12a* is preferentially expressed in isthmus progenitor cells. *Tnf* is expressed in enterochromaffin (EC) cells and leukocytes, and its cognitive receptor *Tnfrsf1a* is expressed in all epithelial cell types. Taken together, these data suggested that *Tgfa*- or *Btc*/Egfr regulated pit cells in an autocrine manner and *Tnfsf12/Tnfrsf12a* regulated isthmus progenitor cells in a paracrine manner.

CellChat analysis predicted various signaling pathways, including BMP, IGF, and ncWNT (Fig. 3a and Supplementary Fig. 15), although pathway enrichment analysis (Fig. 2, Supplementary Figs. 9–13) did not identify significant enrichment of these pathways. We analyzed the expression of all known ligand-receptor for these signaling pathways using our datasets (Supplementary Figs. 17–20), which suggested consistency and inconsistency with previous in vivo mouse studies[15,16,41–43] (see "Discussion").

## EGFR signaling promotes pit cell differentiation and suppresses neck cell fate decisions

Enrichment analysis and ligand-receptor expression profiles strongly suggested that TGFα-EGFR signaling is involved in pit cell differentiation. Therefore, we next utilized a well-established gastric organoid system[44] with some modifications (see "Methods") to examine the effect of this signaling on pit cell differentiation. In all analyses, corpus organoids prepared from the adult mouse stomach were used. Under our serum-free and chemically defined culture conditions, approximately half of the cells in the organoids were MKI67+ proliferating isthmus progenitor cells, and the other cells were immature pre-pit cells weakly expressing GKN2 (Fig. 4c and Supplementary Fig. 21a). Differentiated neck and chief cells were not observed in organoids

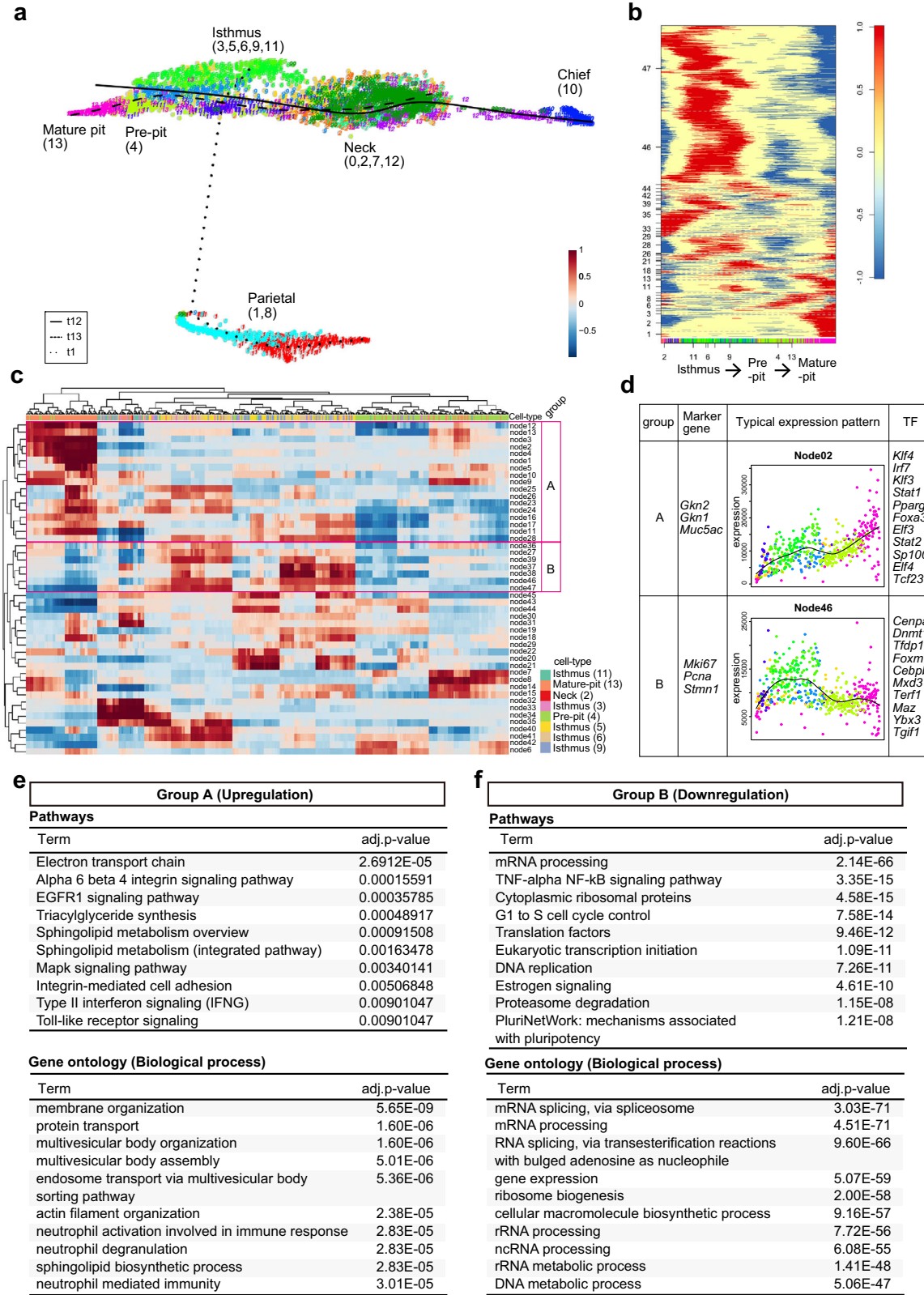

(Supplementary Fig. 21b). Therefore, subsequent analyses primarily focused on the expression of the pit cell markers *Muc5ac*, *Gkn2*, and *Aqp3*, as well as the isthmus progenitor cell marker *Mki67*. Additionally, we analyzed the expression of *Sox9* and *Pgc*, whose expression levels gradually increased from pit to neck cells (Supplementary Figs. 6 and 7). Our gastric organoids could be passaged several times (Supplementary Fig. 21c) and displayed decreased expression of pit cell markers and increased expression of isthmus progenitor cell makers after serial passages (Supplementary Fig. 21d). Therefore, our culture method preferentially propagates isthmus progenitor cells but not pit cells. Culture with TGFα for 6 days significantly increased the expression of pit cell markers *Muc5ac*, *Gkn2*, and *Aqp3*, whereas the isthmus progenitor cell markers and other cell type markers *Mki67*, *Stmn1*, *Sox9*, and *Pgc* were markedly suppressed (Fig. 4a, b and

**Fig. 2 | Characterization of the pit cell differentiation process. a** Pseudotemporal ordering of pit cells, isthmus progenitor cells, neck cells, and parietal cells identified in dataset 2. Principal curves are shown for mature pit cells (t13, heavy dotted line), neck cells (t12, solid line), and parietal cells (t1, fine dotted line). **b** Self-organizing map of binarized pseudotemporal expression profiles along the pit cell differentiation trajectory. The *x*-axis indicates the cells involved in the pit cell lineage that are colored according to cell types, and the *y*-axis indicates the nodes. **c** Hierarchical clustering of co-expression nodes in (**b**). The nodes in groups A and B show upregulation and downregulation patterns of the genes in the pit cell differentiation trajectory, respectively. The *x*-axis indicates the cells involved in the pit cell lineage that are colored according to cell types, and the *y*-axis indicates the nodes. The *x*-axis in (**c**) is not the same as that of (**b**) due to the hierarchical clustering of the cells. The colors for each cell type are shown in the bottom right corner. **d** Characteristics of groups A and B identified in (**c**). Marker genes in each group were shown. Average pseudotemporal expression profile of the representative node is indicated by a black line. The *x*-axis indicates pseudotime from isthmus progenitor cells to pit cells, and the *y*-axis indicates expression level. Colors in each dot represent cell types, which correspond to colors used in (**a**). Among the TFs involved in groups A and B, the top 10 TFs specifically expressed in pit cells and isthmus progenitor cells (LogFC > 0.25) are shown here, respectively.
**e**, **f** Characterization of the pseudotime-dependent genes included in group A (**e**) and group B (**f**). Upper panel, pathway enrichment analysis of the pseudotime-dependent genes. The top ten pathways with significant enrichment (adjusted *p*-value < 0.05) are listed. Lower panel, GO analysis of the pseudotime-dependent genes. The top ten terms with significant enrichment (adjusted *p*-value < 0.05) are listed. Adjusted *p*-value was determined using one-tailed Fisher's exact test with g:SCS method in g:Profiler[36] (pathways) and Fisher's exact test with Benjamini−Hochberg adjustment in Enrichr[37] (GO).

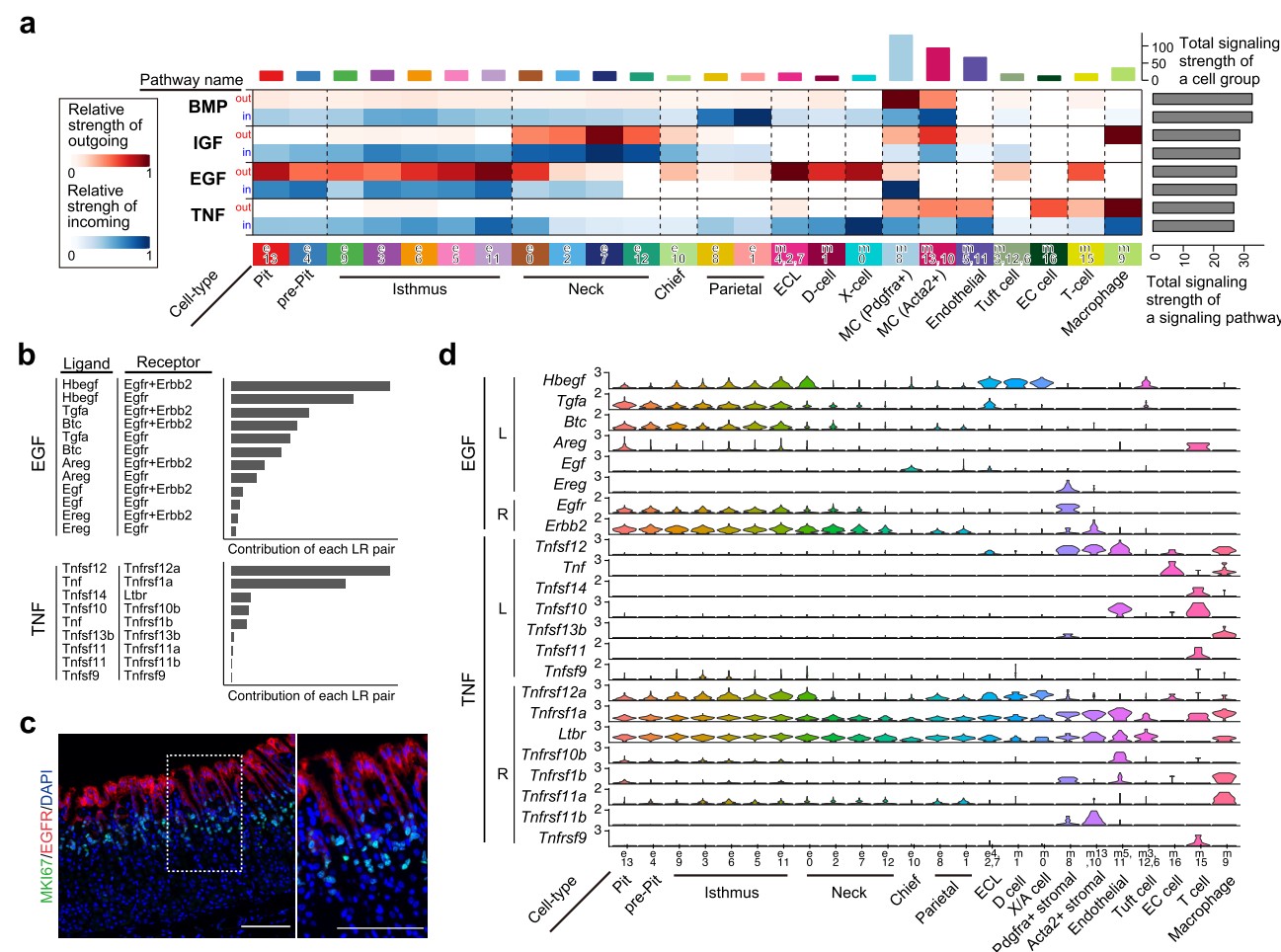

**Fig. 3 | Analysis of signaling pathways and ligand-receptor interactions across gastric cell types. a** CellChat-based analysis of outgoing and incoming signaling in each cell type. The *x*-axis indicates cell types, and the *y*-axis indicates signaling pathways. For each signaling, the outgoing signal intensity is shown in red (upper row), and the incoming signal intensity is shown in blue (lower row). **b** Relative contribution of EGF and TNF ligand-receptor pairs calculated by CellChat. **c** Immunofluorescence staining of adult mouse corpus tissues with EGFR (red), MKI67 (green), and DAPI (blue). Scale bar, 100 μm. Inset shows high-magnification image of boxed area. The image is representative of three independent experiments. **d** Expression of EGF and TNF ligands and receptors across gastric cell types.

Supplementary Fig. 21e), suggesting that TGFα specifically induced pit cell differentiation. Considering that the expression level of chief cell-specific markers (*Troy*, *Gif*, and *Lgr5*) was exceptionally low in our gastric organoids, the effect of TGFα on the expression of these markers was not clearly observed (Supplementary Fig. 21e). Immunofluorescence staining confirmed TGFα-induced upregulation of GKN2 expression and dramatic downregulation of MKI67 expression at the protein levels (Fig. 4c–e). In passage 1 organoids, where the isthmus progenitor cells are enriched (Supplementary Fig. 21d), TGFα similarly induced pit cell differentiation and inhibition of proliferation

(Supplementary Fig. 21f, g). Although we observed a limited number of proliferative pre-pit cells, which express MKI67 and a low level of GKN2, these cells were not increased by TGFα treatment (Supplementary Fig. 21h, i). TGFα-treated organoids do not proliferate after passage (Supplementary Fig. 21j, k). All these data indicated that EGFR signaling induced pit cell differentiation but not pit cell proliferation. Conversely, treatment with the EGFR inhibitor erlotinib markedly decreased the expression of pit cell markers and increased the other cell markers, including *Mki67*, *Sox9*, and *Pgc* (Fig. 4f, g and Supplementary Fig. 22a−e), confirming the specificity of EGFR signaling-

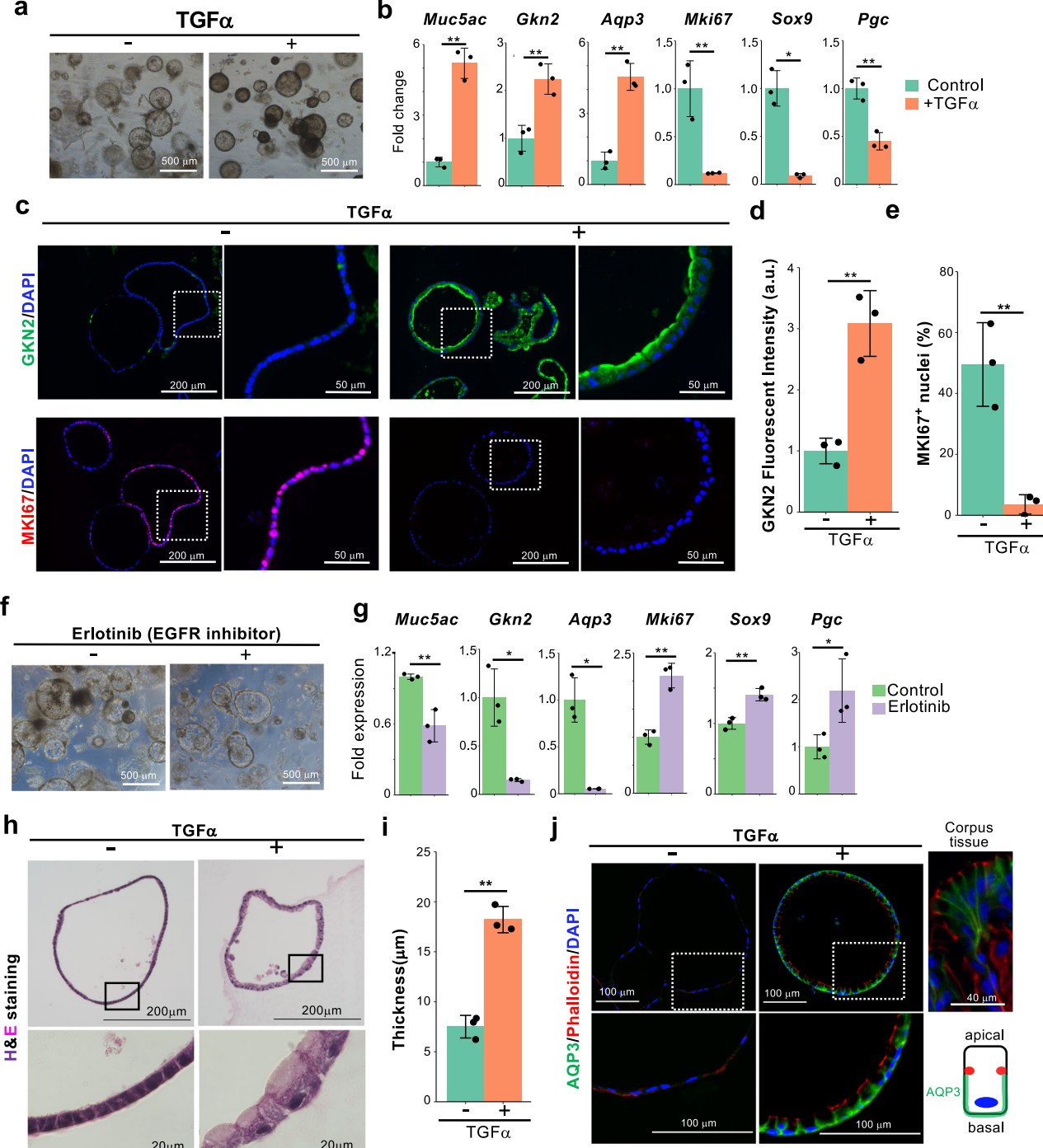

**Fig. 4 | TGFα promotes pit cell differentiation. a** Corpus organoids cultured with or without TGFα for 6 days. The images are representative of three biologically independent samples. **b** qPCR analysis of gastric epithelial cell marker expression in the corpus organoids in (**a**). Data are presented as mean ± SD ($n$ = 3 biologically independent samples). **c** Immunofluorescence staining of corpus organoids in (**a**) with GKN2 (green), MKI67 (red), and DAPI (blue). High-magnification images of the dotted squares are shown on the right side. **d** Quantification of GKN2 fluorescence intensity in (**c**). Each data point represents a mean value of six pictures from three biologically independent samples. Data are presented as mean ± SD. **$p$ = 0.0033. **e** Quantification of the percentage of MKI67⁺ cells in all DAPI⁺ cells in (**c**). Each data point represents the mean value of three pictures from three biologically independent samples. Data are presented as mean values ± SD. **$p$ = 0.0048. **f** Corpus organoids treated with or without an EGFR inhibitor, erlotinib (0.5 μM). **g** qPCR analysis of gastric epithelial cell marker expression in the organoids in (**f**). Data are presented as mean ± SD ($n$ = 3 biologically independent samples). **h** H&E staining of the organoids cultured with or without TGFα. High-magnification images of the squares are shown in the lower panels. **i** Quantification of the thickness of epithelial cells in the organoids in (**h**). Each data point represents the mean value of three pictures from three biologically independent samples. Data are presented as mean ± SD. **$p$ = 0.0004. **j** Immunofluorescence staining of AQP3, phalloidin, and DAPI in corpus organoids cultured with or without TGFα. High-magnification images of the dotted squares are shown in the lower panels. Right upper panel, immunostaining of adult mouse corpus tissue. The lower right cartoon illustrates the location of AQP3 and phalloidin in the pit cell. The images are representative of three independent experiments. Source data are provided as a Source Data file. Significance was calculated by two-tailed Student's $t$-tests for samples with equal variances or two-sided Welch's $t$-tests for samples with unequal variances. *$p$ < 0.05; **$p$ < 0.01.

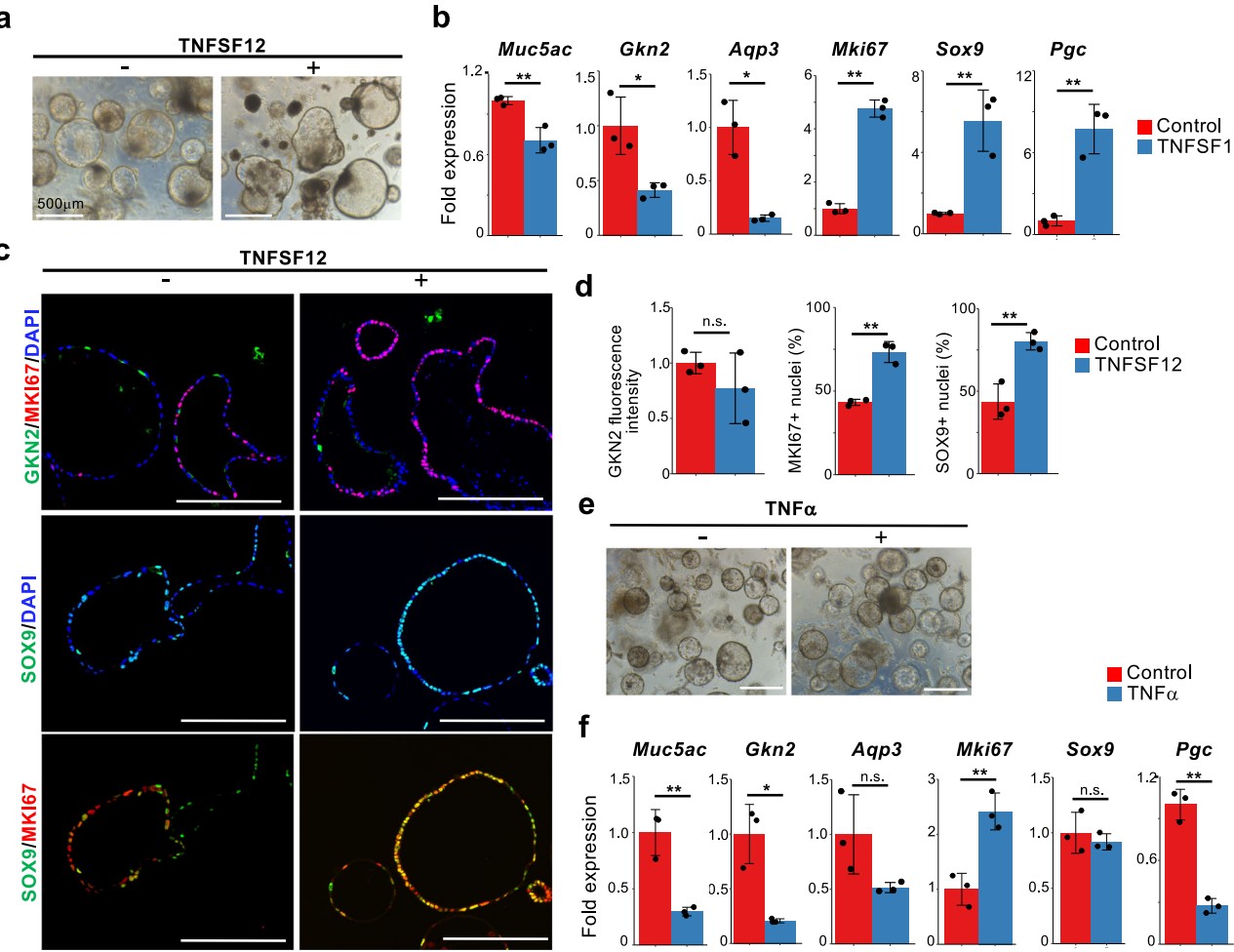

**Fig. 5 | TNFSF12 and TNF maintain isthmus progenitor cells in the undifferentiated state. a** Corpus organoids cultured in the presence or absence of 100 ng/mL TNFSF12 for 6 days. **b** qPCR analysis of gastric epithelial cell marker expression in gastric organoids in (**a**). Data are presented as mean fold changes ± SD ($n = 3$ biologically independent samples). **$p = 0.0068$ for *Muc5ac*; *$p = 0.0197$ for *Gkn2*; *$p = 0.0271$ for *Aqp3*; **$p < 0.0001$ for *Mki67*; **$p = 0.00463$ for *Sox9*; **$p = 0.0032$ for *Pgc*. **c** Immunofluorescence staining of corpus organoids in (**a**) with GKN2 (green), SOX9 (green), MKI67 (red), and DAPI (blue). Scale bar, 200 μm. **d** Quantification of GKN2 fluorescence intensity and percentage of nuclear MKI67+ and SOX9+ in all DAPI+ cells of the gastric organoids in (**c**). Each data point represents the mean value of ten organoids. Data are presented as mean fold changes ±

SD ($n = 3$ biologically independent samples). **$p = 0.0013$ for MKI67+ nuclei; **$p = 0.0060$ for SOX9+ nuclei. **e** Corpus organoids cultured in the presence or absence of 100 ng/mL TNFα. Scale bar, 500 μm. **f** qPCR analysis of gastric epithelial cell marker expression in the gastric organoids in (**e**). Data are presented as mean fold changes ± SD ($n = 3$ biologically independent samples). **$p = 0.0042$ for *Muc5ac*; *$p = 0.0369$ for *Gkn2*; **$p = 0.0058$ for *Mki67*; **$p = 0.0005$ for *Pgc*. Source data are provided as a Source Data file. Significance was calculated by two-tailed Student's *t*-tests for samples with equal variances or two-sided Welch's *t*-tests for samples with unequal variances. *$p < 0.05$; **$p < 0.01$; n.s. (not significant). Data of the control samples are also presented in Fig. 4b.

dependent pit cell differentiation. H&E staining revealed that TGFα also induced morphological changes from a thin proliferating cell to a thick cuboidal morphology (Fig. 4h, i), which is a characteristic of pit cells in vivo. Furthermore, TGFα treatment induced a defined apical-basal polarity in the differentiated gastric organoids. Mature pit cell-specific AQP3 expression was specifically observed at the basolateral membrane just below the adherens junctions stained with phalloidin (Fig. 4j and Supplementary Fig. 22f), which is consistent with the observations in mature adult pit cells in vivo.

A previous study reported that pit cell differentiation was induced by decreasing Wnt3a in culture[45]. The major difference between their culture method and ours is that they used Wnt3a conditioned medium (Wnt3a-CM), and we used recombinant Wnt3a protein (rWnt3a) (Supplementary Fig. 22g). Therefore, we compared the effects of Wnt3A-CM and rWnt3A on the expression of pit, neck, and chief cell markers. The expression level of pit cell markers was quite low in Wnt3a-CM compared to that in rWnt3a (Supplementary Fig. 22h). Importantly, EGF induced the expression of pit cell markers *Muc5ac* and *Gkn2* in rWnt3A medium, but not in Wnt3a-CM medium.

In addition, *Muc6* and *Lgr5* expression levels were more than 100-fold higher in cultures with Wnt3a-CM than in cultures with rWnt3a. Collectively, Wnt3a-CM induced an increase in the expression of neck and chief cell markers and strongly suppressed pit cell differentiation in cultures compared to that by rWnt3a, leading to a different response to EGFR signaling activation.

## NF-κB signaling suppresses the differentiation of isthmus progenitor cells

The enrichment analysis demonstrated the downregulation of TNF-NF-κB signaling during pit cell differentiation (Fig. 2f), suggesting the possibility that TNF-NF-κB signaling regulates isthmus progenitor cells. The previous study reported that IKKβ/NF-κB signaling plays an inhibitory role in *Helicobacter* infection-induced apoptosis and necrosis[46]. However, the role of NF-κB signaling in homeostasis remains elusive. As our ligand-receptor analysis by CellChat identified *Tnfsf12-Tnfrsf12a* as one of the major TNF signaling drivers in isthmus progenitor cells (Fig. 3b, d), we treated gastric organoids with this ligand. TNFSF12 treatment strongly induced the expression of *Mki67*, *Sox9*, and *Pgc* and

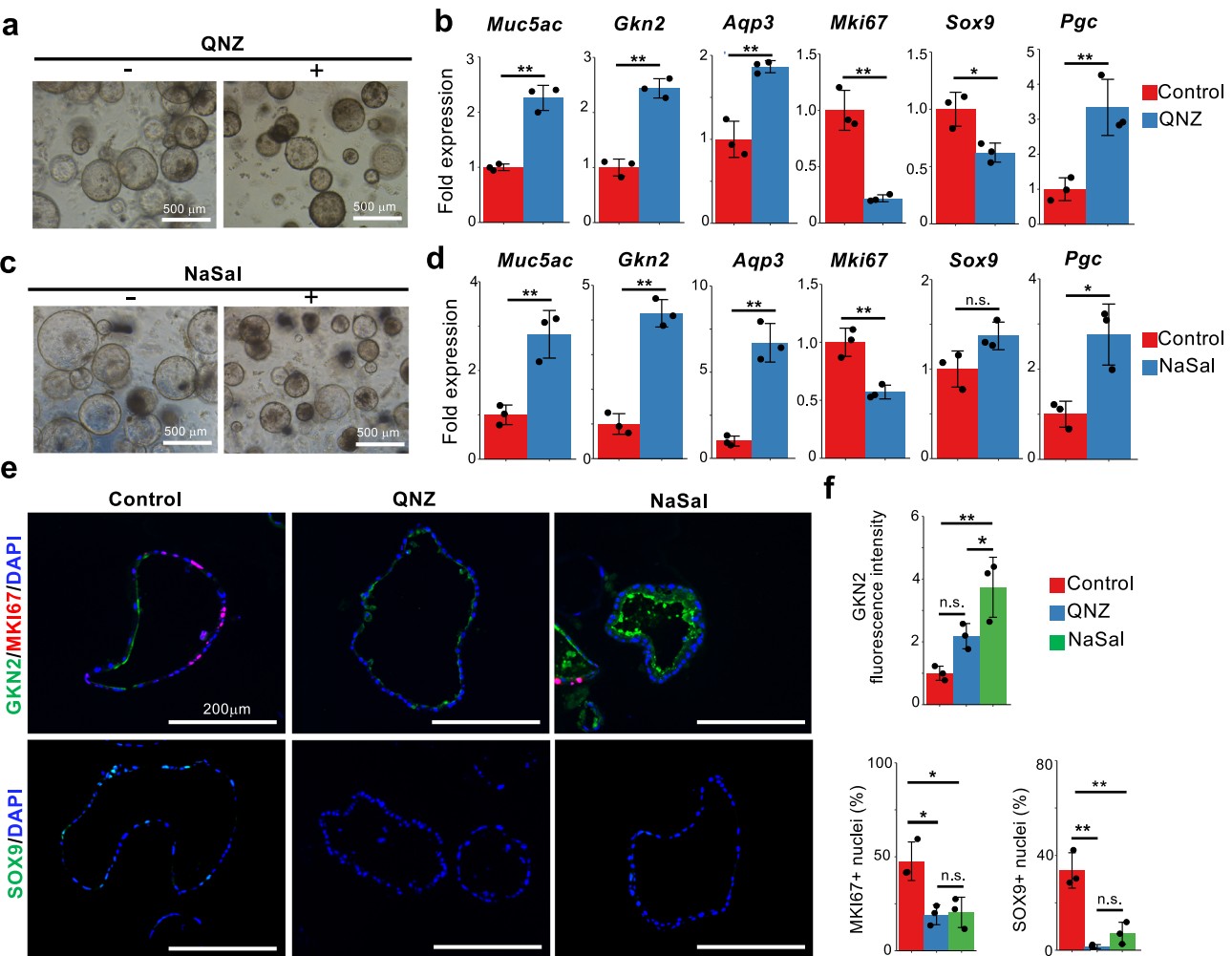

**Fig. 6 | NF-κB inhibition induces pit cell differentiation. a** Corpus organoids cultured in the presence or absence of a NF-κB inhibitor, QNZ (16 nM). **b** qPCR analysis of gastric epithelial cell marker expression in gastric organoids in (**a**). Data are presented as mean fold changes ± SD (*n* = 3 biologically independent samples). **p* = 0.0007 for *MucSac*; **p* = 0.0004 for *Gkn2*; **p* = 0.0027 for *Aqp3*; **p* = 0.0016 for *Mki67*; **p* = 0.0182 for *Sox9*; **p* = 0.0094 for *Pgc*. **c** Corpus organoids cultured in the presence or absence of an NF-κB inhibitor, NaSal (1.25 mM). **d** qPCR analysis of gastric epithelial cell marker expression in gastric organoids in (**c**). Data are presented as mean fold changes ± SD (*n* = 3 biologically independent samples). **p* = 0.0056 for *MucSac*; **p* = 0.0003 for *Gkn2*; **p* = 0.0010 for *Aqp3*; **p* = 0.0054 for *Mki67*; **p* = 0.0141 for *Pgc*. **e** Immunofluorescence staining of corpus organoids with GKN2 (green), MKI67 (red), SOX9 (green), and DAPI (blue). The corpus

organoids were cultured with or without QNZ or NaSal. Scale bar, 200 μm. **f** Quantification of GKN2 fluorescence intensity and the percentage of MKI67+ and SOX9+ cells in the gastric organoids in (**e**). Each data point represents the mean value of at least ten organoids from three biologically independent samples. Data are presented as mean fold changes ± SD. NaSal vs Control **p* = 0.0037, QNZ vs Control *n.s.*, QNZ vs NaSal **p* = 0.0473 for GKN2 fluorescence intensity; **p* = 0.0150, **p* = 0.0121, *n.s.* for MKI67+ nuclei; **p* = 0.0017, **p* = 0.0006, *n.s.* for SOX9+ nuclei. Source data are provided as a Source Data file. Significance was calculated by two-tailed Student's *t*-tests for samples with equal variances or two-sided Welch's *t*-tests for samples with unequal variances in (**b**) and (**d**); significance was determined by one-way ANOVA followed by Tukey's test in (**f**). **p* < 0.05; **p* < 0.01; n.s. not significant.

downregulated pit cell markers (Fig. 5a, b and Supplementary Fig. 23a). Immunofluorescence staining confirmed the increased expression of MKI67, SOX9, and NF-κB in TNFSF12-treated gastric organoids, though GKN2 protein expression displayed no significant change (Fig. 5c, d and Supplementary Fig. 23b, c). Notably, most of the SOX9+ cells in the TNFSF12-treated gastric organoids were MKI67+ (Fig. 5c), suggesting the role of TNFSF12 in maintaining the MKI67+/SOX9+ progenitor cells.

In addition to *Tnfsf12-Tnfrsf12a*, CellChat also identified *TNF-Tnfrsf1a* (Fig. 3b, d) as a major contributor to TNF signaling. Similar to TNFSF12, TNFα treatment increased the expression of progenitor cell markers and decreased the expression of pit cell markers (Fig. 5e, f, Supplementary Fig. 23d–h), although *Sox9* expression showed no significant change. Collectively, TNFSF12 and TNFα maintained the undifferentiated state of isthmus progenitor cells.

In contrast, treatment with the NF-κB inhibitor QNZ increased the expression of pit cell markers and suppressed the expression of *Mki67*

(Fig. 6a, b and Supplementary Fig. 24a). Similar gene expression changes were induced by other NF-κB inhibitors, including sodium salicylate (NaSal) and MG132, albeit at different level (Fig. 6c, d and Supplementary Fig. 24b–d). Immunofluorescence analysis confirmed that NaSal increased GKN2 expression at the protein level and decreased the number of MKI67+ cells and SOX9+ cells (Fig. 6e, f). Although QNZ and NaSal increased the mRNA expression of *Pgc*, its protein expression was not detected (Fig. 6b, d and Supplementary Fig. 24e). These data indicate that NF-κB signaling negatively regulates pit cell differentiation and maintains the isthmus progenitor cells in an undifferentiated state. To further examine whether TNFSF12- and TNFα-mediated maintenance of gastric progenitor cells is mediated by NF-kB, gastric organoids were treated with QNZ in the presence of TNFSF12 or TNFa. QNZ treatment abrogated TNFSF12- and TNFα-mediated decreases in pit cell markers and increases in *Mki67* expression (Supplementary Fig. 24 f–i), suggesting that NF-κB is one of the downstream effectors of TNFSF12 and

TNFα-dependent maintenance of isthmus progenitor cells. Despite the consistent effects of TNFSF12, TNFα, and QNZ on the expression of *Mki67* and pit cell markers, their actions on *Sox9* and *Pgc* were inconsistent, suggesting that additional downstream factors may be involved in the regulation of these genes.

### Downstream mediators of TGFα-mediated pit cell differentiation

We further explored the downstream effectors of EGFR signaling responsible for pit cell differentiation. A pseudotime analysis revealed that the term "actin filament organization" was enriched in pit cell-related genes (Fig. 2e) and that the expression of actin cytoskeleton-related genes, including the subunits of the ARP2/3 complex (*Aprc1a*, *Arpc1b*, *Arpc2*, *Arpc3*, *Arpc4*, and *Arpc5*), were upregulated during pit cell differentiation (Supplementary Data 1). Consistent with this, the ARP2/3 inhibitor CK666 completely blocked TGFα-dependent cell morphological changes; gastric organoids treated with CK666 exhibited an undifferentiated state-specific thin epithelial cell layer, even after treatment with TGFα (Fig. 7a–c). Intriguingly, CK666 downregulated the expression of the pit cell markers and upregulated the expression of *Mki67* and *Sox9* (Fig. 7d). These data suggest that ARP2/3-mediated actin cytoskeletal organization participates in the morphological changes and possibly gene expressions during pit cell differentiation.

The MAPK signaling pathway was also enriched in pit cell differentiation-related genes (Fig. 2e). To examine whether the MAPK pathway is a responsible downstream pathway of TGFα-mediated pit cell differentiation, we treated the gastric organoids with ERK and MEK inhibitors in the presence of TGFα. The ERK inhibitor SCH772984 completely inhibited the expression of pit cell markers and recovered the expression of *Mki67* and *Sox9* (Fig. 7e, f, Supplementary Fig. 25a). Similar to SCH772984, a MEK inhibitor, PD0325901, also suppressed TGFα-mediated gene expression changes (Fig. 7g, h, Supplementary Fig. 25b). Immunofluorescence staining confirmed that PD0325901 suppressed TGFα-induced ERK phosphorylation in the gastric organoids (Fig. 7i, j). Interruption of the other EGFR downstream component STAT3 with Stattic did not affect the expression of pit cell markers in the presence of TGFα (Supplementary Fig. 25c, d), indicating that ERK but not STAT3 is one of the major downstream effectors of TGFα-EGFR-mediated pit cell differentiation in vitro. Immunofluorescence staining of mouse adult corpus tissues confirmed that phosphorylated ERK was specifically detected in mature pit cells and pre-pit cells (Fig. 7k and Supplementary Fig. 25e), indicating that ERK is specifically activated during pit cell differentiation in vivo.

To confirm EGFR signaling-dependent pit cell differentiation in vivo, wild-type mice were administered the EGFR inhibitor erlotinib via oral gavage for 2 weeks, and their stomach tissue was analyzed by immunofluorescence staining. Erlotinib markedly decreased the thickness of GKN2- and AQP3-expressing pit cell region, leading to the luminal shift of parietal cells (Fig. 7l, m). Conversely, the number of MKI67+ cells significantly increased in the mice treated with erlotinib (Fig. 7l, m). These in vivo data further support our in silico and gastric organoid data that EGFR signaling induces pit cell differentiation.

Cumulatively, these in vitro and in vivo experiments validated the in silico-predicted molecular mechanism for gastric epithelial cell differentiation. The EGFR signaling pathway promotes the differentiation of isthmus progenitor cells toward pit cells and concomitantly induces morphological changes, and the NF-κB signaling pathway serves as an anchor for the maintenance of isthmus progenitor cells.

## Discussion

EGFR signaling has been extensively studied in the context of gastric cancers with several EGF receptors, such as HER2 (Erbb2) and EGFR, which are found to be frequently overexpressed due to copy number amplification[47]. In addition, ERK signaling has previously been shown to be involved in atrophy-induced progenitor cell expansion[48]. Therefore, it was unexpected that our single-cell analysis of healthy gastric glands identified EGFR signaling as a differentiation-promoting factor rather than a mitogenic factor. Here, we propose a regulatory mechanism for healthy gastric gland homeostasis. EGFR-ERK signaling promotes the differentiation of isthmus progenitor cells toward pit cells, whereas TNFSF12-NF-κB signaling maintains the isthmus progenitor cells in an undifferentiated state. It would be interesting to examine how NF-κB signaling inhibits EGFR-MAPK signaling to maintain the balance between self-renewal and differentiation of gastric progenitor cells in future studies.

Increased production of TGFα in the gastric mucosa of Ménétrier's disease has been reported[49]. Consistent with this, TGFα-overexpressing mice recapitulated the characteristics of Ménétrier's disease, such as substantial pit cell hyperplasia and a marked reduction of parietal cells and chief cells[19,20,50]. Based on these findings, TGFα signaling has been assumed to be a mitogenic factor that promotes pit cell proliferation[51]. In this study, our single-cell RNA-seq data, organoid-based assay, and in vivo assay clearly indicated that TGFα-EGFR-ERK signaling promotes pit cell differentiation but inhibits pit cell proliferation under healthy gastric gland homeostasis. An early study reported the expressions of TGFα and its receptor transcripts in parietal cell-enriched fractions by northern blotting[52]. However, our single-cell data clearly showed that EGF ligands (*Tgfa*, *Areg*, *Btc*, and *Hbegf*) and receptors (*Egfr* and *Erbb2*) are mostly expressed in the pit cell lineage but rarely in the parietal cells (Fig.3a, d). Our enrichment analysis showing downregulation of EGFR signaling during parietal cell differentiation in healthy gastric glands (Supplementary Fig. 12d) may explain the loss of parietal cell phenotypes in Ménétrier's disease patients, where TGFα is overexpressed. These detailed datasets in healthy gastric glands will help to understand the disease phenotypes in future studies.

Previous studies have reported single-cell transcriptomic analyses of gastric cells[12,42,53,54]. In this study, we leveraged the power of Quartz-Seq2, which has an extremely high gene detection rate with no prominent batch effect (Supplementary Fig. 2). The average gene count in our analysis was 6013 (dataset 1) and 6270 (dataset 2), which is three times higher than the previous 10X Genomics-based study, with an average of 1923 genes[53]. In addition, our reanalysis showed that the average number of detected genes in the recent study[54] was 3474 (Supplementary Fig. 2e), which indicates the high number of detected genes in our datasets. This allowed us to perform enrichment analysis using a considerable number of marker genes. We were also able to capture pseudotime estimation using the FateID algorithm and identify the regulatory pathways of gastric epithelial cell differentiation processes in unprecedented detail.

CellChat and ligand-receptor expression analyses identified cell–cell communication between key signaling pathways. In this study, CellChat data suggested parietal cells, which highly express *Bmpr1b*, *Bmpr2*, and *Acvr2a*, as the prominent receivers of BMP ligands (Fig. 3a and Supplementary Fig. 19c). These data are consistent with the loss of parietal cell phenotype observed in parietal cell-specific NOG-GIN transgenic mice[15]. Although previous studies reported that SHH is expressed in parietal cells[41,55], our single-cell analysis and CellChat results suggested that SHH ligands are secreted from pit and isthmus progenitor cells and act primarily on the receptor-expressing stromal cells (Supplementary Figs. 17b and 20b). Therefore, enhanced gastric epithelial proliferation induced by the SHH signaling inhibitor cyclopamine[55], as well as the pit cell hyperplasia and hypochlorhydria in parietal-specific SHH KO mice, might be explained by the defects in the SHH-responsive stromal cells. CellChat analysis also suggested that IGF signaling is activated in the neck cells in an autocrine manner and a paracrine manner from stromal cells (Fig. 3a and Supplementary Figs. 17a and 19a). A previous study suggested autocrine and negative

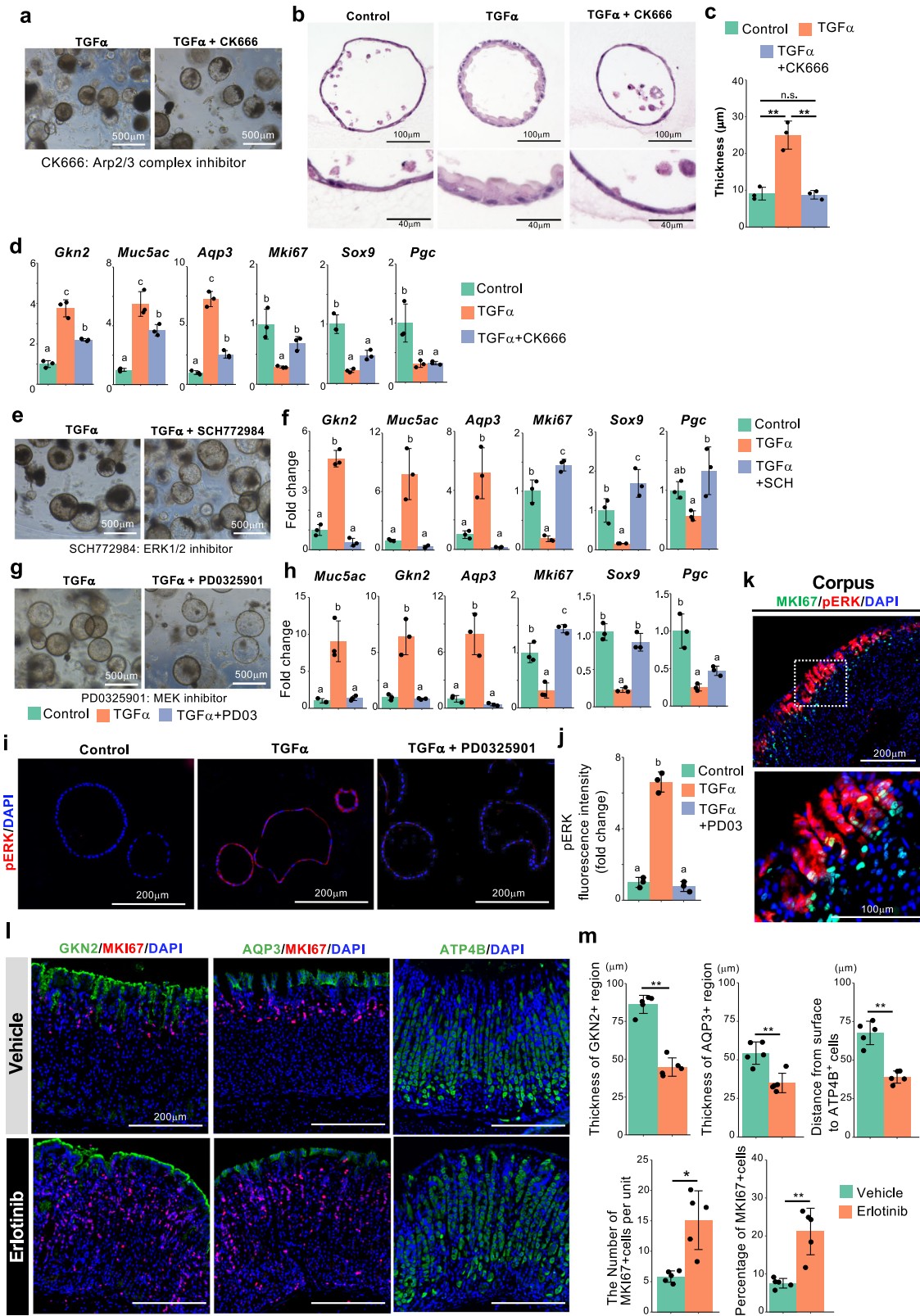

paracrine regulation of IGF signaling based on the expression of *Igf1* and *Igf1r* in gastric progenitors and of its negative regulator *Igfbp2* in parietal cells[43]. It would be interesting to analyze how parietal cells affect neck cell differentiation via IGF signaling regulation.

We identified several transcription factors that were enriched in specific lineages of gastric epithelial cells. For example, *Pparg*, *Pbx1*, and *Ybx2* were enriched in the pit, neck, and parietal cell lineages,

respectively (Fig. 2d and Supplementary Figs. 10c and 12c). However, the precise functions of these factors in gastric epithelial cell differentiation have not yet been elucidated. Previous research demonstrated that PPARγ plays a role in anti-inflammatory responses by inhibiting inflammatory genes, including NF-κB[56], suggesting that PPARγ may promote pit cell differentiation by inhibiting NF-κB. Although *Pbx1* knockout mice show diminished proliferation and

**Fig. 7 | ERK is a downstream effector of TGFα-mediated pit cell differentiation.**
**a** Corpus organoids cultured in the presence of TGFα with or without CK666. The
images are representative of three biologically independent samples. **b** H&E
staining of the organoids in (**a**). Insets show high-magnification images of boxed
areas. **c** Quantification of the thickness of the cells in the organoids in (**b**). Data are
presented as mean ± SD (each data point represents the mean value of six pictures
from $n = 3$ biologically independent samples). **d** qPCR analysis of gastric epithelial
cell marker expression in the organoids in (**a**). Data are presented as mean ± SD
($n = 3$ biologically independent samples). **e** Corpus organoids cultured in the pre-
sence of TGFα with or without SCH772984. **f** qPCR analysis of gastric epithelial cell
marker expression in gastric organoids in (**e**). Data are presented as mean ± SD
($n = 3$ biologically independent samples). **g** Corpus organoids cultured in the pre-
sence of TGFα with or without PD0325901. **h** qPCR analysis of gastric epithelial cell
marker expression in gastric organoids in (**g**). Data are presented as mean ± SD
($n = 3$ biologically independent samples). **i** Immunofluorescence staining of corpus
organoids with pERK and DAPI. Corpus organoids were treated with or without
TGFα and PD0325901. **j** Quantification of of pERK fluorescence intensity in (**i**). Data

are presented as mean ± SD ($n = 3$ biologically independent samples). Each data
point represents the mean value of at least ten organoids. **k** Immunofluorescence
staining of adult mouse corpus tissues with MKI67 (green), pERK (red), and DAPI
(blue). A high-magnification image of the dotted square is shown in the lower panel.
Images are representative of four independent experiments. **l** Immunofluorescence
images of GKN2 (green), ATP4B (green), and MKI67 (red) in the corpus tissues of
the mice treated with erlotinib or vehicle. **m** Quantification of the corpus tissues in
(**l**). The lengths of GKN2$^+$ and AQP3$^+$ regions in each gastric unit, the distance from
the luminal surface to the upper end of ATP4B$^+$ parietal cells, and the number and
percentage of MKI67$^+$ cells were measured. Each data point represents the mean
value of at least 40 gastric units. Data are presented as mean ± SD ($n = 5$ mice).
Source data are provided as a Source Data file. Significance was calculated by two-
tailed Student's $t$-tests for samples with equal variances or two-sided Welch's $t$-tests
for samples with unequal variances in (**m**); significance was calculated by one-way
ANOVA followed by Tukey's post hoc test at the 0.05 significance level in (**c**), (**d**), (**f**),
(**h**), and (**j**). *$p < 0.05$; **$p < 0.01$.

accelerated differentiation of chondrocytes, leading to defects in
endochondral bone development[57], the role of PBX1 in the stomach
remains unclear. *Ybx2* is primarily expressed in germ cells[58], and
inactivation of YBX2 leads to infertility[59,60]. In addition, YBX2 has been
shown to control the activation of brown adipocyte tissue by stabi-
lizing mRNAs encoding proteins involved in mitochondrial function[61],
suggesting that YBX2 may regulate mitochondrial biogenesis during
parietal cell differentiation. Further investigation is required to con-
firm the specific roles of these lineage-enriched transcription factors in
gastric epithelial cell differentiation.

A recent study reported the Iqgap3-Ras-Erk axis-dependent pro-
liferation of isthmus progenitor cells[26]. Our single-cell RNA-seq data
confirmed the preferential expression of *Iqgap3* in progenitor cells
(Supplementary Fig. 4b). In our study, however, ERK regulates pit cell
differentiation rather than progenitor cell proliferation. It would be
interesting to examine how IQGAP3 knockout affects pERK level and
pit cell differentiation under in vivo normal homeostasis.

In conclusion, our analysis revealed the roles of TGFα-EGFR-ERK
signaling in pit cell differentiation and TNFSF12-NF-kB signaling in
isthmus progenitor cell maintenance in healthy gastric gland home-
ostasis. Further exploration of neck and parietal cell differentiation
processes will lead to a deeper understanding of the mechanisms
underlying multilineage differentiation of isthmus progenitor cells.

Certain limitations were noted in this study. Although our in silico
analysis inferred various signaling networks, we did not validate the
signaling pathways involved in neck and parietal cell differentiation. In
our gastric organoids, the differentiated neck cell marker lectin GS-II
and the parietal cell marker ATP4B were not detected. Previous studies
have shown that chief cells can be cultured in gastric organoids[14,62], but
there is no efficient protocol for inducing fully differentiated parietal
cells in gastric organoids. The establishment of culture methods, such
as a sophisticated microchip system suitable for parietal cell differ-
entiation, will be needed to analyze the differentiation mechanisms of
these lineages.

## Methods
### Ethics approval
All animal experiments were performed according to the procedures
approved by the animal welfare and ethical review panel of the Nara
Institute of Science and Technology (approval number 1702 and 2201)
and the RIKEN Animal Experiment Committee.

### Cell isolation for the single-cell analysis
Glandular stomach tissue was isolated from 10-week-old C57BL/6J
mice. Briefly, mice were euthanized by cervical dislocation. The iso-
lated stomach was opened by sagittal dissection and rinsed with
phosphate-buffered saline (PBS). Gastric unit layers were separated

from muscle layers using forceps under a stereoscopic microscope.
For dataset 1, corpus and antrum gastric units were treated with 4 mg/
mL pronase (Roche, Basel, Switzerland)/PBS at 37 °C for 5 min and then
with 0.4 U/mL collagenase (SERVA)/HBSS (Gibco) at 37 °C for 15 min to
weaken cell–cell attachment. The clamps of gastric units were washed
on a 20-μm cell strainer (pluriSelect, Leipzig, Germany) and treated
with 2 mg/mL pronase at 37 °C for 15 min. The dissociated cell sus-
pension was filtered sequentially through 20-μm and 6-μm cell strai-
ners. The single cells retained on the 6-μm cell strainers were
resuspended in 0.1% BSA/PBS (Cell-1). The aggregated cells on the 20-
μm cell strainer were treated with 0.1 mg/mL DNaseI (Roche) and fil-
tered again through a 6-μm cell strainer. The single cells retained on
the filter were resuspended in 0.1% BSA/PBS (Cell-2). The mixture of
Cell-1 and Cell-2 was stained with PI and used for single-cell sorting. For
dataset 2, only corpus gastric units were isolated with some mod-
ification as follows: In the first dissociation step, gastric units were
treated with 0.4 U/mL collagenase 37 °C for 15 min and then treated
with 4 mg/mL pronase/PBS at 37 °C for 5 min. After the resuspension in
0.1% BSA/PBS, the isolated cells were stained with Hoechst 33432 and
then with TO-PRO-3 in order to remove cellular debris and dead cells
using cell sorting. In both datasets, the biological replicate samples
were isolated from three different mice by two different opera-
tors (Fig. 1b).

### Generation of single-cell RNA-seq data
Single-cell RNA-seq was performed using Quartz-Seq2 according to the
previously described methods[24] using v3.1 or v3.2a RT primers, which
have 384 unique cell barcodes with a length of 14 or 15 nucleotides.
Single cells were isolated in 384-well PCR plates with 1 μL of lysis buffer
using the MoFlo Astrios EQ Cell Sorter (Beckman Coulter, Brea, CA,
USA). In dataset 1, PI-positive dead cells were removed, and the singlet
gate population was sorted (Supplementary Fig. 1a). In dataset 2, after
TO-PRO-3-positive dead cells and Hoechst33432-negative cell debris
were removed, the singlet gate population was sorted (Supplementary
Fig. 1b). The removal of cell debris and cell aggregates were confirmed
by the phase contrast microscope (Supplementary Fig. 1c). Summit
software (V6.3.0.16900) was used for flow cytometry data collection
and data visualization. A DNA sequencing library was prepared from
six 384-well PCR plates (2304 wells) in dataset 1 and 12 plates (4608
cells) in dataset 2. The sequencing library was analyzed using the
NextSeq 500/550 High Output v2 Kit (75 cycles) (Illumina). Sequence
specification was as follows: Read1, 22 or 23 cycles; Index1, 6 cycles;
and Read2, 64 or 63 cycles. To obtain the digital expression matrix,
fastq files were processed using Drop-seq tools (version 2.4.1), FASTX-
Toolkit (version 0.0.14), STAR (version 2.7.8a.), and
correct_barcode.py[63]. The GRCm39 primary genome assembly refer-
ence genomes and Gencode Release M27 (GRCm39) were used.

## Quality control

If the detected gene count was less than 2500, the cell was defined as low-quality and excluded from further analyses. Approximately 78 and 13% of the cells were removed from datasets 1 and 2, respectively (Supplementary Fig. 2a, c). Cell populations isolated from tissues contained cells of various sizes and cell debris (Supplementary Fig. 1c). It is difficult to distinguish between small cells and cell debris based on light scattering alone in a cell-sorter. In addition, cell debris cannot be separated by the signal intensity of PI dye because cell debris is not easily stained by the PI dye. Therefore, the high percentage of low-quality cells in dataset 1 might be due to the contamination by cellular debris. In dataset 2, isolated cells were stained with Hoechst 33432, and cellular debris (Hoechst 33432-negative population) was removed at the cell sorting steps, which improved the percentage of the cells that passed the quality control. After the removal of low-quality cells, the average gene count per cell is 6013 in dataset 1 and 6270 in dataset 2 (Supplementary Fig. 2b, d). Almost all gastric cells have less than 20% of mitochondrial UMI counts, while parietal cells (cluster 0 in dataset 1, cluster 8 and 1 in dataset 2) had higher mitochondrial UMI counts, which is consistent with high mitochondria content in parietal cells[64]. In both datasets 1 and 2, the biological replicates prepared by independent operators were sorted into different plates. These different plate-derived samples were uniformly clustered in UMAP (Supplementary Fig. 2e), confirming that there was no obvious batch effect. As for the analysis of non-major gastric cell types (Supplementary Fig. 14a), the cells from datasets 1 and 2 were combined, and the cells with less than 1250 gene counts were removed from the analysis as low-quality cells.

## Single-cell data analysis

Single-cell transcriptome data were obtained from 384-well plates. Secondary analyses, such as cell clustering and marker gene identification, were performed using Seurat4 (version 4.1.0)[65]. Since there was no significant batch effect in the data, no integration process was performed (Supplementary Fig. 2e). To draw stacked violin plots using Seurat4, the custom function StackedVlnPlot was used (https://divingintogeneticsandgenomics.rbind.io/post/stacked-violin-plot-for-visualizing-single-cell-data-in-seurat/). For the analysis of cell–cell communication, we used CellChat (version 1.4.0)[40].

## Pseudotime analysis

To reconstruct the differentiation trajectory of gastric epithelial cells based on single-cell transcriptomes, a pseudotemporal analysis was performed using FateID (version 0.1.9)[29]. The software was installed in R directly from GitHub (https://github.com/dgrun/FateID) using devtools (version 2.3.2; https://cran.r-project.org/web/packages/devtools). As for dataset 1, pseudotemporal ordering of corpus gastric units were constructed using clusters 8, 2, 4, 3, 10, and 99 (0). Antrum pit cells (cluster 1) and antrum pre-pit cells (cluster 6) were excluded for accurate lineage estimation. FateID endpoint clusters were set to corpus pit cells (cluster 8), neck cells (cluster 3), and parietal cells (cluster 0). As for dataset 2, pseudotemporal ordering were constructed using all major epithelial cells, including mature pit cells (cluster 13), pre-pit cells (cluster 4), isthmus progenitor cells (clusters 3, 5, 6, 9, 11), neck cells (clusters 0, 2, 7, 12), chief cells (cluster 10), and parietal cells (cluster 1, 8). FateID endpoint clusters were set to corpus pit cells (cluster 13) (t13), neck cells (cluster 12) (t12), and parietal cells (cluster 1) (t1). The default settings were used for the remaining parameters except prcurve function (trthr=0.65). The resulting trajectory and branches from the computed fate bias were visualized using t-distributed stochastic neighbor embedding (tSNE, used for dataset 1) and Uniform Manifold Approximation and Projection (UMAP, used for dataset 2).

## Hierarchical clustering of gene co-expression nodes

To group gene co-expression nodes with similar patterns of variation in pseudotime, the R package pheatmap (version 1.0.12; http://cran.r-project.org/web/packages/pheatmap) was used to draw heatmaps of average expression levels of genes belonging to each node and to perform hierarchical clustering on a set of nodes. Then similar expression pattern groups were divided using cutree function.

## Corpus gastric organoid culture

Corpus organoids were cultured as described previously[44] with some modifications. Briefly, stomach tissues were incubated with 10 mM EDTA/PBS, and gastric glands were mechanically stripped from the muscle layer using forceps. The isolated gastric glands were dissociated using the dissociation buffer (54.9 mM D-sorbitol and 43.3 mM sucrose in PBS) by gentle shaking for 3 min at room temperature. The gastric glands were washed with PBS, resuspended in Matrigel (Corning, Inc., Corning, NY, USA) on ice, and plated on a 24-well plate. The cells were cultured in the basal medium (advanced DMEM/F12 (Gibco) supplemented with 2 mM GlutaMax (Gibco), 10 mM HEPES, 1 × N-2, 1 × MACS® NeuroBrew-21 (Miltenyi, Bergisch Gladbach, Germany), 100 U/mL penicillin/100 mg/mL streptomycin (Nacalai, Kyoto, Japan), 100 ng/mL Wnt3a (R&D, Minneapolis, MN, USA), 1 mg/mL R-spondin1, 0.1 μM LDN-193189 (Hello Bio, Princeton, NJ, USA), 100 ng/mL FGF10 (Fujifilm Wako Pure Chemical Cooperation, Osaka, Japan), 10 nM gastrin (Sigma, St. Louis, MO, USA), 1 mM N-acetylcysteine (Sigma-Aldrich), and 10 μM Y-27632 (BLD Pharm, Shanghai, China). R-spondin1 was purified from the culture medium of HA-R-Spondin1-Fc 293 cells (Trevigen, Gaithersburg, MD USA) in-house. Two modifications were made in the organoid culture medium from the previous report;[44] a BMP receptor inhibitor, LDN-193189, was used instead of NOGGIN, and EGF was removed to analyze the effect of EGFR signaling on isthmus progenitor cell differentiation. To compare the effects of rWnt3a and Wnt3a-CM, Wnt3a-CM was supplemented instead of rWnt3a.

For the pit cell differentiation assay, corpus organoids were cultured in the basal medium with or without 0.5 μM erlotinib (Combi-Blocks, San Diego, CA, USA), 12.5 ng/mL TGFα (R&D), 1 μM SCH772984 (Cayman Chemical, Ann Arbor, MI, USA), PD0325901 (1 μM) or 50 μM Stattic (BLD Pharm) for 6 days. To analyze changes in cell morphology, corpus organoids were cultured in the basal medium for 3 days and subsequently cultured with 25 ng/mL TGFα and 25 μM CK666 (Sigma-Aldrich) for an additional 3 days. To analyze ERK phosphorylation, corpus organoids were cultured in basal medium with or without 0.4 μM erlotinib and 1 μM PD0325901 for 3 days and subsequently cultured with 12.5 ng/mL TGFα for 45 min. To analyze the function of NF-κB signaling, gastric organoids were cultured in basal medium with 100 ng/mL TNFSF12 (Peprotech, Rocky Hill, NJ, USA) or 100 ng/mL TNFα (Peprotech) for 6 days. To analyze the effect of the NF-κB inhibitor, gastric organoids were cultured in basal medium for 4 days and subsequently cultured with 16 nM QNZ (BLD Pharm), 1.25 mM NaSal (Nacalai), or 3 μM MG132 (Fujifilm Wako Pure Chemical Cooperation) for an additional 2 days. To analyze the effect of the NF-κB inhibitor QNZ in the presence of TNFSF12 or TNFα, gastric organoids were cultured in basal medium with or without TNFSF12 or TNFα for 4 days and subsequently cultured with 80 nM QNZ for an additional 2 days.

## Mice

All mice were fed ad libitum and housed in a specific pathogen-free facility under the following conditions: a 12-h light-dark cycle, ambient temperature of 23–24 °C, and 50–70% humidity. For erlotinib treatment, C57BL/6J mice (7–10 weeks old) were orally gavaged with erlotinib (100 mg/kg in 6% Captisol) or vehicle (6% Captisol) daily for 2 weeks.

## Immunofluorescence

To prepare frozen sections for immunofluorescence and H&E staining, gastric organoids in Matrigel were fixed with 10% formalin/ PBS overnight at 4 °C, washed with PBS, and embedded in O.C.T. compound (Tissue-Tek). The frozen section was prepared at 5 μm thickness by a Cryostat CM1860 (Leica, Nußloch, Germany). For the immunostaining of adult mouse stomach tissues and gastric organoids, frozen sections were microwaved in 10 mM citrate buffer (pH 6.0) for 10 min and blocked with PBS with 5% fetal bovine serum for 1 h at room temperature. Sections were then incubated with primary antibodies overnight at 4 °C, washed with PBS, and incubated with secondary antibodies for 1 h at room temperature. Primary and secondary antibodies are listed in the "Reporting Summary". Immunostaining results were visualized using a KEYENCE micro- scope (BZ-X710; Osaka, Japan) or a confocal laser scanning micro- scope (LSM710, Oberkochen, Germany). The fluorescence signal intensity of GKN2, pERK, and NF-kB, the number of MKi67$^+$ and SOX9$^+$ cells, and the thickness of corpus organoids were quantified using Fiji (National Institutes of Health).

## Antibodies

AGR2 (Abcam, ab76473, 1:400), ATP4B (ATLAS ANTIBODIES, HPA045400, 1:400), AQP3 (Abcam, ab125219, 1:400), BASP1 (Noves Biologicals, NBP1-68958, 1:1000), CD4 (Santa Cruz, sc-13573, 1:100), CD31 (BD Pharmingen, 557355, 1:100), Cleaved Caspase-3 (Cell Signal- ing, D175, 1:400), EGFR (Abcam, ab52894, 1:400), pERK (Cell Signaling, 4370S, 1:200), GKN2 (Abcan, ab188866, 1:800), IGF-1R (Cell Signaling, 3027S, 1:200), KRT20 (Proteintech, 17329-1-AP, 1:400), KRT7 (Abcam, ab181598, 1:400), MKI67 (Abcam, ab15580, 1:400), MKI67 (Thermo Fisher, 14-5698-82, 1:1000), NFKB1 (Abcam, ab32360, 1:400), PDGFRa (Invitrogen, 14-1401-81, 1:200), PGC (Abcam, ab31464, 1:2000), SMA (Novus Biologicals, NBP1-30894, 1:400), Somatostatin (Santa Cruz, sc-7819, 1:800), SOX9 (Abcam, ab185230, 1:400). Donkey anti-Rat IgG (H +L) Alexa Fluor 488 (Invitrogen, A-21208, 1:1000), Donkey anti Rabbit IgG (H+L) Alexa Fluor 488 (Invitrogen, A-21206, 1:1000), Donkey anti Rabbit IgG (H+L) Alexa Fluor 555 (Invitrogen, A-31572, 1:1000), Donkey anti-Sheep IgG (H+L) Alexa Fluor 594 (Invirogen, A-11016, 1:1000), Anti- IgG (H+L chain) (Rabbit) pAb-HRP (MBL, 458, 1:10,000).

## Gene expression analysis by quantitative PCR

Total RNA was isolated from gastric organoids using ReliaPrep™ RNA Miniprep Systems (Promega, Madison, WI, USA) and was used to synthesize cDNA using ReverTra Ace qPCR RT Master Mix (TOYOBO, Osaka, Japan) following the manufacturer's protocol. qRT-PCR was performed using THUNDERBIRD qPCR Mix (TOYOBO) and the CFX96 Real-Time System (BIO-RAD, Hercules, CA, USA). Gene expression levels were normalized to β-actin expression in all experiments. Pri- mers are listed in Supplementary Table 1.

## Western blotting

Isolated corpus and antrum gastric glands were lysed using ice-cold lysis buffer (25 mM Tris-HCl, 150 mM NaCl, 1% NP-40, 0.5% Na-Deoxy- cholate, 0.1% SDS, 1 mM EDTA) with a protease inhibitor cocktail (Complete EDTA free, Sigma-Aldrich) and rotated at 4 °C for 1 h. Lysate was centrifuged at 4 °C for 20 min, and the supernatant was used for western blotting. The protein concentration of the supernatant was measured by Bradford protein assay. The supernatant was mixed with sample buffer and incubated at 95 °C for 5 min. Samples were sepa- rated on SDS-PAGE gels and transferred to Nitrocellulose membranes. The membranes were blocked by Blocking One (Nacalai) at room temperature for 1 h and incubated with primary antibodies overnight at 4 °C. The membranes were washed with TBST and incubated with HRP-conjugated secondary antibody at room temperature for 1 h. After washing with TBST, the membranes were incubated with SuperSignal West Femto Maximum Sensitivity Substrate (Thermo Fisher Scientific, Waltham, MA, USA) and imaged using a gel imager (LAS-4000, Fujifilm, Tokyo, JAPAN).

## Statistics and reproducibility

Comparisons were performed with the two-tailed Student's $t$-test for samples with equal variances and Welch's $t$-test for samples with unequal variances. For multiple comparison, one-way ANOVA followed by Tukey's test was used. The calculation was performed using R (version 4.2.1). Values of $p < 0.05$ were considered statistically sig- nificant. Significance was indicated in all experiments as follows: *$p < 0.05$; **$p < 0.01$. No statistical method was used to predetermine the sample size. The experiments were not randomized. The investi- gators were not blinded to allocation during experiments and out- come assessment.

## Reporting summary

Further information on research design is available in the Nature Portfolio Reporting Summary linked to this article.

## Data availability

Raw data and digital expression matrix for the single-cell RNA-seq in this study have been deposited in Gene Expression Omnibus (GEO) under accession code GSE216139. Source data are provided with this paper.

## Code availability

R scripts for Seurat processing are available on GitHub [https://github. com/satijalab/Seurat]. StackedVlnPlot is available on [https:// divingintogeneticsandgenomics.rbind.io/post/stacked-violin-plot-for- visualizing-single-cell-data-in-seurat/]. FateID for R is available on GitHub [https://github.com/dgrun/FateID]. Devtools and pheatmap are available on [https://cran.r-project.org/web/packages/devtools] and [http://cran.r-project.org/web/packages/pheatmap], respectively. correct_barcode.py was deposited in Zenodo (https://doi.org/10.5281/ zenodo.1118222)[63].

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

## Acknowledgements

This work was supported by JSPS KAKENHI Grant Numbers JP17H00861, JP20H04505 (to A.K.), JP19K10029, JP22K06789 (to H.T.), and JP19H03214 (to Y.S.); LOTTE Research Promotion Grant from the LOTTE foundation (to H.T.); and funds from JST CREST (JPMJCR16G3) and the Projects for Technological Development, Research Center Network for Realization of Regenerative Medicine from the Japan Agency for Medical Research and Development (AMED, JP21bm0404073 (to I.N.) and JP21lm0203014 (to A.K.)). This study was also supported by the Joint Usage/Research Program of Medical Research Institute, Tokyo Medical and Dental University and Medical Research Center Initiative for High Depth Omics. We thank Ms. Yui Fukatsu for her technical assistance. We thank Dr. Mariko Yamane for her comments on the data analysis. We thank the members of the Laboratory for Bioinformatics Research at the RIKEN Center for Biosystems Dynamics Research, Manabu Ishii, Akihiro Matsushima, and Takumi Ichikawa, for IT infrastructure management.

## Author contributions

Conceptualization, H.T., Y.S., I.N., and A.K.; Investigation and formal analysis, H.T., Y.S., M.Y., K.T., Y.I., T.H., and A.I.; Writing—review & editing, H.T., Y.S., M.Y., T.H., I.N., and A.K.; Supervision, A.K. and I.N., Funding acquisition, H.T., Y.S., I.N., and A.K.

## Competing interests

Y.S. and I.N. consult for Knowledge Palette, Inc. and are on their Scientific Advisory Board. The other authors declare no competing interests.
