## [Peer Review File · Nature Communications]

Single-cell transcriptomics uncovers EGFR signaling-mediated gastric progenitor cell differentiation in stomach homeostasisREVIEWER COMMENTS

Reviewer #1 (Remarks to the Author):

In this study by Takada et al., the authors demonstrate a detailed single-cell transcriptomic analysis of the gastric corpus and antrum at homeostasis. They highlight a Quartz-Seq2 technique as well as a FateID to determine cell lineage transcriptional changes. Through this analysis, the authors were able to determine preferentially expressed pathways during gastric cell lineage specification. In detail, they showed that the EGFR-ERK pathway was important in driving pit cell differentiation, and the NF-kappaB pathway preferentially increased undifferentiated cells through pit cell de-differentiation. The novelty of this paper lies in the new single-cell RNA technique and determination of cell lineage transcriptional changes with validation in an in vitro organoid system. Stronger in vivo data to support their in vitro findings would greatly strengthen the impact of this study.

Major Comments:

1. The authors state in the introduction that “These progenitor cells then differentiate into multiple epithelial cell lineages, including pit cells, neck cells, parietal cells, chief cells, and endocrine cells, which migrate bidirectionally from the isthmus to the luminal surface and to the gland base 8-10. Neck cells in the corpus function not only as mucus-secreting cells, but also as progenitors for mature chief cells 11,12.”. While this has been the prevalent accepted theory, more recent work including PMID 31598973 and 31422913 using single cell transcriptomic and a variety of in vivo lineage tracing techniques have shown that the basal chief cells may represent an additional “independent zone” that is not dependent on the canonical isthmal stem cell for maintenance. These data do not immediately effect the conclusions of the current work, as much of this paper explores the relationship between the isthmal stem cell and the pit lineage. However, it does bring up how the FateID differentiation trajectory analyses that the authors performed on the differentiation trajectory from isthmus cells to neck and to chief cells should be interpreted.

In addition, the bulk of this paper focuses on pit cell fate decisions and the pathways discovered during this process bioinformatically. The authors may want to consider refocusing the manuscript to highlight these data in more detail, and moving the parietal and chief cell lineage analyses (Fig 3,4) to the supplemental data. Especially, as the authors do not show the ability to manipulate these parietal and chief cell lineage pathways (insulin signaling and NOTCH signaling) in their organoid system (“no efficient protocols for the induction of fully differentiated chief cells and parietal cells in gastric organoids”).

2. The authors describe their QC for the RNA-seq experiment using 2,500 gene counts and >40% of mitochondrial UMI counts. In other studies (PMID 31422913), cells with greater than 10-20% of mitochondrial UMI counts are excluded. Can the authors explain their more liberal inclusion criteria?

3. The authors use their single-cell data set to define and identify mature pit cell markers such as AQP3. In Figure 1D, there is double immunofluorescence showing costaining of AQP3 and chief cell marker, PGC. The authors should consider repeating this experiment with AQP3 and a pan-pit cell marker such as GKN2 or MUC5AC. This would more definitively demonstrate that AQP3 expression is limited to a specific more “mature” pit cell population.

KRT7 and KRT20 experiments should likewise have co-staining with a pan-pit marker to again show that there are definitive GKN2/MUC5AC positive pit cells in the corpus region that are KRT7/20 negative.

Finally, the authors should consider performing similar in vivo protein expression correlation experiments with their cluster 8/Corpus-specific pit cell markers SCTR and BASP1.

4. In the fourth paragraph of the results section, the authors discuss some unexpected results from their single-cell dataset including the finding that PGC and TFF2 transcriptional expression is more broad than canonically thought. Many of these findings have been shown before especially for the TFF2 expression (PMID 20708616). Consider moving this paragraph to the discussion section.

It is also unclear what the authors mean by this sentence “These results suggest that Pgc expression is an indicator of the chief cell lineage rather than a cell type-specific marker”. They show decreasing Pgc expression in the pit cells which are not considered part of the chief cell lineage. In addition, comment #1 sheds doubt on a direct isthmal stem cell to chief cell connection.

5. The authors state that the ACTA2+ cells are primarily located at the bottom of the gastric glands, however the staining for the ACTA2, at least for the corpus, shows that it is expressed at the isthmal/pit region. Please clarify this discrepancy.

6. For Figure 2D and subsequent similar figures, can the authors explain the “Expression pattern” images? Specifically, what do the axis and differing colors denote?

7. In Figure 3, the authors assess pathways enriched in the chief cell lineage. Please see point #1. In addition, the authors state that their “observations suggest that EGFR signaling is activated during pit cell differentiation but is suppressed during neck/chief cell differentiation”. It is unclear how this conclusion is reached as according to 3D and 3E, the EGFR1 pathway is enriched in Tree I/II which represents the isthmal precursor cells that are probably akin to Tree III in Figure 2E. The EGFR pathway is not enriched in Figure 3E Tree III/IV which more likely represent the proposed neck/chief cell differentiation pathway.

8. The authors focus on EGFR and NF-kappaB signaling in their organoid work. In general, the NF-kB portion of the manuscript is at once novel/intriguing and very immature. It is mostly conjecture with TNFa and a single drug, whereas this is obviously a complex signaling pathway with TNFa only one input

(and with TNF α affecting other pathways as well). They should consider including ligand/receptor analysis for the NF-kappaB pathway as well in Figure 5. It would strengthen their conclusions to have NF-kappaB pathway (ligand/receptors) be increased preferentially in cluster 1 compared to the 2/8 mature pit clusters. They should also use other agents: activators and inhibitors.

9. The authors functionally investigate the consequences on lineage differentiation using an in vitro organoid culture technique. They mention that they “used a well-established gastric organoid system, with certain modifications”. Based on their Methods section, they have omitted Noggin and EGF in the culture conditions. These were included in the culture conditions for the previously well-established protocols cited in references 34 and 35. Can the authors explain the decision to omit these factors and the consequences or anticipated consequences for doing so?

10. Related to point #9, Bartfield et al. showed as the authors indicated that at least with human stomach organoids that they were able to culture pit, neck, chief, and maybe parietal cells in reference 35. The authors show IF staining of KI-67 and GKN2. Are any other lineage markers present in these gastric organoids? If they are not, is this a consequence of their culture condition modifications discussed above?

If only undifferentiated dividing cells and pit-like cells are the only in vitro populations in their organoids, then point #1 (focusing more on the pit cell lineage and pathways, rather than additional lineages that can not be assessed using their in vitro system) is further supported.

11. For their in vitro work with inhibitors and activators of the various pathways shown in Figure 7 and 8, it would be worthwhile to include data (eg possibly western blots for activated downstream targets of EGFR pathway) to show that the treatments are affecting the pathways as the authors intended.

12. The authors show that manipulation of the EGFR pathway and NF-kappaB pathways increases the transcriptional profile of the neck-chief cell lineage (Figure 6). Is this reflected at the protein level? For example, are the authors able to detect this increase in these markers (SOX9 and PGC) with EGFR inhibition?

13. The authors describe effects of apical-basal polarity through in Figure 6J through activation of the EGFR pathway using TGFA. The authors should consider providing higher resolution confocal imaging here to highlight this.

14. These organoid data do show certain transcriptional and protein level changes with pathway manipulation. To fully make the case that these pathways are important for pit cell lineage specification, the authors should consider in vivo experiments such as treating WT mice pharmacologically with the

same treatments as the organoids and showing similar effects. These data would be especially important in light of the recent study PMID 33951440. The data in Figure 7G are an excellent start to confirming their organoid work in vivo, but further experiments should be considered.

15. For the experiments in Figure 7, please include untreated organoid controls for reference. This is especially important to understand the magnitude of changes as the authors are using relative change ratios here.

16. In addition to the citations highlighted above that should be discussed, there are several previous studies that could be integrated with the findings here (note: the authors have done a reasonable job of citation here, but when so many pathways and cells are touched on, it is difficult).

-NF- κ B signaling in gastric epithelium has previously been addressed in vivo: PMID 19962981

- The papers the authors interpret ref 17 as implying “Activation of NOTCH signaling in parietal cells induces the dedifferentiation of parietal cells into undifferentiated progenitor cells and ultimately leads to adenomas” and “cell-specific activation of NOTCH signaling dedifferentiates parietal cells into progenitor cells with the ability to differentiate...” which is a bit misleading. In ref. 17, the authors activated constitutive Notch using a constitutive promoter in the parietal cell lineage, but that promoter can be active in progenitors, so another (perhaps more likely) interpretation of those findings is that activation of Notch during parietal cell differentiation can lock cells in a progenitor state rather than dedifferentiate mature cells.

-“ Its ligand, Igf1, and an upstream regulator of Igf1, Ghr, were also highly expressed in neck and chief cells. These expression patterns were consistent with the results of the enrichment analysis, which showed that genes related to neck/chief and parietal cell differentiation but not pit cell differentiation are enriched in the insulin signaling pathway” There was an early -omics study also indicating a progenitor and parietal cell feedback loop of IGF signaling with IGF and IGF α r in progenitor autocrine signaling and IGF β P2 from parietal cells suppressing the IGF pathway: PMID 12409607

-IQGAP3 has emerged as a key isthmal gene that can regulate RAS (PMID 33293280), which of course is critical in ERK signaling (and downstream of the EGFR signaling highlighted here). Was IQGAP3 found? There have been several other papers on K-Ras and gastric epithelial homeostasis (PMID 26677984), as well as earlier literature showing effects of inhibiting pERK on homeostasis (PMID 23589310), which should probably be referenced.

The authors should also relate their work in the context of other single-cell transcriptomic analyses of the stomach (PMID 31481545 and 31422913). To highlight the differences in their single cell technique compared to the Chromium based techniques, and also discuss the strength of their FateID analyses

Finally, there was a recent study highlighting pit-parietal cell differentiation decisions involving AMPK and KLF4, which would be informative to integrate: PMID 32243780.

Minor Comments:

1. Introduction, third paragraph, second sentence, please add “cells” after parietal, and consider changing “numbers” to “number of transitional cells”.
2. Introduction, third paragraph, fifth sentence, please consider changing TGFa transgenic mice to “TGFa overexpressing transgenic mice”.
3. Introduction, fourth paragraph, first sentence, please add “protocols” after single-cell RNA sequencing.
4. For Figure 1, can the authors increase the resolution of the inset pictures for 1E. In addition, for 1D, there appears to be decreased resolution for KRT7/PGC and KRT20/PGC corpus pictures.
5. Page 15, last line “HE” should be “H&E”.
6. For Figure 7G, it is difficult to appreciate for the Corpus picture that the pERK positive cells are preferentially the pre-pit cells. Decreasing the gain or using confocal microscopy may provide more conclusive data.
7. The authors discuss the role of EGFR signaling in cancer as an analogous pathway during normal stomach homeostasis. This is unlikely to be the case and the link between the role of EGFR in cancer and at homeostasis should be de-emphasized. RTK-Ras pathway activation per TCGA (PMID 25079317) work is usually a late consequence in the CIN subtype of cancers as oftentimes result of, as the authors correctly state, “copy number amplification”. This would not be a transcription factor mediated differentiation process as in normal stomach homeostasis. In addition, GC is preceded by intestinal metaplasia and SPEM through the Correa sequence. And eventual activation of the EGFR pathway would be effecting a different population of cells than in the normal stomach.
8. Page 22, last sentence of the first paragraph, “out” should be replaced with “our”.

Reviewer #2 (Remarks to the Author):

05_04_2021

In this manuscript titled "EGFR signaling is a potent inducer of differentiation but not proliferation of gastric progenitor cells in stomach epithelial homeostasis", the authors used the single-cell RNA sequencing method to obtain gene expression profiles for 15 cell types in normal mouse gastric cell. The authors show that two types of pit cells and stromal cells which characterize the gastric unit in health mouse stomach cells. Moreover, the author also demonstrated the regulatory pathway involved in differentiation from isthmus progenitor cells to pit cell and neck/chief cells. EGFR signaling promotes pit cells differentiation and suppresses neck and chief cells fate decisions. The findings of this work have translational potential to gastric disease patients, as the defects in gastric progenitor cell differentiation are associated with gastric disorder and even gastric cancer. The study is interesting; however, the findings rely on a single health mouse model, and the cell differentiation measurement could include additional measures.

Major comments:

- This study is overall descriptive, and the conclusion was drawn mainly based on single cell sequencing data analysis. Now days, single cell RNA sequencing is quite high throughput, a few thousand cells can be easily sequenced at a relative low cost for each sample, and fresh samples are easily collected from mouse experiments. However, only a total of 493 cells were analyzed in the entire study and these 493 cells were separated into 17 clusters to infer epithelial and stromal cell lineages. Given that the epithelial cell lineages are highly diverse, a clear limit of this study is that only a very small number of cells were analyzed for each cell lineages and therefore the results may not be reproducible, and the authors should analyze more cells from different mouse models.
- Another major limit of this study is the experimental design. The authors tried to profile the expression dynamics of progenitor cell differentiation toward mature epithelial cell. However, longitudinal sampling was not performed, and the authors only analyzed samples collected at a single timepoint, which represents only a snapshot of this dynamic process. The authors tried to infer expression dynamics using pseudotemporal analysis, however the pseudotemporal analysis is mainly knowledge based and the results can change totally with different parameter setting. It is hard to tell whether the trajectory results are reproducible.
- a key part of this study is single cell pathway enrichment analysis to infer the molecular drivers of lineage differentiation and cell fate regulation. A pathway usually contains up to a hundred or few hundred genes, given a high dropout rate of single cell sequencing, only a small number of genes can be detected, make the pathway analysis very challenging especially when a small number of cells are analyzed, which may produce uncertain results with a lot of noise. However, no validation experiments were performed in larger cohort.

- The authors performed organoid experiments to validate the molecular mechanism for gastric epithelial cell differentiation. The organoid experiments are not ideal for such purpose because of the striking differences in their 3D structures, tissue layout, and morphology. The authors should not infer the lineage differentiation solely based on marker genes the cells expressed without taking into consideration the tissue structure and cell morphology and cell function. The function of the cells are not evaluated.
- As for data processing, the filter used in this study is not proper, the authors retained cells with up to 40% of mitochondrial UMI counts for subsequent analysis. the cutoff 40% seems a bit high, as most of other studies used 15% or at most 20%. Therefore, the authors probably included some dying cells or cells in poor quality for analysis, and these cells may impact clustering and cell type identification.
- No method describe how the cell doublets were handled and it is unclear whether the doublets (should be removed) were included in their analysis.

Minor comments:

- Figure 1D: The scale bar of the immunofluorescences stain pit cell makers is not consistent in Corpus and Antrum tissues. Please clarify.
- Figure 6H: The scale bar of the HE staining is not accurate in low and high amplification. Please clarify.
- Figures 7: The author further explore the downstream gene of EGFR signaling which responsible for pit cell differentiation, based on pathway analysis, the ARP2/3 inhibitor CK666 were treated with gastric organoid. However, although the author compared the organoid and differentiation makers between nontreated vs TGFa in Figure6, the negative control or untreated group should also be included in Figure 7.
- Again, obviously, the scale bar for the Fig.7B for the top and low panels is not accurate.

Reviewer #3 (Remarks to the Author):

This study describes results from single-cell RNA sequencing method, Quartz-Seq2, to obtain gene expression profiles from different cell types in healthy mouse stomachs. Cells are clustered to identify different subsets and pseudotemporal analyses are done to predict development of different subsets, and differential gene expression and 3-D organoid studies are used to predict important pathways for

gastric epithelial cell differentiation. The data are interpreted as evidence that EGFR signaling is involved in promoting the development from progenitor cells to mucous producing pit cells.

Weaknesses of the study include the limited number of cells (493 total) analyzed and pseudotemporal analyses are done on a limited subset of these cells, so these are even smaller datasets. The cells appear to be from a mixture of both the corpus and antrum, which is a bit confusing as the focus of the paper is on corpus cells. The results are descriptive in nature, differentiation pathways are “suggested” “putative” based on differential gene expression analyses of a limited dataset of epithelial cells. Organoids are used to investigate possible signals involved in differentiation, but these are problematic. For example, changes in organoids are assumed to be caused by altered differentiation of stem cells but expansion of existing pit-like cells cannot be excluded. The author’s state that to the best of their knowledge this is the first study to generate gene expression analyses of whole gastric units, but a review of the literature identified at least 3 published studies in mouse and humans PMID: 31481545, PMID: 32835664, PMID: 32532891, PMID: 33691112. The several other reports of single cell data generated from the stomach limit the novelty of data presented. This paper does provide an additional dataset for the field, however the conclusions are speculative and may have limited impact on the community and wider field.

Reviewer #4 (Remarks to the Author):

Takada et al. used a single cell RNA sequencing approach to analyze the gene expression profile in normal mouse gastric units. They found 15 different cell types revealing expression dynamics of progenitor cell differentiation toward the pit, neck, chief and the parietal cell lineages. Pathway analyses and subsequent organoid experiments revealed that the EGFR ERK signaling pathway promoted pit cell differentiation, while the NFκB pathway maintained gastric progenitor cells in an undifferentiated state. They conclude that the activation of the EGFR signaling pathway results in a differentiation of isthmus cells and does not have a mitogenic function.

The manuscript is well written and the results are very convincing, especially as the authors functionally validated their scRNA data in gastric organoids. Some issues remain that needs to be addressed:

Major:

The scRNA (as far as this reviewer understood) is based solely on a single scRNA run, and only 493 cells were further analyzed. The results in only a few cells per identified “cell type”. The data seems valid, as many known markers are found allowing a clear separation of known cell types. Nevertheless, the

authors claim to have established a unique expression atlas of the stomach glands. They indeed also find novel interesting genes to be expressed in the known cell types. Nevertheless, it is necessary to perform a second independent run of scRNA analysis from independent mice to really convincingly “... generate a systematic atlas of gene expression profiles...” as stated by the authors. The two runs should be compared to document consistency, and the data of both runs then analyzed together to increase the number of cells per identified cell type.

Figure 5:

Ligand and receptor expression patterns are coming from the same data set as the pathway enrichment analysis. Confirm findings of pathway enrichment in specific cell types by double immunohistochemistry.

Figure 6:

- The authors cite a study of HUMAN gastric organoids (Bartfeld et al) and state that these organoids express markers of pit, neck, and chief cells. Actually, the initial report of gastric organoids (Barker et al, Cell Stem Cell, 2010) reported that gastric organoids express markers of neck (Muc6) and chief cells (GIF and PepC). And these organoids did NOT express pit cell markers under normal growth conditions. Only in Wnt3a reduction medium pit cells were formed. As the authors use similar culturing conditions as published by Barker et al, this has to be resolved.
- Correct citation Bartfeld et al to Barker et al or other murine study referred to
- Are those organoids established from corpus glands or antrum glands? It is referred to later as corpus organoids, but please state in the beginning.
- The authors mention slight modifications to the original protocol. Please specify.
- Show long term culturing of the organoids, e.g. passaging over >10 passages to confirm long term maintenance of organoids in used medium
- Figure 6A/S9B ‘When the gastric organoids were cultured with TGFalpha for 6 days, many organoids became dark in color, suggesting that morphological changes were induced’: This is not really clearly visible in the provided pictures. Darkening of organoids often is due to apoptosis and cells being shed into the lumen. Did the author’s check apoptotic markers to exclude a morphological change due to emerging apoptosis? And what is the effect of long term culturing with Tgfallpha? Can they be maintained long term? Or does the differentiation towards pit cells stop their proliferation?
- Figure 6B: What about the other neck cell makers (Muc6, Tff2) and chief cell markers (Gif, Troy, Lgr5)?
- Figure 6C: Please add stainings for Muc6/GSII and Gif/Pgc on the organoids to confirm finding of mainly pit cells, and not chief / neck cells in their organoid cultures.
- Figure 6F-G: The authors need to prove that the EGFR pathway is indeed inhibited upon erlotinib treatment (western blot / IHC) and show by IHC the validity of qPCR data.
- Figure 6G: please add neck cell markers and more chief cell markers incl. Lgr5/Troy

- Figure 6K-L: The authors need to show that the NF κ B signaling pathway is indeed upregulated after TNF α treatment (western blot / IHC) and by IHC that the pit cells are indeed gone in the organoids.
- Figure 6L: add more chief and neck cell markers.

Figure 7:

- Figure E-F: Again the authors should make sure that their treatment with the ERK inhibitor downregulated the ERK signaling cascade by protein data.

Figure S9G: add more chief and neck cell markers.

Minor:

Figure 1E: the expression of ACTA2 at the bottom of glands seems to be only true in the antrum, while in the corpus the expression seems also to be more prominent in the pit region, just as PDGFR α . Please comment and possibly specify/correct (also in the discussion p. 22).

Figure S3: Please shown other putative stem cell markers: CCK2R, Axin2, Lgr5, Aqp5, eR1+, Troy

Figure 5B Please comment the high expression of Igf signaling ligands in the stroma of gastric glands

Are the TFs presented in Figure 2, 3 and 4 the only ones or selected in one way or the other?

The authors identified for various cell lineages potentially novel cell type specific markers. They might want to discuss these markers and potentially published literature in the discussion part.

Citation. "TGF α -mediated gene regulation was repeatedly observed with passaged corpus organoids (Figures S9A and S9B)." -> not sure what the authors want to say with this sentence and the referred figures (maybe mistake, should be S9B and S9C?). What does "passaged" mean?

Citation "Similarly, the TGF α -dependent induction of pit cell markers and the decrease in isthmus progenitor cell markers were also observed in the antrum organoids (Figures S9C and S9D)." -> Fig. S9D and S9E?

S10: Please specify how genes for the clustering (“EGFR downstream effectors”) were selected.

Typo on page 16: (Figures s 6K and 6L)

Citation Discussion “Although the previous report identified Tgfa expression in parietal and chief cells...”
Which previous report?

Discussion: “Currently there are no efficient protocols for the induction of fully differentiated chief cells...” Chief cells have been cultured successfully in organoids derived from Troy cells (Stange et al, Cell, 2013). Only parietal cells have not been successfully cultured in organoids.

REVIEWER COMMENTS

Thank you very much for providing precious comments and suggestions for our manuscript. We answered all the questions raised by reviewers as described below. In the revised manuscript we indicated **the revised descriptions following reviewers' suggestion in red** and **the newly added sentences after additional experiments in blue**, respectively.

Reviewer #1 (Remarks to the Author):

In this study by Takada et al., the authors demonstrate a detailed single-cell transcriptomic analysis of the gastric corpus and antrum at homeostasis. They highlight a Quartz-Seq2 technique as well as a FateID to determine cell lineage transcriptional changes. Through this analysis, the authors were able to determine preferentially expressed pathways during gastric cell lineage specification. In detail, they showed that the EGFR-ERK pathway was important in driving pit cell differentiation, and the NF-kappaB pathway preferentially increased undifferentiated cells through pit cell de-differentiation. The novelty of this paper lies in the new single-cell RNA technique and determination of cell lineage transcriptional changes with validation in an in vitro organoid system. Stronger in vivo data to support their in vitro findings would greatly strengthen the impact of this study.

Major Comments:

1. The authors state in the introduction that "These progenitor cells then differentiate into multiple epithelial cell lineages, including pit cells, neck cells, parietal cells, chief cells, and endocrine cells, which migrate bidirectionally from the isthmus to the luminal surface and to the gland base 8-10. Neck cells in the corpus function not only as mucus-secreting cells, but also as progenitors for mature chief cells 11,12.". While this has been the prevalent accepted theory, more recent work including PMID 31598973 and 31422913 using single cell transcriptomic and a variety of in vivo lineage tracing techniques have shown that the basal chief cells may represent an additional "independent zone" that is not dependent on the canonical isthmal stem cell for maintenance. These data do not immediately effect the conclusions of the current work, as much of this paper explores the relationship between the isthmal stem cell and the pit lineage. However, it does bring up how the FateID differentiation trajectory analyses that the authors performed on the differentiation trajectory from isthmus cells to neck and to chief cells should be interpreted.

In addition, the bulk of this paper focuses on pit cell fate decisions and the pathways discovered during this process bioinformatically. The authors may want to consider

refocusing the manuscript to highlight these data in more detail, and moving the parietal and chief cell lineage analyses (Fig 3,4) to the supplemental data. Especially, as the authors do not show the ability to manipulate these parietal and chief cell lineage pathways (insulin signaling and NOTCH signaling) in their organoid system (“no efficient protocols for the induction of fully differentiated chief cells and parietal cells in gastric organoids”).

We agree with these points. We moved the results of neck, chief, and parietal lineage analyses to supplementary figures (Supplementary Fig. 8–11). Following the reviewer’s suggestion, we have added text to describe the role of chief cells as the stem cells (Introduction, P3, L18) and excluded the chief cells from the endpoint of differentiation trajectories in this analysis (Fig. 2a and Supplementary Fig. 7a).

2. The authors describe their QC for the RNA-seq experiment using 2,500 gene counts and >40% of mitochondrial UMI counts. In other studies (PMID 31422913), cells with greater than 10-20% of mitochondrial UMI counts are excluded. Can the authors explain their more liberal inclusion criteria?

We thank the reviewer for pointing out our filtering criteria. Parietal cells require large number of mitochondria than the other gastric epithelial cells in order to meet the high energy demand of the proton pump (PMID: 20128020, Karam 2010 WJG). For clarity, we have added violin plots showing the mitochondrial UMI counts across all cell types (Supplementary Fig. 2b, d right). As shown here, parietal cells (cluster 0 in dataset 1, cluster 8 and 1 in dataset 2) have markedly higher mitochondrial gene ratio, whereas other gastric cell types have less than 20% mitochondrial gene ratio. Importantly, parietal cells exhibit the same or higher gene count as compared to other epithelial cell types (Supplementary Fig. 2b, d left), suggesting that the high mitochondrial ratio in parietal cells is not due to the low quality of the cells. In the revised version, we excluded the cells < 2,500 gene counts as low quality cells and did not use mitochondrial UMI counts for quality control. We have added information in the Methods section to describe the quality control in more detail (P23, L16). The extremely high mitochondrial UMI counts are not only observed in parietal cells but also in brown adipocytes, which have abundant mitochondria (PMID: 31573981). In the case of single cell sequencing of brown adipocytes, the cells with > 50% mitochondrial ratio were excluded as low-quality cells, suggesting that our criteria are reasonable for mitochondria-rich cells in the stomach.

3. The authors use their single-cell data set to define and identify mature pit cell markers such as AQP3. In Figure 1D, there is double immunofluorescence showing costaining of AQP3 and chief

cell marker, PGC. The authors should consider repeating this experiment with AQP3 and a pan-pit cell marker such as GKN2 or MUC5AC. This would more definitively demonstrate that AQP3 expression is limited to a specific more “mature” pit cell population.

KRT7 and KRT20 experiments should likewise have co-staining with a pan-pit marker to again show that there are definitive GKN2/MUC5AC positive pit cells in the corpus region that are KRT7/20 negative.

Finally, the authors should consider performing similar in vivo protein expression correlation experiments with their cluster 8/Corpus-specific pit cell markers SCTR and BASP1.

We agree with these points. Because the GKN2 and AQP3 antibodies were raised in the same species (rabbit), we did not perform double immunostaining. Furthermore, we also tested several Muc5ac antibodies. However, they were not all successful for immunofluorescence staining of mouse stomach tissues. Therefore, we performed double immunostaining with MKI67 and GKN2 or AQP3. In the new Fig. 1e, GKN2 expression was detected in broader area including MKI67⁺ pre-pit regions, whereas AQP3 expression was detected at the top of the glands distant from MKI67⁺ cells. The width of AQP3⁺ mature pit cells and GKN2⁺ pit cells revealed that AQP3⁺ area is significantly shorter than GKN2⁺ area (Fig. 1f). These data strongly support that AQP3 is the specific marker for mature pit cells.

KRT7, KRT20, and GKN2 antibodies were again raised in the same species (rabbit). Therefore, we performed double staining with MKI67 and KRT7 or KRT20 antibodies. As shown in Supplementary Fig. 3d, KRT20 was expressed at the surface of the antrum gland that was distant from MKI67⁺ cells, whereas KRT7 was expressed in a broader area, including MKI67⁺ pre-pit regions. These data support that KRT7 is expressed in the antrum pre-pit and mature pit cells, whereas KRT20 is expressed in the antrum mature pit cells.

We attempted immunofluorescence staining using a BASP1 antibody (NBPI-68958); however, this antibody was unsuccessful on tissue sections. Therefore, we compared BASP1 expression between corpus and antrum glands using western blotting (Supplementary Fig. 3c). BASP1 expression was significantly higher in corpus units than in antrum units, supporting the notion that BASP1 is preferentially expressed in corpus pit cells.

4. In the fourth paragraph of the results section, the authors discuss some unexpected results from their single-cell dataset including the finding that PGC and TFF2 transcriptional expression is more broad than canonically thought. Many of these findings have been shown before especially for the TFF2 expression (PMID 20708616). Consider moving this paragraph to the

discussion section.

According to the reviewer's suggestion, we have removed the sentences about *Tff2* expression and described the consistency of *Pgc* expression pattern with the previous reports by adding the citations as shown below (P6, L19).

“One of the chief cell markers *Pgc* was expressed in most gastric epithelial cells at mRNA level, as previously reported^{12,26},”

It is also unclear what the authors mean by this sentence “These results suggest that *Pgc* expression is an indicator of the chief cell lineage rather than a cell type-specific marker”. They show decreasing *Pgc* expression in the pit cells which are not considered part of the chief cell lineage. In addition, comment #1 sheds doubt on a direct isthmal stem cell to chief cell connection.

We agree that this sentence misleads the readers into thinking that pit cells are part of chief cell lineage. Therefore, we have removed this sentence.

5. The authors state that the ACTA2⁺ cells are primarily located at the bottom of the gastric glands, however the staining for the ACTA2, at least for the corpus, shows that it is expressed at the isthmal/pit region. Please clarify this discrepancy.

We agree that the description about the ACTA2⁺ and PDGFR α ⁺ stromal cells was inaccurate. Therefore, we measured spatial distribution of the two stromal cells and added the quantification data (Supplementary Fig. 12d, e). In corpus, PDGFR α ⁺ cells are distributed with two peaks in the upper and lower sections of glands, whereas ACTA2⁺ cells are mainly located in the middle of the glands. In antrum, PDGFR α ⁺ cells and ACTA2⁺ cells are located at the upper and lower sections of the glands, respectively. These quantification data clearly showed the distinct distributions of the two stromal cells. We removed the sentence about the antrum tissues to clarify that the aim of this paper is the analysis of corpus, and described the spatial localization of PDGFR α ⁺ cells ACTA2⁺ cells in corpus as follows (P10, L2):

“Notably, ACTA2⁺ stromal cells were primarily located in the middle section, whereas PDGFR α ⁺ stromal cells were primarily located in the upper and lower sections of the corpus gastric glands (Supplementary Fig. 12e).”

6. For Figure 2D and subsequent similar figures, can the authors explain the “Expression pattern”

images? Specifically, what do the axis and differing colors denote?

We appreciate your pointing out the lack of explanation about “Expression pattern.” We have added the following sentences in Figure Legends of Fig. 2d and subsequent similar figures (P31, L5):

“Average pseudotemporal expression profile of the representative node is indicated by a black line. The x-axis indicates the pseudotime from isthmus progenitor cells to pit cells and the y-axis indicates the expression level. Colors in each dot represent cell types, which corresponds to colors used in Fig. 2a.”

7. In Figure 3, the authors assess pathways enriched in the chief cell lineage. Please see point #1. In addition, the authors state that their “observations suggest that EGFR signaling is activated during pit cell differentiation but is suppressed during neck/chief cell differentiation”. It is unclear how this conclusion is reached as according to 3D and 3E, the EGFR1 pathway is enriched in Tree I/II which represents the isthmal precursor cells that are probably akin to Tree III in Figure 2E. The EGFR pathway is not enriched in Figure 3E Tree III/IV which more likely represent the proposed neck/chief cell differentiation pathway.

We intended to emphasize that EGFR1 pathway is enriched in the pit cell-related gene group (Fig. 2e) but not in the neck cell-related gene group (Supplementary Fig. 8d) nor the parietal cell-related gene group (Supplementary Fig. 10e) to claim that EGFR1 signaling is specifically related to pit cell differentiation. However, we agree with the reviewer that this conclusion is overstated. Therefore, we have removed this sentence.

8. The authors focus on EGFR and NF-kappaB signaling in their organoid work. In general, the NF-kB portion of the manuscript is at once novel/intriguing and very immature. It is mostly conjecture with TNFa and a single drug, whereas this is obviously a complex signaling pathway with TNFa only one input (and with TNFa affecting other pathways as well). They should consider including ligand/receptor analysis for the NF-kappaB pathway as well in Figure 5. It would strengthen their conclusions to have NF-kappaB pathway (ligand/receptors) be increased in preferentially in cluster 1 compared to the 2/8 mature pit clusters. They should also use other agents: activators and inhibitors.

We thank the reviewer for encouraging us to add additional data supporting the concept of the NF-κB pathway. According to the reviewer’s comments, we have performed the following experiments.

Using CellChat, a tool to infer cell-cell communication networks, we analyzed TNF signaling network across all cell types. CellChat predicted that TNF signaling activation in isthmus progenitor cells follows a paracrine manner (Supplementary Fig. 13b). Among the known TNF ligand-receptor pairs, *Tnfsf12-Tnfrsf12a* and *Tnf-Tnfrsf1a* were predicted as the dominant contributors in gastric cells (Fig. 3b). *Tnfsf12* is expressed in stromal cells and endothelial cells, and its cognate receptor *Tnfrsf12a* is preferentially expressed in isthmus progenitor cells (Fig. 3c). *Tnf* is expressed in EC cells and leucocytes, and its cognate receptor *Tnfrsf1a* is ubiquitously expressed in gastric cells (Fig. 3c). Therefore, we hypothesized that *Tnfsf12-Tnfrsf12a* is one of the upstream pathways of NF- κ B signaling in the isthmus progenitor cells. To test this hypothesis, we analyzed the effects of TNFSF12 using the corpus organoids. TNFSF12 decreased the expression of pit cell markers and increased the expression of *Mki67* and *Sox9* (Fig. 5b). Immunofluorescence analysis confirmed that TNFSF12 increased the number of MKI67⁺/SOX9⁺ cells (Fig. 5c, d). These data suggest that TNFSF12 maintains the undifferentiated state of isthmus progenitor cells and this is similar to TNF α (Fig. 5f).

To further confirm the effects of NF- κ B inhibitors on pit cell differentiation, we tested other NF- κ B inhibitors, sodium salicylate (NaSal) and MG132. Similar to QNZ, both NaSal and MG132 increased the expression of pit cell markers and decreased the expression of cell proliferation marker *Mki67*, albeit at different levels (Fig. 6d and Supplementary Fig. 20d). Immunofluorescence analysis confirmed that NaSal and QNZ both decreased the number of MKI67⁺ cells and NaSal significantly increased GKN2 expression at the protein level (Fig. 6e, f). These data further support our finding that NF- κ B signaling maintains the gastric progenitor cells in an undifferentiated state.

9. The authors functionally investigate the consequences on lineage differentiation using an in vitro organoid culture technique. They mention that they “used a well-established gastric organoid system, with certain modifications”. Based on their Methods section, they have omitted Noggin and EGF in the culture conditions. These were included in the culture conditions for the previously well-established protocols cited in references 34 and 35. Can the authors explain the decision to omit these factors and the consequences or anticipated consequences for doing so?

We thank the reviewer for pointing out the differences between the organoid culture conditions used in this paper and in the previous paper. Two changes were made in our organoid culture medium compared to reference 38 (PMID: 25105065). First, we used LDN-193189, another inhibitor of BMP signaling, instead of NOGGIN. Second, we removed EGF because our single cell RNA-seq analyses

suggested that EGFR signaling is not needed for progenitor cell proliferation but linked to pit cell differentiation. Therefore, we added a description on this in the Methods section (P27, L1).

“Two modifications were made in the organoid culture medium from the previous report³⁹; a BMP receptor inhibitor, LDN-193189, was used instead of NOGGIN, and EGF was removed to analyze the effect of EGFR signaling on isthmus progenitor cell differentiation.”

10. Related to point #9, Bartfield et al. showed as the authors indicated that at least with human stomach organoids that they were able to culture pit, neck, chief, and maybe parietal cells in reference 35. The authors show IF staining of KI-67 and GKN2. Are any other lineage markers present in these gastric organoids? If they are not, is this a consequence of their culture condition modifications discussed above?

If only undifferentiated dividing cells and pit-like cells are the only in vitro populations in their organoids, then point #1 (focusing more on the pit cell lineage and pathways, rather than additional lineages that can not be assessed using their in vitro system) is further supported.

We thank the reviewer for raising the concern that the gastric organoids used in this paper have different characteristics compared to the human gastric organoids in the previous paper (Bartfield et al., PMID: 25307852). As human and murine organoids may have different properties, we analyzed the difference between our organoids and the previously published murine organoids (Barker et al., PMID: 20085740). Barker et al. showed that *Muc6*, *Pgc*, and *Lgr5* were expressed in their organoids and pit cells were formed only when Wnt3a concentration was decreased. In our gastric organoids, MKI67⁺ cells and GKN2⁺ cells were the major populations, and there were no GSII⁺ and PGC⁺ cells (Supplementary Fig. 17a, b). The major difference between the culture methods is that they used Wnt3a-conditioned medium (Wnt3a-CM) and we used recombinant Wnt3a protein (rWnt3a) (Supplementary Fig. 18g). Therefore, we examined the effects of Wnt3a-CM and rWnt3a on the expression of gastric epithelial cell markers. The expression level of pit cell markers was quite low in Wnt3a-CM compared to that in rWnt3a (Supplementary Fig. 18h). Importantly, EGF induced pit cell marker expression in rWnt3a-containing medium, but not in WNT3a-CM. In addition, *Muc6* and *Lgr5* expression levels were more than 100-fold higher in cultures with WNT3a-CM than in cultures with rWnt3a (Supplementary Fig. 18h). Collectively, WNT3a-CM induced the higher expression of neck and chief cell markers and strongly suppressed pit cell differentiation compared to rWNT3a, leading to the different response to EGFR signaling activation.

11. For their in vitro work with inhibitors and activators of the various pathways shown in Figure 7 and 8, it would be worthwhile to include data (eg possibly western blots for activated downstream targets of EGFR pathway) to show that the treatments are affecting the pathways as the authors

intended.

We agree with these points. Therefore, we analyzed the effect of EGFR signaling inhibitors on ERK phosphorylation by immunofluorescence staining. As expected, the EGFR inhibitor erlotinib completely suppressed TGF α -induced ERK phosphorylation in gastric organoids (Supplementary Fig. 18b, c). The MEK inhibitor PD0325901 also suppressed pit cell differentiation (Fig. 7h) and inhibited ERK phosphorylation in gastric organoids (Fig. 7i, j). Regarding NF- κ B signaling, immunofluorescence staining confirmed that TNFSF12 and TNF α significantly increased the level of NF- κ B1 compared to control organoids (Supplementary Fig. 19b, c, e, f).

12. The authors show that manipulation of the EGFR pathway and NF-kappaB pathways increases the transcriptional profile of the neck-chief cell lineage (Figure 6). Is this reflected at the protein level? For example, are the authors able to detect this increase in these markers (SOX9 and PGC) with EGFR inhibition?

We thank the reviewer for these comments. To answer this question, we have analyzed the protein expression of SOX9 in the gastric organoids. Consistent with qPCR results, erlotinib (Supplementary Fig. 18d, e) and TNFSF12 (Fig. 5c, d) increased the number of SOX9⁺ cells. The NF- κ B inhibitors QNZ and NaSal decreased the number of SOX9⁺ cells (Fig. 6e). Although *Pgc* mRNA expression was induced in gastric organoids treated with the EGFR inhibitor erlotinib (Fig. 4g), and NF- κ B inhibitors QNZ and NaSal (Fig. 6b, d), PGC protein was not detected in any of these organoids (Supplementary Fig. 18d, 20e).

13. The authors describe effects of apical-basal polarity through in Figure 6J through activation of the EGFR pathway using TGF α . The authors should consider providing higher resolution confocal imaging here to highlight this.

According to reviewer's suggestion, we have added immunostaining data taken by confocal microscopy (Supplementary Fig. 18f).

14. These organoid data do show certain transcriptional and protein level changes with pathway manipulation. To fully make the case that these pathways are important for pit cell lineage specification, the authors should consider in vivo experiments such as treating WT mice pharmacologically with the same treatments as the organoids and showing similar effects. These data would be especially important in light of the recent study PMID 33951440. The data in

Figure 7G are an excellent start to confirming their organoid work *in vivo*, but further experiments should be considered.

We thank the reviewer for encouraging us to perform *in vivo* experiments. To confirm EGFR signaling-dependent pit cell differentiation *in vivo*, we administered erlotinib to wild-type mice for two weeks and analyzed their stomach tissues. We found that erlotinib treatment significantly decreased the length of GKN2- and AQP3-expressing pit cell region, leading to the luminal shift of parietal cells (Fig. 7l, m). Conversely, the number of MKI67⁺ cells was high in the mice treated with erlotinib. These *in vivo* data further support our *in silico* and gastric organoid data that EGFR signaling promotes pit cell differentiation.

15. For the experiments in Figure 7, please include untreated organoid controls for reference. This is especially important to understand the magnitude of changes as the authors are using relative change ratios here.

We thank the reviewer for pointing out the missing control. We have included the data for the control organoids in all qPCR analyses (Fig. 7d, f, h; Supplementary Fig. 21a, b, d).

16. In addition to the citations highlighted above that should be discussed, there are several previous studies that could be integrated with the findings here (note: the authors have done a reasonable job of citation here, but when so many pathways and cells are touched on, it is difficult).

We thank the reviewer for these very helpful comments. We added proper citations and expanded our discussion as follows.

- The papers the authors interpret ref 17 as implying “Activation of NOTCH signaling in parietal cells induces the dedifferentiation of parietal cells into undifferentiated progenitor cells and ultimately leads to adenomas” and “cell-specific activation of NOTCH signaling dedifferentiates parietal cells into progenitor cells with the ability to differentiate...” which is a bit misleading. In ref. 17, the authors activated constitutive Notch using a constitutive promoter in the parietal cell lineage, but that promoter can be active in progenitors, so another (perhaps more likely) interpretation of those findings is that activation of Notch during parietal cell differentiation can lock cells in a progenitor state rather than dedifferentiate mature cells.16.

“Activation of NOTCH signaling in parietal cells and their progenitor cells maintained the progenitor cells in an undifferentiated state” (Introduction, P4, L5)

“Downregulation of Delta-Notch signaling in parietal cell differentiation is consistent with the previous study showing the role of NOTCH signaling in antagonizing parietal cell differentiation¹⁷.” (Results, P8, L15)

-NF-κB signaling in gastric epithelium has previously been addressed in vivo: PMID 19962981

We added the reference and inserted the following sentence.

“The previous study reported that IKKβ/NF-κB signaling plays an inhibitory role in Helicobacter infection-induced apoptosis and necrosis⁴².” (Results, P13, L22)

-“ Its ligand, Igf1, and an upstream regulator of Igf1, Ghr, were also highly expressed in neck and chief cells. These expression patterns were consistent with the results of the enrichment analysis, which showed that genes related to neck/chief and parietal cell differentiation but not pit cell differentiation are enriched in the insulin signaling pathway” There was an early -omics study also indicating a progenitor and parietal cell feedback loop of IGF signaling with IGF and IGFr in progenitor autocrine signaling and IGFBP2 from parietal cells suppressing the IGF pathway: PMID 12409607

We have added the citation and expanded our discussion as follows:

“A previous study suggested autocrine and negative paracrine regulation of IGF signaling based on the expression of *Igf1* and *Igf1r* in gastric progenitors and of its negative regulator *Igfbp2* in parietal cells³⁷. (Discussion, P19, L14)

-IQGAP3 has emerged as a key isthmal gene that can regulate RAS (PMID 33293280), which of course is critical in ERK signaling (and downstream of the EGFR signaling highlighted here). Was IQGAP3 found? There have been several other papers on K-Ras and gastric epithelial homeostasis (PMID 26677984), as well as earlier literature showing effects of inhibiting pERK on homeostasis (PMID 23589310), which should probably be referenced.

We have added the citations and discussed consistency and inconsistency between our study and the previous studies as follows:

“In addition, ERK signaling has previously been shown to be involved in atrophy-induced progenitor cell expansion⁴².” (Discussion, P17, L6)

“A recent study reported the *Iqgap3*-Ras-Erk axis-dependent proliferation of isthmus progenitor cells²⁴. Our single-cell RNA-seq data confirmed the preferential expression of

Iqgap3 in progenitor cells (Supplementary Fig. 4b). In our study, however, ERK regulates pit cell differentiation rather than progenitor cell proliferation. It would be interesting to examine how IQGAP3 knockout affects pERK level and pit cell differentiation under *in vivo* normal homeostasis.” (Discussion, P19, L18).

The authors should also relate their work in the context of other single-cell transcriptomic analyses of the stomach (PMID 31481545 and 31422913). To highlight the differences in their single cell technique compared to the Chromium based techniques, and also discuss the strength of their FateID analyses

According to the reviewer’s suggestion, we emphasized the advantage of Quartz-Seq2 method as follows:

“Previous studies have reported single-cell transcriptomic analyses of gastric cells^{12,36,48,49}. In this study, we leveraged the power of Quartz-Seq2, which has an extremely high gene detection rate with no prominent batch effect (Supplementary Fig. 2). The average gene count in our analysis was 6,013 (dataset 1) and 6,270 (dataset 2), which is three times higher than the previous 10X Genomics-based study, with an average of 1923 genes⁴⁸. In addition, our reanalysis showed that the average number of detected genes in the recent study⁴⁹ was 3,474 (Supplementary Fig. 2e), which indicates the high number of detected genes in our datasets. This allowed us to perform enrichment analysis using a considerable number of marker genes. We were also able to capture accurate pseudotime estimation using the FateID algorithm and identify the regulatory pathways of gastric epithelial cell differentiation processes in unprecedented detail.” (Discussion, P18, L14)

Finally, there was a recent study highlighting pit-parietal cell differentiation decisions involving AMPK and KLF4, which would be informative to integrate: PMID 32243780.

We have added the citation and inserted the following sentence.

“pathways related to mitochondrial electron transport and oxidative phosphorylation were identified (Supplementary Fig. 10e), which is consistent with the previously reported increased mitochondrial biogenesis during parietal cell differentiation³³.” (Results, P8, L19)

Minor Comments:

1. Introduction, third paragraph, second sentence, please add “cells” after parietal, and consider changing “numbers” to “number of transitional cells”.

We have corrected the typos as the reviewer suggested. (P4, L1-2)

2. Introduction, third paragraph, fifth sentence, please consider changing TGFa transgenic mice to “TGFa overexpressing transgenic mice”.

We have revised this phrase as the reviewer suggested. (P4, L7)

3. Introduction, fourth paragraph, first sentence, please add “protocols” after single-cell RNA sequencing.

We have inserted the word “protocols” as the reviewer suggested. (P4, L15)

4. For Figure 1, can the authors increase the resolution of the inset pictures for 1E. In addition, for 1D, there appears to be decreased resolution for KRT7/PGC and KRT20/PGC corpus pictures.

We have replaced the inset images of corpus stained with ACTA2/PDGFR α and ACTA2/PECAM1 with high resolution images (new Supplementary Fig. 12d). Also, we have replaced KRT7/PGC and KRT20/PGC images with KRT7/MKI67 and KRT20/MKI67 images (Supplementary Fig. 3d) according to the reviewer’s major comment #3.

5. Page 15, last line “HE” should be “H&E”.

We have corrected this as the reviewer suggested. (P12, L23)

6. For Figure 7G, it is difficult to appreciate for the Corpus picture that the pERK positive cells are preferentially the pre-pit cells. Decreasing the gain or using confocal microscopy may provide more conclusive data.

We have replaced the original figures with the high-resolution images (new Fig. 7k).

7. The authors discuss the role of EGFR signaling in cancer as an analogous pathway during normal stomach homeostasis. This is unlikely to be the case and the link between the role of EGFR in cancer and at homeostasis should be de-emphasized. RTK-Ras pathway activation per TCGA (PMID 25079317) work is usually a late consequence in the CIN subtype of cancers as oftentimes result of, as the authors correctly state, “copy number amplification”. This would not be a transcription factor mediated differentiation process as in normal stomach homeostasis. In addition, GC is preceded by intestinal metaplasia and SPEM through the Correa sequence. And eventual activation of the EGFR pathway would be effecting a different population of cells than in the normal stomach.

Following the reviewer’s advice, we removed the following sentences from the Discussion section to de-emphasize the link between the role of EGFR in cancer and at homeostasis:

~~“Indeed, an antibody against HER2, trastuzumab, has demonstrated remarkable therapeutic effects by inhibiting HER2-dependent cell proliferation signals and antibody-dependent cell-mediated cytotoxicity” (Discussion, P17, L4)~~

8. Page 22, last sentence of the first paragraph, “out” should be replaced with “our”.

We removed this paragraph and replaced with a new paragraph discussing about CellChat-based analysis (Discussion P19, L1).

Reviewer #2 (Remarks to the Author):

05_04_2021

In this manuscript titled “EGFR signaling is a potent inducer of differentiation but not proliferation of gastric progenitor cells in stomach epithelial homeostasis”, the authors used the single-cell RNA sequencing method to obtain gene expression profiles for 15 cell types in normal mouse gastric cell. The authors show that two types of pit cells and stromal cells which characterize the gastric unit in health mouse stomach cells. Moreover, the author also demonstrated the regulatory pathway involved in differentiation from isthmus progenitor cells to pit cell and neck/chief cells. EGFR signaling promotes pit cells differentiation and suppresses neck and chief cells fate decisions. The findings of this work have translational potential to gastric disease patients, as the defects in gastric progenitor cell differentiation are associated with gastric disorder and even gastric cancer. The study is interesting; however, the findings rely on a single health mouse model, and the cell differentiation measurement could include additional measures.

Major comments:

- This study is overall descriptive, and the conclusion was drawn mainly based on single cell sequencing data analysis. Now days, single cell RNA sequencing is quite high throughput, a few thousand cells can be easily sequenced at a relative low cost for each sample, and fresh samples are easily collected from mouse experiments. However, only a total of 493 cells were analyzed in the entire study and these 493 cells were separated into 17 clusters to infer epithelial and stromal cell lineages. Given that the epithelial cell lineages are highly diverse, a clear limit of this study is that only a very small number of cells were analyzed for each cell lineages and therefore the results may not be reproducible, and the authors should analyze more cells from different mouse models.

Following the reviewer's suggestion, we simplified the description on the single-cell RNA-seq data related to Fig. 1, 2, and 3. Second, to increase the number of cells analyzed, we performed additional single-cell RNA-seq and further analyzed 3,993 cells of corpus gastric units from wild-type mice. Pseudotime analysis of the first and second datasets showed EGFR signaling and NF- κ B signaling as the enriched pathways in pit cell- and isthmus cell- gene groups, respectively (Fig. 2e, f, Supplementary Fig. 7e, f). Similarly, lysosome-related terms were enriched in neck cell-related gene group (Supplementary Fig. 8d, Supplementary Fig. 9d). These data confirmed the high reproducibility of our lineage analysis.

- Another major limit of this study is the experimental design. The authors tried to profile the expression dynamics of progenitor cell differentiation toward mature epithelial cell. However, longitudinal sampling was not performed, and the authors only analyzed samples collected at a single timepoint, which represents only a snapshot of this dynamic process. The authors tried to infer expression dynamics using pseudotemporal analysis, however the pseudotemporal analysis is mainly knowledge based and the results can change totally with different parameter setting. It is hard to tell whether the trajectory results are reproducible.

The reviewer asked us to conduct longitudinal sampling instead of pseudotime analysis. Our aim is to understand the mechanism of stem cell differentiation in healthy homeostasis without any damage. Under these conditions, gastric epithelial cells are continuously turned over, and cellular composition of gastric glands is nearly identical at any time point in the adult stomach. Even if we perform longitudinal sampling under healthy homeostasis, we will obtain similar single-cell RNA-seq data from different mice. In fact, the biological replicate data from different mice overlap nicely (Supplementary Fig. 2e). Therefore, tracing differentiation process by longitudinal sampling would require the pseudotime analysis.

To validate the reliability of pseudotime analysis, we used the two single-cell RNA-seq datasets (dataset 1 and 2) with different batches of samples and demonstrated high reproducibility for the enrichment analysis of pseudotime-dependent genes (Fig. 2 and Supplementary Figs. 7, 8, and 9). We also took advantage of the gastric organoid assay for isthmus progenitor to pit cell differentiation. We confirmed that EGFR signaling and NF- κ B signaling promoted and suppressed pit cell differentiation in gastric organoids, respectively. Finally, we validated the role of EGFR signaling in pit cell differentiation by *in vivo* analysis. As for neck and parietal cell differentiation, we have not validated these results using the gastric organoid assay. Therefore, we moved the results of neck and parietal cell lineages to the supplementary data (Supplementary Figs. 8, 9, 10, and 11).

We have also added the following text to the Discussion section to point out the limitation of pseudotime analysis.

“Certain limitations were noted in this study. Although our *in silico* analysis inferred various signaling networks, we did not validate the signaling pathways involved in neck and parietal cell differentiation.” (P20, L5)

- a key part of this study is single cell pathway enrichment analysis to infer the molecular drivers of lineage differentiation and cell fate regulation. A pathway usually contains up to a hundred or few hundred genes, given a high dropout rate of single cell sequencing, only a small number of genes can be detected, make the pathway analysis very challenging especially when a small number of cells are analyzed, which may produce uncertain results with a lot of noise. However, no validation experiments were performed in larger cohort.

We thank the reviewer for these comments. In the previously published Chromium-based analysis (PMID: 31481545), although 29,351 cells were analyzed from healthy, damaged, and inflamed stomachs, the average gene counts per cell was only 1,923. In our single-cell RNA-seq experiments, the cells with an average 6,013 and 6,270 gene counts per cell were analyzed with dataset 1 and 2, respectively. The cells with gene counts < 2,500 were removed as low-quality cells. In dataset 2, we analyzed ten times more cells (3,993 cells) than in dataset 1 and demonstrated high reproducibility of our enrichment analysis (please refer to the response to major comment #2). Moreover, using the gastric organoid assay, we have confirmed that EGFR signaling and NF-κB signaling regulate pit cell differentiation and progenitor cell maintenance, respectively, which is consistent with the enrichment analysis results. These data clearly demonstrated the high reliability of our single-cell RNA-seq data and lineage analysis.

- The authors performed organoid experiments to validate the molecular mechanism for gastric epithelial cell differentiation. The organoid experiments are not ideal for such purpose because of the striking differences in their 3D structures, tissue layout, and morphology. The authors should not infer the lineage differentiation solely based on marker genes the cells expressed without taking into consideration the tissue structure and cell morphology and cell function. The function of the cells are not evaluated.

We thank the reviewer for these comments. We agree that the gastric organoids are different from gastric glands in their structures. To validate the role of EGFR signaling in pit cell differentiation *in vivo*, we administered the EGFR inhibitor erlotinib to wild-type mice. We found that erlotinib treatment significantly decreased the number of pit cells and

increased the number of MKI67⁺ isthmus progenitor cells (Fig. 7l, m), which is consistent with the results of our lineage analysis and gastric organoid assay that showed EGFR signaling promotes pit cell differentiation.

- As for data processing, the filter used in this study is not proper, the authors retained cells with up to 40% of mitochondrial UMI counts for subsequent analysis. the cutoff 40% seems a bit high, as most of other studies used 15% or at most 20%. Therefore, the authors probably included some dying cells or cells in poor quality for analysis, and these cells may impact clustering and cell type identification.

Parietal cells have a significantly larger number of mitochondria than the other gastric epithelial cells to meet the high energy demands of the proton pump (PMID: 20128020, Karam 2010 WJG). We have added a figure showing the mitochondrial UMI counts across all cell types after the removal of cells < 2,500 gene counts (Supplementary Fig.2b and 2d). In our single-cell RNA-seq data, including the newly added data, the mitochondrial ratio was 7.5-8.5% on average after excluding low-quality cells. As shown in dataset 1, parietal cells have a higher mitochondrial proportion (average of 18% mitochondrial gene ratio) and other gastric cell types have an average of 7% mitochondrial gene ratio. Similarly, in dataset 2, parietal cells and other gastric cell types had an average of 21% and 5% mitochondrial gene ratio, respectively. Due to this unique property of parietal cells, we included the cells with > 20% mitochondrial gene ratio, otherwise the majority of the parietal cells would be excluded from the analysis. We added text to describe the detailed quality control process in Methods (P23, L16). In the previous study that performed single cell sequencing of brown adipocytes, which have abundant mitochondria (PMID: 31573981), the cells with < 50% mitochondrial ratio were used for analysis, suggesting that our criteria is reasonable for dealing with mitochondria-rich cells in the stomach.

- No method describe how the cell doublets were handled and it is unclear whether the doublets (should be removed) were included in their analysis.

Thank you for pointing this out. Most single-cell RNA-seq in previously published studies are performed by stochastic sampling using microfluidics which cannot remove doublets from a cell population. In microfluidics, even if sorted singlet cells are used, aggregated singlets or two singlets may enter one droplet, resulting in a pseudo-doublet. Our single-cell RNA-seq method is a plate-based method, which allows singlet fractionated by cell sorter to be directly collected on 384-well plates. This minimizes the risk of doublet contamination. In both datasets, doublet removal by cell-sorter was performed. We have added new Supplementary Fig. 1 showing the process of single cell sorting. After the removal of dead

cells and cell debris, the cells in the singlet gate were sorted to 384 plates (Supplementary Fig. 1c, d). We confirmed that the singlet gate (R5) contains neither debris nor cell aggregates (Supplementary Fig. 1e, middle panel) and that the aggregate gate (R8) contains cell doublets and triplets (Supplementary Fig. 1e, right panel, black arrowheads).

Minor comments:

- Figure 1D: The scale bar of the immunofluorescence stain pit cell makers is not consistent in Corpus and Antrum tissues. Please clarify.

We thank the reviewer for this comment. Since antrum gastric glands are much shorter than corpus gastric glands, we took images of antrum tissues at 40x magnification and corpus tissues at 20x magnification in the previous version. Therefore, the length of scale bar was different in the corpus and antrum. However, in the revised version, we removed most of the sentences and figures about Antrum to simplify the description about the single cell RNA-seq data, according to your comment “This study is overall descriptive”, because the aim of this paper is the analysis of corpus tissues.

- Figure 6H: The scale bar of the HE staining is not accurate in low and high amplification. Please clarify.

As described in the Figure legend (this figure was moved to Fig. 4h in the revised manuscript), scale bar in low and high magnification picture is 200 μm and 50 μm , respectively. For clarity, we have added the length of the scale bar in those images.

- Figures 7: The author further explore the downstream gene of EGFR signaling which responsible for pit cell differentiation, based on pathway analysis, the ARP2/3 inhibitor CK666 were treated with gastric organoid. However, although the author compared the organoid and differentiation makers between nontreated vs TGF α in Figure6, the negative control or untreated group should also be included in Figure 7.

We thank the reviewer for pointing out the missing control. We have included the data for the control organoids in this analysis and subsequent similar analysis (Fig. 7b–d, f, h and Supplementary Fig. 21a, b, d).

- Again, obviously, the scale bar for the Fig.7B for the top and low panels is not accurate.

Thank you for pointing out that the scale bar in the high magnification image of TGF α treated organoid was not accurate. We have replaced this figure with the new one, and for the clarity, we have added the length of scale bar in the images.

Reviewer #3 (Remarks to the Author):

This study describes results from single-cell RNA sequencing method, Quartz-Seq2, to obtain gene expression profiles from different cell types in healthy mouse stomachs. Cells are clustered to identify different subsets and pseudotemporal analyses are done to predict development of different subsets, and differential gene expression and 3-D organoid studies are used to predict important pathways for gastric epithelial cell differentiation. The data are interpreted as evidence that EGFR signaling is involved in promoting the development from progenitor cells to mucous producing pit cells.

Weaknesses of the study include the limited number of cells (493 total) analyzed and pseudotemporal analyses are done on a limited subset of these cells, so these are even smaller datasets. The cells appear to be from a mixture of both the corpus and antrum, which is a bit confusing as the focus of the paper is on corpus cells. The results are descriptive in nature, differentiation pathways are “suggested” “putative” based on differential gene expression analyses of a limited dataset of epithelial cells. Organoids are used to investigate possible signals involved in differentiation, but these are problematic. For example, changes in organoids are assumed to be caused by altered differentiation of stem cells but expansion of existing pit-like cells cannot be excluded. The author’s state that to the best of their knowledge this is the first study to generate gene expression analyses of whole gastric units, but a review of the literature identified at least 3 published

studies in mouse and humans PMID: 31481545, PMID: 32835664, PMID: 32532891, PMID: 33691112. The several other reports of single cell data generated from the stomach limit the novelty of data presented. This paper does provide an additional dataset for the field, however the conclusions are speculative and may have limited impact on the community and wider field.

We thank the reviewer for these comments.

We agree that we analyzed the mixture of corpus and antrum cells (dataset 1) and performed pseudotemporal analysis using a small number of cells in the original manuscript, which raised concerns about the reliability of the analysis. To clarify this, we further analyzed 3993 cells only from corpus gastric units (dataset 2). In both dataset1 and 2, the biological replicates prepared from different mice by independent operators were sorted into different plates. These different plate-derived samples were uniformly clustered in UMAP (Supplementary Fig. 2e), confirming that there was no obvious batch effect and these analyses were highly reproducible. Moreover, in both datasets, EGFR signaling (Fig. 2e, Supplementary Fig. 7e), NF-kB signaling (Fig. 2f, Supplementary

Fig. 7f), and lysosome-related terms (Supplementary Fig. 8d, Supplementary Fig. 9d) were enriched in the gene groups related to pit cell differentiation, isthmus cell maintenance, and neck cell differentiation, respectively. These data confirmed the robustness of our lineage and enrichment analyses.

We agree that bioinformatics analysis suggests cell clustering, differentiation trajectories, and their regulatory mechanisms. To validate the results of cell clustering, we analyzed the expression of cell-type-specific markers via immunofluorescence staining (Fig. 1e, Supplementary Figs. 3d, 4a, and 12d). Importantly, we demonstrated that the *in vivo* expression patterns of newly identified markers such as AQP3, KRT7, and KRT20 were consistent with the cell clustering results. To validate the results of lineage analysis, we analyzed the function of EGFR signaling and NF- κ B signaling using gastric organoids. According to the reviewer's suggestion, we counted the number of MKI67⁺ and GKN2⁺/MKI67⁺ cells to exclude the possibility that EGFR signaling promotes the expansion of pre-pit cells. The number of MKI67⁺ proliferating cells was significantly decreased in TGF α -treated organoids (Fig. 4e). Also, the number of GKN2⁺/MKI67⁺ was not higher in TGF α -treated organoids in comparison to the control organoids (Supplementary Fig. 17h, i). These data clearly indicated that EGFR signaling does not expand pit cells but inhibits their proliferation. Furthermore, to validate the role of EGFR signaling, we performed *in vivo* experiments. We administered the EGFR inhibitor erlotinib to wild-type mice for two weeks and analyzed their stomach tissues. We found that erlotinib treatment decreased GKN2- and AQP3-expressing pit cells, leading to the luminal shift of parietal cells (Fig. 7l, m). Conversely, the number of MKI67⁺ cells was increased in the mice treated with erlotinib. These data indicate that EGFR signaling is required for pit cell differentiation *in vivo*, strongly supporting the results of our *in silico* and gastric organoid analyses.

We apologize that we have overstated the novelty of our work with the following sentence:

“to the best of our knowledge, this is the first study to generate a systematic atlas of gene expression profiles of whole gastric units.” We have removed this sentence. Our major finding is the identification of the regulatory mechanisms of gastric epithelial cell differentiation but not cell characterization. In this paper, we took advantage of a highly sensitive single-cell RNA-seq method, Quartz-Seq2 and analyzed the cells with an average 6,013 and 6,270 gene counts per cell in datasets 1 and 2, respectively. The method has more than three times higher gene detection rate than the ordinary 10X Genomics-based study, with an average of 1923 genes per cell (PMID:31481545 and PMID:32835664). Moreover, there was no significant batch effect on these datasets (Supplementary Fig. 2). This high gene detection rate of Quartz-Seq2 provided us with a detailed pathway analysis of gastric epithelial cell differentiation. To clarify the difference between our work

and the previous work, we cited the previous studies and added the following text to the Discussion section:

“Previous studies have reported single-cell transcriptomic analyses of gastric cells^{12,36,48,49}. In this study, we leveraged the power of Quartz-Seq2, which has an extremely high gene detection rate with no prominent batch effect (Supplementary Fig. 2). The average gene count in our analysis was 6,013 (dataset 1) and 6,270 (dataset 2), which is three times higher than the previous 10X Genomics-based study, with an average of 1923 genes⁴⁸. In addition, our reanalysis showed that the average number of detected genes in the recent study⁴⁹ was 3,474 (Supplementary Fig. 2e), which indicates the high number of detected genes in our datasets. This allowed us to perform enrichment analysis using a considerable number of marker genes. We were also able to capture accurate pseudotime estimation using the FateID algorithm and identify the regulatory pathways of gastric epithelial cell differentiation processes in unprecedented detail. (P18, L14)

Reviewer #4 (Remarks to the Author):

Takada et al. used a single cell RNA sequencing approach to analyze the gene expression profile in normal mouse gastric units. They found 15 different cell types revealing expression dynamics of progenitor cell differentiation toward the pit, neck, chief and the parietal cell lineages. Pathway analyses and subsequent organoid experiments revealed that the EGFR ERK signaling pathway promoted pit cell differentiation, while the NFkB pathway maintained gastric progenitor cells in an undifferentiated state. They conclude that the activation of the EGFR signaling pathway results in a differentiation of isthmus cells and does not have a mitogenic function.

The manuscript is well written and the results are very convincing, especially as the authors functionally validated their scRNA data in gastric organoids. Some issues remain that needs to be addressed:

Major:

The scRNA (as far as this reviewer understood) is based solely on a single scRNA run, and only 493 cells were further analyzed. The results in only a few cells per identified “cell type”. The data seems valid, as many known markers are found allowing a clear separation of known cell types. Nevertheless, the authors claim to have established a unique expression atlas of the stomach glands. They indeed also find novel interesting genes to be expressed in the known cell types.

Nevertheless, it is necessary to perform a second independent run of scRNA analysis from independent mice to really convincingly "... generate a systematic atlas of gene expression profiles..." as stated by the authors. The two runs should be compared to document consistency, and the data of both runs then analyzed together to increase the number of cells per identified cell type.

According to the reviewer's suggestion, we performed additional single-cell RNA-seq using corpus gastric units (dataset 2). In previous and newly obtained datasets, the biological replicate samples were isolated from three different mice by two different operators. All clusters consisted of cells from both biological replicate (Supplementary Fig.2). In dataset 2, we stained the cells with Hoechst 33342 to remove cell debris at the cell sorting step. Although Hoechst 33342 staining induced slight cell damage and death, the cell recovery rate after single-cell RNA-seq (about 78%) considerably improved compared to that of dataset 1 (about 13%) (Supplementary Fig. 2a, c). Therefore, for the analysis of corpus major epithelial cell types, dataset 2 was used as the main dataset (Fig. 1) and dataset 1 was used to validate the reproducibility (Supplementary Fig. 3). Importantly, newly identified pit cell markers such as *Aqp3*, *Basp1*, *Krt7*, and *Krt20* were consistently observed in dataset 1 (Supplementary Fig. 3b) and dataset 2 (Supplementary Fig. 4b), supporting the reliability of cell clustering results. In addition, enrichment analysis of both datasets identified EGFR signaling and NF- κ B signaling in pit cell- and isthmus cell-related genes, respectively, as the enriched pathways (Fig. 2e, f, and Supplementary Fig. 7e, f). These results demonstrated the robustness of our single-cell RNA-seq data and pseudotime analysis. With regard to the analysis of the minor gastric epithelial cell types, including stromal cells and endocrine cells, we used the combined datasets 1 and 2 because of their low cell numbers. The clustering of these data (Supplementary Fig. 12a, b) further confirmed the reproducibility of our single cell analysis.

Figure 5: Ligand and receptor expression patterns are coming from the same data set as the pathway enrichment analysis. Confirm findings of pathway enrichment in specific cell types by double immunohistochemistry.

We thank the reviewer for encouraging us to examine the expression of ligands and receptors in stomach tissues. According to the reviewer's suggestion, we performed immunofluorescence staining of corpus tissue using antibodies for EGFR and IGF1-R. EGFR was expressed in pit cells with the highest expression at the luminal surface (Fig. 3d), whereas IGF1R is preferentially expressed in parietal and chief cells (Supplementary Fig. 14a), consistent with the pathway enrichment analysis results. As for the TNF signaling receptor TNFRSF12a, the antibody against TNFRSF12a was unsuccessful with mouse stomach tissues.

Figure 6:

- The authors cite a study of HUMAN gastric organoids (Bartfeld et al) and state that these organoids express markers of pit, neck, and chief cells. Actually, the initial report of gastric organoids (Barker et al, Cell Stem Cell, 2010) reported that gastric organoids express markers of neck (Muc6) and chief cells (GIF and PepC). And these organoids did NOT express pit cell markers under normal growth conditions. Only in Wnt3a reduction medium pit cells were formed. As the authors use similar culturing conditions as published by Barker et al, this has to be resolved.

We thank the reviewer for raising the concern about the difference between our work and the previous work. The major difference between the medium components used in the previous study and our study is that they used Wnt3a-conditioned medium (Wnt3a-CM) and we used recombinant Wnt3a protein (rWnt3a) (Supplementary Fig. 18g). Barker et al. have shown that rWNT3a protein supports organoid growth as well as Wnt3a-CM at day 7, but they have not analyzed the characteristics of the organoids cultured with Wnt3a-CM and rWnt3a in detail. Thus, we examined how Wnt3a-CM affects EGFR signaling-mediated pit cell differentiation and the expression of neck and chief cell markers. The expression of pit cell markers was not induced by EGF in the presence of Wnt3a-CM (Supplementary Fig. 18h), consistent with the previous report (Baker et al.). Interestingly, *Muc6* and *Lgr5* expression levels were more than 50 times higher and pit cell marker expression levels were approximately 400 times lower with Wnt3a-CM than with rWNT3a. Collectively, Wnt3a-CM induced the higher expression levels of neck and chief cell markers and strongly suppressed the expression of pit cell markers compared to rWnt3A, which caused the different response to the activation of EGF signaling. The functional difference between Wnt3a-CM and rWnt3a has been elucidated in a recent study using colon organoids (PMID: 33643277); Wnt3a-CM, but not rWnt3a, was able to support the long-term survival of human or mouse colon organoids.

- Correct citation Bartfeld et al to Barker et al or other murine study referred to

We have cited the paper (Reference 39, Barker et al.) and described the difference between WNT3a-CM and rWNT3a as described above.

“A previous study reported that pit cell differentiation was induced by decreasing Wnt3a in culture³⁹.” (P13, L6)

- Are those organoids established from corpus glands or antrum glands? It is referred to later as corpus organoids, but please state in the beginning.

Thank you for your comment to make the text easier to read. We have added the following sentence:

“In all analyses, corpus organoids prepared from adult mouse stomach were used.” (P11, L21)

- The authors mention slight modifications to the original protocol. Please specify.

Two changes were made in our organoid culture condition compared to reference 38 (Mahe et al., PMID: 25105065). First, we used LDN-193189, an inhibitor for BMP type I receptors, instead of NOGGIN because LDN-193189 has BMP-antagonistic activity similar to that of NOGGIN and is cheaper than NOGGIN. Second, we did not use EGF, because our single cell RNA-seq analysis suggests that EGFR signaling is not needed for isthmus progenitor cell maintenance and is linked to pit cell differentiation. We have added a description of this in the Methods section.

“Two modifications were made in the organoid culture medium from the previous report³⁸; a BMP receptor inhibitor, LDN-193189, was used instead of NOGGIN, and EGF was removed to analyze the effect of EGFR signaling on isthmus progenitor cell differentiation.” (P27, L1)

- Show long term culturing of the organoids, e.g. passaging over >10 passages to confirm long term maintenance of organoids in used medium

Following the reviewer’s suggestion, we examined the long-term culture of gastric organoids with our medium. We found that the gastric organoids could be passaged until passage 4 (Supplementary Fig. 17c), but the number of the organoids decreased after serial passages. This data is consistent with the report showing that Wnt3a-CM, but not rWnt3a, maintains the long-term survival of mouse colonoid cultures (Wilson et al., 2021, PMID: 33643277).

- Figure 6A/S9B ‘When the gastric organoids were cultured with TGFalpha for 6 days, many organoids became dark in color, suggesting that morphological changes were induced’: This is not really clearly visible in the provided pictures. Darkening of organoids often is due to apoptosis and cells being shed into the lumen. Did the author’s check apoptotic markers to exclude a morphological change due to emerging apoptosis? And what is the effect of long term culturing with Tgalpha? Can they be maintained long term? Or does the differentiation towards pit cells stop their proliferation?

We thank the reviewer for these comments. We agree that darkening is also observed in apoptotic organoids. We have conducted cleaved-caspase-3 staining and found that apoptotic cells were not found

in most of TGF α -treated organoids. The apoptotic cells were only detected in the cells shed into the lumen of the organoids (Supplementary Fig. 17k). Because it is difficult to accurately distinguish between cell morphological change and cell shedding using phase contrast microscopy, the morphological changes were confirmed by H&E staining as shown in Fig. 4h. Therefore, we removed this sentence “When the gastric organoids were cultured with TGF α for 6 days, many organoids became dark in color” from the revised manuscript.

We attempted the long-term culture of TGF α -treated organoids. We found that they cease their proliferation and cannot be maintained after passage (Supplementary Fig. 17j), which is consistent with the result that the percentage of MKI67⁺ proliferating cells is less than 4% in TGF α -treated organoids (Fig. 4e).

- Figure 6B: What about the other neck cell makers (*Muc6*, *Tff2*) and chief cell markers (*Gif*, *Troy*, *Lgr5*)?

Following the reviewer’s suggestion, we have analyzed the expression of those markers. First of all, the expression levels of *Muc6*, *Gif*, *Troy*, and *Lgr5* were quite low in our gastric organoids because our organoids do not contain neck cells and chief cells (Supplementary Fig. 17b). In contrast, *Tff2* was highly expressed in our organoids because *Tff2* was ubiquitously expressed not only in neck cells but also in isthmus progenitor cells as shown in Fig. 1b. We found that TGF α decreased *Tff2* expression but did not affect other neck and chief cell markers (Supplementary Fig. 17e).

- Figure 6C: Please add stainings for *Muc6*/GSII and *Gif*/Pgc on the organoids to confirm finding of mainly pit cells, and not chief / neck cells in their organoid cultures.

We thank the reviewer for this comment. We have added the images of the gastric organoids stained for GSII and PGC (Supplementary Fig. 17b), showing that our gastric organoids did not contain GSII⁺ and PGC⁺ cells. We stained the gastric organoids and corpus tissues at the same time, confirming that the lack of staining of GSII and PGC in the gastric organoids was not due to a failure of the immunofluorescence staining.

- Figure 6F-G: The authors need to prove that the EGFR pathway is indeed inhibited upon erlotinib treatment (western blot / IHC) and show by IHC the validity of qPCR data.

We thank the reviewer for raising this critical point. First, we have confirmed that erlotinib treatment suppressed TGF α -mediated ERK phosphorylation using immunofluorescence staining

(Supplementary Fig. 18b, c). Second, we have analyzed the expression of GKN2, MKI67, and SOX9 in erlotinib-treated organoids using immunofluorescence staining (Supplementary Fig. 18d, e). We found that erlotinib treatment decreased GKN2 expression and increased the percentage of SOX9⁺ cells but did not affect the number of MKI67⁺ cells at protein level.

- Figure 6G: please add neck cell markers and more chief cell markers incl. Lgr5/Troy

We have included the gene expression data for neck and chief cell markers (Supplementary Fig. 18a). We found that erlotinib treatment significantly increased the expression of *Tff2*, *Troy* and *Gif*, supporting the notion that EGFR-signaling inhibits neck and chief cell markers.

- Figure 6K-L: The authors need to show that the NFκB signaling pathway is indeed upregulated after TNFα treatment (western blot / IHC) and by IHC that the pit cells are indeed gone in the organoids.

We thank the reviewer for raising this critical point. We confirmed the increase of NF-κB1 protein expression in TNFα-treated organoids using immunofluorescence staining (Supplementary Fig. 19e, f). We also added the immunostaining data showing that GKN2 expression significantly decreased in TNFα-treated organoids (Supplementary Fig. 19g, h).

- Figure 6L: add more chief and neck cell markers.

We have included the gene expression data for neck and chief cell markers (Supplementary Fig. 19d). As described above, the expression level of neck and chief cell markers except for *Tff2* were significantly low in our gastric organoids. We found that TNFα increased the expression of *Tff2* and did not affect the expressions of *Muc6*, *Troy*, *Gif*, and *Lgr5*.

Figure 7:

- Figure E-F: Again the authors should make sure that their treatment with the ERK inhibitor downregulated the ERK signaling cascade by protein data.

We thank the reviewer for raising this critical point. Because it is difficult to measure the activity of SCH772984, which inhibits the phosphorylation of ERK substrates but not the phosphorylation of ERK, we additionally examined the effects of the MEK inhibitor PD0325901. We showed that

PD0325901 suppresses pit cell differentiation similar to SCH772984 (Fig. 7g, h) and inhibits ERK phosphorylation in gastric organoids (Fig. 7i, j).

Figure S9G: add more chief and neck cell markers.

We have added the data for the expression of those markers (Supplementary Fig. 20a). We found that the NF- κ B inhibitor QNZ increased the expression of *Tff2* but did not affect the expression of *Muc6*, *Troy*, *Gif*, and *Lgr5*.

Minor:

Figure 1E: the expression of ACTA2 at the bottom of glands seems to be only true in the antrum, while in the corpus the expression seems also to be more prominent in the pit region, just as PDGFR α . Please comment and possibly specify/correct (also in the discussion p. 22).

We agree that the description about the ACTA2⁺ and PDGFR α ⁺ stromal cells was inaccurate. Therefore, we measured spatial distribution of the two stromal cells along the gastric glands (Supplementary Fig. 12e). In corpus, PDGFR α ⁺ cells were distributed with two peaks at the upper and lower sections of the glands, whereas ACTA2⁺ cells were mainly located at the middle section of the glands. We have removed the original sentence and replaced it with the following sentence:

“Notably, ACTA2⁺ stromal cells were primarily located in the middle section, whereas PDGFR α ⁺ stromal cells were primarily located in the upper and lower sections of the corpus gastric glands (Supplementary Fig. 12e).” (P10, L2)

To clarify that the aim of this study is the analysis of corpus unit, we removed the description about the antrum units. We have also removed the relevant part of the description in the Discussion section.

Figure S3: Please show other putative stem cell markers: CCK2R, Axin2, Lgr5, Aqp5, eR1+, Troy

We added the violin plots showing the expression of these genes (Supplementary Fig. 4b)

Figure 5B Please comment the high expression of Igf signaling ligands in the stroma of gastric glands

We have added the following sentence in Discussion part.

“CellChat analysis also suggested that IGF signaling is activated in the neck cells in an autocrine manner and a paracrine manner from stromal cells (Fig. 3a and Supplementary Fig. 13c, 15a).” (P19, L12)

Are the TFs presented in Figure 2, 3 and 4 the only ones or selected in one way or the other?

The full list of TFs in each group is shown in Supplementary Table 1-5. In Fig. 2 and a subsequent similar analysis, the top 10 TFs specifically expressed in each cell type were selectively shown in the list. For example, in the case of pit cell differentiation-related group A (Fig. 2d), we selected the top 10 TFs (ranked by LogFC) that were specifically expressed in pit cells. We have added text to Figure Legends to describe how we selected the TFs.

“Among the TFs involved in group A and B, the top 10 TFs specifically expressed in pit cells and isthmus progenitor cells (LogFC > 0.25) are shown here, respectively.” (P31, L8)

The authors identified for various cell lineages potentially novel cell type specific markers. They might want to discuss these markers and potentially published literature in the discussion part.

For the TFs involved in pit cell differentiation and isthmus progenitor cell maintenance, we discussed that as follows:

“Pit cell-related group A contained several novel Transcription factors (TFs) such as *Foxa3*, *Pparg*, and *Tcf23*, in addition to the previously reported pit cell differentiation-related TF *Klf4*²⁷. In contrast, the isthmus progenitor cell-related group B contained *Dnmt1*, *Cebpb*, *Foxm1* (Fig. 2d), and *Hmga1* (Supplementary Table 1), which are associated with proliferative gastric cancer²⁸⁻³¹.” (P7, L10)

We also discussed the expression of isthmus progenitor cell markers identified in the previous literatures and newly identified in our analysis as follows (P6, L8):

“Although some of the previously identified isthmus progenitor cell-specific markers (*Stmn1* and *Iqgap3*)^{12,24} and the novel markers (*Tcf19* and *Hmgb2*) were identified, pan-stem cell markers, such as *Sox2*, *Prom1*, and *Bmi1*, did not exhibit isthmus cell-specific expression (Fig. 1b and Supplementary Fig. 4b).”

Citation. “TGF α -mediated gene regulation was repeatedly observed with passaged corpus organoids (Figures S9A and S9B).” -> not sure what the authors want to say with this sentence and the referred figures (maybe mistake, should be S9B and S9C?). What does “passaged”

mean?

We appreciate you pointing out the mistake and we have corrected it. In the revised manuscript, these figures are Supplementary Figs. 17f and g. The aim of this experiment was to confirm whether the isthmus progenitor cells enriched in passaged organoids could respond to TGF α and differentiate into pit cells. We explained “passaged corpus organoids” in the figure legend as follows:

“The organoids were grown without TGF α for 6 days and then dissociated with a recombinant enzyme, TrypLE Express, to generate passaged organoids. The passaged organoids were further grown with or without TGF α (25 ng/mL) for 6 days.” (P44, L32)

Citation “Similarly, the TGF α -dependent induction of pit cell markers and the decrease in isthmus progenitor cell markers were also observed in the antrum organoids (Figures S9C and S9D).” -> Fig. S9D and S9E?

We thank the reviewer for pointing out the Figure numbering error. To clarify that the aim of this paper is the analysis of corpus epithelial cells, we have removed the description about the antrum organoids.

S10: Please specify how genes for the clustering (“EGFR downstream effectors”) were selected.

We have removed this data in the revised paper. The enrichment analysis using dataset 2 showed that Mapk signaling is enriched in pit cell-related gene group. This data suggest that Mapk signaling is the downstream effector of EGFR signaling more clearly than the previous expression analysis of EGFR downstream effectors.

Typo on page 16: (Figures s 6K and 6L)

We have corrected the typo.

Citation Discussion “Although the previous report identified Tgfa expression in parietal and chief cells...” Which previous report?

Thank you for pointing out the missing citation. We have added the citation as follows:

“An early study reported the expressions of TGF α and its receptor transcripts in parietal cell-enriched fractions by northern blotting⁴⁷.” (P18, L5)

Reference 47: Beauchamp, R. D., Barnard, J. A., McCutchen, C. M., Cherner, J. A. & Coffey, R. J. Localization of transforming growth factor alpha and its receptor in gastric mucosal cells. Implications for a regulatory role in acid secretion and mucosal renewal. *J Clin Invest* **84**, 1017–1023 (1989).

Discussion: “Currently there are no efficient protocols for the induction of fully differentiated chief cells...” Chief cells have been cultured successfully in organoids derived from Troy cells (Stange et al, Cell, 2013). Only parietal cells have not been successfully cultured in organoids.

Following the reviewer’s suggestion, we have removed the phrase “chief cell” in this sentence (P20, L9).

REVIEWER COMMENTS

Reviewer #1 (Remarks to the Author):

In this revised manuscript by Takada et al., the authors use a detailed single-cell transcriptomic analysis of the gastric corpus and antrum at homeostasis through Quartz-Seq2 to determine preferentially expressed pathways during gastric cell lineage specification. In detail, they showed that the TGF α -EGFR-ERK pathway is important in driving pit cell differentiation, and the TNFSF12-NF-kappaB pathway preferentially increased undifferentiated cells through pit cell de-differentiation. The novelty of this paper lies in the new single-cell RNA technique and determination of cell lineage transcriptional changes with validation in an in vitro organoid system. Improved in vivo data that support their in vitro findings has now greatly improved the impact of this study.

The authors have thoroughly addressed our previous reviewer comments in a detailed fashion and all previous points have now been sufficiently addressed in a thoughtful manner. We provide a few minor comments and philosophical points of interest below.

Minor comments:

1. Given the multiple datasets incorporating corpus/antrum in dataset #1 and corpus only in dataset #2. The authors should consider moving Supp Fig 1A/B into the main manuscript to guide the reader.
2. Page 6 line 3, the authors are referring to progenitor cell "clusters".
3. For Figure 2C, the hierarchical clustering of the nodes has included "neck cells" and "pre-pit cells" in similar colors. It is difficult to differentiate them. Also, if the authors are focused on the isthmal->pit differentiation process, why were neck cells included?
4. On Page 9, the discussion of dataset 1 seems a bit extraneous as it supports dataset 2. Perhaps the authors can shorten this and integrate it into the discussion of dataset 2 results.
5. Figure 3 A is very complex. Do the authors need to show all of these pathways? Are these all of the "significant" pathways. If not, the authors should consider shortening the list as they only discuss a few of the pathways in the text.

6. "leucocytes" is misspelled on page 11.

7. It is interesting that the gastric organoids "de-differentiate" through passaging as the authors point out on page 12. Do the authors believe that this is due to signaling pathways changing with passaging? Could the actual stress/injury of passaging be causing this as well?

8. Another philosophical point that the authors point out is the differences between rWnt3A and Wnt3a-CM. The authors should be credited for undergoing this detailed experiment. They seem to indicate that CM induces the neck/chief cell lineage compared to rWNT3a. Do they believe this to do due to the quantitate levels of Wnt3a? Or there another component in the conditioned media that is contributing? Have the authors tried 2D vs 3D culturing systems? It appears that a 2D system may more replicate the "pit region" and a 3D organoid system may more resemble the deeper "gland region" (PMID: 36198679). Results may be similar to your rWnt3A vs Wnt3a-CM findings.

9. For the IF pictures of treated/untreated organoids (Fig 4C vs Fig 5C and Fig 6E), the authors should consider being consistent with the axis labeling (ie keeping the treated or untreated left vs right or up vs down).

10. Page 15, "Mapk" nomenclature should be MAPK.

11. In Supp Fig 1B the labeling "Operator for 1-cell isolation is different for each biological replicate" is a bit unclear. The authors should consider rephrasing this.

12. On page 26, "captizol" should be Captisol.

Reviewer #2 (Remarks to the Author):

11_29_2022

In this manuscript titled "EGFR signaling is a potent inducer of differentiation but not proliferation of gastric progenitor cells in stomach epithelial homeostasis", the authors used the single-cell RNA sequencing and a gastric organoid assay revealed that EGFR-ERK signaling promotes pit cells differentiation and suppresses neck and chief cells fate decisions, while NF-KB signaling maintains gastric

progenitor cells in an undifferentiated state. Overall, authors have addressed most of the comments raised by reviewer. However, there is still some issues to remain:

1. In Figure 1b, by scRNAseq of violin plot, authors demonstrated SOX9 is equally expressed across pit cells, isthmus progenitor cells, neck cells, Chief cell and even parietal cells. However, authors used SOX9 as isthmus progenitor marker in Figure 4b&4g, Figure 5b-5f, Figure 6b&6d as well as Figure 7d,7f,7h. Authors need to explain why used SOX9 as marker of isthmus progenitor cells although equally expressed in all different gastric epithelial cells.

2. The evidence of NF- κ B signaling maintains gastric progenitor cells in undifferentiated state was not so sound and strong. Authors just used TNFSF12 and TNF α treated gastric organoids and detected the markers of pit cells (Muc5ac, Gkn2 and Aqp3) and progenitor cells (Mk67, Sox9 and Pgc) by q-PCR. The changes caused by TNFSF12 and TNF α may or may not be through NF- κ B. Genetic approaches may need to be applied. In addition, the data of Sox9 and Pgc were not consistent between Figure 5b and Figure 5f. Authors may need to explain.

3. In page 17 the discussion section, authors claimed that “NF- κ B mediated DUSP1 induction might inhibit EGFR-MAPK signaling”, There is less evidence to state this conclusion.

4. The big concern for this manuscript is the clinical relevance. In Figure 7i, authors treated mice using EGFR inhibitor Erlotinib and found that Pit cells significantly reduced but progenitor zone were expanded. This is hard to explain in the oncology space, EGFR inhibitors or neutralized antibody are commonly be used since EGFR is highly unregulated or amplified in many tumor types. As everyone knows, stem cells or progenitors are the root for tumorigenesis and therapy resistance and even metastases, it will be problematic if cancer patients used EGFR inhibitors in the clinics.

Reviewer #3 (Remarks to the Author):

This revised manuscript addresses the differentiation of gastric progenitor cells in the corpus region of the stomach during healthy homeostasis. This study relies heavily on results from single-cell RNA sequencing method, Quartz-Seq2, to obtain gene expression profiles from different cell types in healthy mouse stomachs. Cells are clustered to identify different subsets and pseudotemporal analyses are done to predict development of different subsets, and differential gene expression and 3-D organoid studies are used to predict important pathways for gastric epithelial cell differentiation. The data are interpreted as evidence that EGFR signaling is involved in promoting the development from progenitor cells to mucous producing pit cells.

The authors have done a lot of work and a nice job addressing many issues raised in the previous reviews by several reviewers. A major concern listed by several reviewers was the limited amount of transcriptional data from a single experiment of a mixture of antrum and corpus. In the revised version an additional experiment that includes sequencing more cells from the corpus provides more data and evidence of reproducibility. There are still concerns about how much of an impact this study will have on the field, as it is not clear how these data, which focus primarily on pit cell differentiation in healthy mouse stomachs, will help understand disease phenotypes in humans with gastric disorders (atrophic gastritis, intestinal metaplasia, and gastric cancer).

Reviewer #4 (Remarks to the Author):

In the revised manuscript by Takeda et al. with the title `EGFR signaling is a potent inducer of differentiation but not proliferation of gastric progenitor cells in stomach epithelial homeostasis` all of my mentioned major and minor points were largely addressed.

However, I still have some minor comments/questions:

1. Introduction P3, L17-19, Citation: However, a recent study suggested the self-renewal ability of chief cells as stem cells to maintain themselves. – Here, the authors missed the publications by Stange et al., Cell, 2013 and Leushacke et al., Nature 2017.
2. Suppl. Figure 3 and P5, L15: Why does the authors only show the validation of the pit cell marker Baps1 and no data for Sctr?
3. P6, L9: The authors should comment on the new identified isthmus marker Tcf19 and Hmgb2.
4. P10, L9 and Figure 3a: The authors only mention the high input of EGF signaling in pit cells but what is with the high output of EGF signaling in these cells?
5. Figure 2D: Authors should add to the overview the logFC to the corresponding TFs to make it clear that the TFs mentioned are the top10.
6. Suppl. Figure 17: The authors should rename the phrase “passaged organoids” it in the manuscript (P12), figure legends and corresponding Suppl. Figure 17 as it is not clear what is meant by it.
7. Figure 4h: The authors should higher the magnification of the HE staining to clearly show the cuboidal cells.
8. P20, L9: Removal of the phrase “chief cell” in the sentence. As noted previously, chief cells could be successfully cultured in organoids. The paper (Stange et al., Cell, 2013, PMID: 24120136) should be cited here and not the phrase “chief cell” deleted.

9. Discussion: The author show in Figure 2 and Suppl. Figures 7,8,9,10 a lot of TFs. Some were also mentioned in the manuscript part. Comments to this should be included in the discussion.

Dear Reviewers,

We extend our gratitude to the reviewers for reviewing our manuscript. Their constructive comments were invaluable for improving the quality of our work. We have addressed the comments raised by the reviewers and have detailed our changes below. In the manuscript, the revised sections are in a red font.

REVIEWER COMMENTS

Reviewer #1 (Remarks to the Author):

In this revised manuscript by Takada et al., the authors use a detailed single-cell transcriptomic analysis of the gastric corpus and antrum at homeostasis through Quartz-Seq2 to determine preferentially expressed pathways during gastric cell lineage specification. In detail, they showed that the TGF α -EGFR-ERK pathway is important in driving pit cell differentiation, and the TNFSF12-NF-kappaB pathway preferentially increased undifferentiated cells through pit cell de-differentiation. The novelty of this paper lies in the new single-cell RNA technique and determination of cell lineage transcriptional changes with validation in an in vitro organoid system. Improved in vivo data that support their in vitro findings has now greatly improved the impact of this study.

The authors have thoroughly addressed our previous reviewer comments in a detailed fashion and all previous points have now been sufficiently addressed in a thoughtful manner. We provide a few minor comments and philosophical points of interest below.

Minor comments:

1. Given the multiple datasets incorporating corpus/antrum in dataset #1 and corpus only in dataset #2. The authors should consider moving Supp Fig 1A/B into the main manuscript to guide the reader.

Thank you for your comments. According to the reviewer's suggestion, we have moved Supp Fig. 1a, b to the main Fig. 1a, b.

2. Page 6 line 3, the authors are referring to progenitor cell "clusters".

We have revised this to read "progenitor cell clusters" on P6 L3.

3. For Figure 2C, the hierarchical clustering of the nodes has included "neck cells" and "pre-pit cells" in similar colors. It is difficult to differentiate them. Also, if the authors are focused on the isthmal->pit differentiation process, why were neck cells included?

Following the reviewer's comment, we have changed the colors for each cell type, as shown in Fig. 2c. FateID is a software that uses all cell clusters as progenitor cells, except for manually defined terminally differentiated cell clusters. FateID computes the fate bias of these cells within each lineage. In our analysis, we defined the terminally differentiated cell clusters as follows: cluster 13 for the pit cell lineage, cluster 1 for the parietal cell lineage, and cluster 12 for the neck cell lineage (P27, L12-14). As a result, FateID automatically assigned one of the neck cell clusters (cluster 2) to the pit cell lineage, presumably because cluster 2 was a border cluster that was very close to those of isthmus progenitor cells and immature pre-pit cells. We believe that this is an inherent limitation of the FateID algorithm.

4. On Page 9, the discussion of dataset 1 seems a bit extraneous as it supports dataset 2. Perhaps the authors can shorten this and integrate it into the discussion of dataset 2 results.

We have followed the reviewer's suggestion and integrated the discussion of dataset 1 into the dataset 2 results. (Pit cell lineage: P8, L1, Neck cell lineage: P8, L16, Parietal cell lineage: P9, L9)

5. Figure 3 A is very complex. Do the authors need to show all of these pathways? Are these all of the "significant" pathways. If not, the authors should consider shortening the list as they only discuss a few of the pathways in the text.

Following the reviewer's suggestion, we minimized the list of outgoing and incoming pathways in the new Figure 3a, highlighting BMP, IGF, NOTCH, EGF, and TNF signaling as discussed in the manuscript. All pathways have been moved to Supplementary Fig. 15.

6. "leucocytes" is misspelled on page 11.

We have corrected this typo.

7. It is interesting that the gastric organoids "de-differentiate" through passaging as the authors point out on page 12. Do the authors believe that this is due to signaling pathways changing with passaging? Could the actual stress/injury of passaging be causing this as well?

Thank you for your comment. We predicted that serial passaging of the organoids resulted in the enrichment of isthmus progenitor cells and the decline of differentiated pit cells, leading to increased expression of isthmus progenitor cell markers and decreased expression of pit cell markers. This prediction was supported by the data in Supplementary Fig. 20j, which demonstrated that the differentiated pit cells did not regenerate

the organoids after passage. Therefore, one factor contributing to the “de-differentiation” of gastric organoids through serial passaging is the enrichment of isthmus progenitor cells. However, we cannot exclude the possibility that stress induced by serial passaging could alter the characteristics of organoids, as mentioned by the reviewer. The dramatic decrease in the expression of pit cell markers could not be fully accounted for by progenitor cell enrichment. Thus, to avoid the effects of serial passaging, our analysis was conducted with passage 0 (P0) organoids.

8. Another philosophical point that the authors point out is the differences between rWnt3A and Wnt3a-CM. The authors should be credited for undergoing this detailed experiment. They seem to indicate that CM induces the neck/chief cell lineage compared to rWNT3a. Do they believe this to do due to the quantitate levels of Wnt3a? Or there another component in the conditioned media that is contributing? Have the authors tried 2D vs 3D culturing systems? It appears that a 2D system may more replicate the "pit region" and a 3D organoid system may more resemble the deeper "gland region" (PMID: 36198679). Results may be similar to your rWnt3A vs Wnt3a-CM findings.

Thank you for your feedback. In our study, we used rWnt3a to eliminate the potential impacts of the components present in the conditioned medium and the inconsistency arising from lot-to-lot variation in FBS on the gene expression of the gastric organoids. We reasoned that a chemically defined medium would enable us to more accurately analyze the direct effects of growth factors and small compounds on gastric marker gene expression. We have not yet started 2D culture system. However, we would like to explore the differentiation tropism between 2D and 3D culture systems using our chemically defined medium in future.

9. For the IF pictures of treated/untreated organoids (Fig 4C vs Fig 5C and Fig 6E), the authors should consider being consistent with the axis labeling (ie keeping the treated or untreated left vs right or up vs down).

Following this suggestion, we have revised Fig. 5c so that the axis labeling is consistent with Figures 4c and 6e.

10. Page 15, "Mapk" nomenclature should be MAPK.

We have revised this on P16 L3 and L4.

11. In Supp Fig 1B the labeling "Operator for 1-cell isolation is different for each biological replicate" is a bit unclear. The authors should consider rephrasing this.

We have revised this sentence to “Each biological replicate was prepared by a different operator.”

12. On page 26, "captizol' should be Captisol.

We have revised this (P28 L7).

Reviewer #2 (Remarks to the Author):

11_29_2022

In this manuscript titled "EGFR signaling is a potent inducer of differentiation but not proliferation of gastric progenitor cells in stomach epithelial homeostasis", the authors used the single-cell RNA sequencing and a gastric organoid assay revealed that EGFR-ERK signaling promotes pit cells differentiation and suppresses neck and chief cells fate decisions, while NF-KB signaling maintains gastric progenitor cells in an undifferentiated state. Overall, authors have addressed most of the comments raised by reviewer. However, there is still some issues to remain:

1. In Figure 1b, by scRNAseq of violin plot, authors demonstrated SOX9 is equally expressed across pit cells, isthmus progenitor cells, neck cells, Chief cell and even parietal cells. However, authors used SOX9 as isthmus progenitor marker in Figure 4b&4g, Figure 5b-5f, Figure 6b&6d as well as Figure 7d,7f,7h. Authors need to explain why used SOX9 as marker of isthmus progenitor cells although equally expressed in all different gastric epithelial cells.

Thank you for your comments. In this study, we used *Sox9* as a gene whose expression levels increased from pit cells toward neck cells. To clearly show the differences in *Sox9* expression among cell types, multiple clusters of the same cell types in Fig. 1d were combined into one cluster, as shown in Supplementary Fig. 6a, b. The violin plot revealed the highest expression of *Sox9* in neck cells and decreased expression in isthmus progenitor and pit cells. *Sox9* was detected as a marker gene for neck cells (p-value = 1.46E-247). Feature plots also revealed that *Sox9* is weakly expressed within a subset of *Mki67*-expressing isthmus cells but highly expressed in neck cells (Supplementary Fig. 7a), which is complementary to the expression of the pit cell lineage marker *Muc5ac* (Supplementary Fig. 7d). As the expression of *Sox9* is inversely correlated with pit cell differentiation, *Sox9* could serve as a useful marker for monitoring the effects of TGF α on pit cell differentiation. Similarly, we analyzed *Pgc* as its expression increased from pit cells towards neck cells (Supplementary Fig. 6b and Supplementary Fig.7b, 7e). To clarify the intent of analyzing the expression of *Sox9* and *Pgc*, we have added the following text to the Results section (P11, L24).

"Therefore, subsequent analyses primarily focused on the expression of the pit cell markers *Muc5ac*, *Gkn2*, and *Aqp3*, as well as the isthmus progenitor cell marker *Mki67*. Additionally, we analyzed the expression of *Sox9* and *Pgc*, whose expression levels gradually increased from pit to neck cells (Supplementary Fig. 6, 7)."

2. The evidence of NF- κ B signaling maintains gastric progenitor cells in undifferentiated state was not so sound and strong. Authors just used TNFSF12 and TNF α treated gastric organoids and detected the markers of pit cells (*Muc5ac*, *Gkn2* and *Aqp3*) and progenitor cells (*Mki67*, *Sox9* and *Pgc*) by q-PCR. The changes caused by TNFSF12 and TNF α may or may not be through NF- κ B. Genetic approaches may need to be applied. In addition, the data of *Sox9* and *Pgc* were not consistent between Figure 5b and Figure 5f. Authors may need to explain.

As mentioned in comment 1, we realized that the description of the marker genes used in the qPCR analysis was ambiguous, leading to confusion.

To examine whether TNFSF12 and TNF α suppressed pit cell differentiation through NF- κ B, gastric organoids were treated with the NF- κ B inhibitor QNZ in the presence of TNFSF12 or TNF α . As shown in Supplementary Fig. 23f-i, inhibition of NF- κ B abrogated TNFSF12- or TNF α -mediated suppression of pit cell markers (*Muc5ac*, *Gkn2*, and *Aqp3*) and increased the expression of the isthmus progenitor cell marker *Mki67*, indicating that NF- κ B is one of the downstream effectors of TNFSF12 and TNF α -dependent maintenance of isthmus progenitor cells.

Although the above results suggest that NF- κ B serves as a downstream mediator of TNFSF12 and TNF α , it is possible that additional downstream factors play a role in the regulation of gastric epithelial cell differentiation. In particular, TNFSF12 and TNF α differentially regulate the expression of the neck cell lineage markers *Sox9* and *Pgc*, suggesting the involvement of different downstream factors. Therefore, we have added the following text to the Results section (P15, L4).

“To further examine whether TNFSF12- and TNF α -mediated maintenance of gastric progenitor cells is mediated by NF- κ B, gastric organoids were treated with QNZ in the presence of TNFSF12 or TNF α . QNZ treatment abrogated TNFSF12- and TNF α -mediated decreases in pit cell markers and increases in *Mki67* expression (Supplementary Fig. 23 f-i), suggesting that NF- κ B is one of the downstream effectors of TNFSF12 and TNF α -dependent maintenance of isthmus progenitor cells. Despite the consistent effects of TNFSF12, TNF α , and QNZ on the expression of *Mki67* and pit cell markers, their actions on *Sox9* and *Pgc* were inconsistent, suggesting that additional downstream factors may be involved in the regulation of these genes.”

3. In page 17 the discussion section, authors claimed that “NF- κ B mediated DUSP1 induction might inhibit EGFR-MAPK signaling”, There is less evidence to state this conclusion.

According to the reviewer’s suggestion, we have revised the sentence as follows.

“one possible explanation is that NF- κ B signaling induces the expression of the MAP kinase phosphatase DUSP1.” (P18, L11)

“It would be interesting to examine whether NF- κ B-mediated DUSP1 induction inhibits EGFR-MAPK signaling in gastric epithelial cells.” (P18, L17)

4. The big concern for this manuscript is the clinical relevance. In Figure 7I, authors treated mice using EGFR inhibitor Erlotinib and found that Pit cells significantly reduced but progenitor zone were expanded. This is hard to explain in the oncology space, EGFR inhibitors or neutralized antibody are commonly be used since EGFR is highly unregulated or amplified in many tumor types. As everyone knows, stem cells or progenitors are the root for tumorigenesis and therapy resistance and even metastases, it will be problematic if cancer patients used EGFR inhibitors in the clinics.

Thank you for your comments. Chronic infection with *H. pylori* induces genome instability (PMID:33989515), which increases the mutation rate in gastric epithelial cells. According to Correa's gastric cancer model, multiple steps are required to transform healthy gastric epithelial cells into gastric cancer cells (PMID:22188910). Additionally, sequencing studies of tumors from patients with gastric cancer have revealed various mutated genes, including *TP53*, *PTEN*, and *CDH1*, as well as copy number alterations in *EGFR*, *KRAS*, and *EGFR2/HER2* (PMID:31164161). The progressive accumulation of these mutations leads to the development of gastric cancer. In light of these differences in cellular conditions between gastric cancer cells and healthy gastric epithelial cells, the outcome of erlotinib treatment could be different between healthy and pathological conditions. In this study, we investigated the regulatory mechanisms of healthy gastric progenitor cells and found that EGFR signaling promotes the differentiation of progenitor cells toward pit cells, and inhibition of EGF signaling with erlotinib inhibits the differentiation. Our data also revealed that erlotinib increased the number of progenitor cells but did not generate cancer cells in the healthy gastric epithelium.

Reviewer #3 (Remarks to the Author):

This revised manuscript addresses the differentiation of gastric progenitor cells in the corpus region of the stomach during healthy homeostasis. This study relies heavily on results from single-cell RNA sequencing method, Quartz-Seq2, to obtain gene expression profiles from different cell types in healthy mouse stomachs. Cells are clustered to identify different subsets and pseudotemporal analyses are done to predict development of different subsets, and differential gene expression and 3-D organoid studies are used to predict important pathways for gastric epithelial cell differentiation. The data are interpreted as evidence that EGFR signaling is involved in promoting the development from progenitor cells to mucous producing pit cells.

The authors have done a lot of work and a nice job addressing many issues raised in the previous reviews by several reviewers. A major concern listed by several reviewers was the limited amount of transcriptional data from a single experiment of a mixture of antrum and corpus. In the revised version an additional experiment that includes sequencing more cells from the corpus provides more data and evidence of reproducibility. There are still concerns about how much of an impact this study will have on the field, as it is not clear how these data, which focus primarily on pit cell differentiation in healthy mouse stomachs, will help understand disease phenotypes in humans with gastric disorders (atrophic gastritis, intestinal metaplasia, and gastric cancer).

We appreciate your acknowledgement of the quality and reproducibility of our data. We believe that understanding the molecular mechanisms involved in gastric epithelial cell differentiation during normal homeostasis is a crucial prerequisite for the identification of the disease-specific signaling network. We hope that our findings will inspire further research into the pathogenesis of gastric disorders.

Reviewer #4 (Remarks to the Author):

In the revised manuscript by Takeda et al. with the title `EGFR signaling is a potent inducer of differentiation but not proliferation of gastric progenitor cells in stomach epithelial homeostasis` all of my mentioned major and minor points were largely addressed.

However, I still have some minor comments/questions:

1. Introduction P3, L17-19, Citation: However, a recent study suggested the self-renewal ability of chief cells as stem cells to maintain themselves. – Here, the authors missed the publications by Stange et al., Cell, 2013 and Leushacke et al., Nature 2017.

We thank the reviewer for pointing out these missing citations. We have cited these papers in the revised manuscript (P3, L20; References 13, 14).

2. Suppl. Figure 3 and P5, L15: Why do the authors only show the validation of the pit cell marker Baps1 and no data for Sctr?

We tested the Sctr antibody (14172-1-AP). However, we observed non-specific antibody binding in both immunofluorescence staining and western blotting. Therefore, we did not use this data in our manuscript.

3. P6, L9: The authors should comment on the new identified isthmus marker Tcf19 and Hmgb2.

According to the reviewer's suggestion, we added comments on Tcf19 and Hmgb2 in the Results section as follows (P6, L11):

“Previous studies have shown that TCF19 increases cell cycle progression in the gastric cancer cell line MKN-45²⁷ and that HMGB2 is a downstream effector of CENPU-mediated cell proliferation in AGS cells²⁸, which is consistent with the highly proliferative state of isthmus progenitor cells.”

4. P10, L9 and Figure 3a: The authors only mention the high input of EGF signaling in pit cells but what is with the high output of EGF signaling in these cells?

Thank you for your comments. The detailed expression pattern of EGF ligands is shown in Figure 3d, which shows that *Tgfa* and *Btc* were preferentially expressed in the pit cell lineage (P10, L23). To state the consistency between CellChat prediction and the expression pattern of EGF ligands more clearly, we inserted the red-colored sentence as follows (P10, L22).

“Single-cell RNA-seq data showed that EGF ligands are expressed in several epithelial cell types but not in stromal cells, which is consistent with a high EGF signaling output in pit cells as predicted by CellChat (Fig. 3a), and that the predominant EGF ligands expressed in pit cells were *Tgfa* and *Btc* (Fig. 3d).”

5. Figure 2D: Authors should add to the overview the logFC to the corresponding TFs to make it clear that the TFs mentioned are the top10.

Following the reviewer’s suggestion, we have added Supplementary Tables 6 and 7, summarizing the Log2FC for the pseudotime-dependent TFs identified in each lineage.

6. Suppl. Figure 17: The authors should rename the phrase “passaged organoids” in the manuscript (P12), figure legends and corresponding Suppl. Figure 17 as it is not clear what is meant by it.

Thank you for your comments. We renamed the phrase “passaged organoids” with “passage 1 organoids”. (P12, L16)

7. Figure 4h: The authors should higher the magnification of the HE staining to clearly show the cuboidal cells.

Following the reviewer’s suggestion, we have replaced the pictures in Figure 4h with new pictures taken at 100x magnification. These images clearly show the morphological differences in cell height between gastric organoids treated with or without TGF α .

8. P20, L9: Removal of the phrase “chief cell” in the sentence. As noted previously, chief cells could be successfully cultured in organoids. The paper (Stange et al., Cell, 2013, PMID: 24120136) should be cited here and not the phrase “chief cell” deleted.

Following the reviewer’s suggestion, we have added the following sentence to the Discussion section (P21, L21).

“Previous studies has shown that chief cells can be cultured in gastric organoids^{14,66},”

9. Discussion: The author show in Figure 2 and Suppl. Figures 7,8,9,10 a lot of TFs. Some were also mentioned in the manuscript part. Comments to this should be included in the discussion.

We thank the reviewer for encouraging us to discuss the lineage-enriched transcription factors. We have added the following paragraph to the Discussion part (P20, L16).

“We identified several transcription factors that were enriched in specific lineages of gastric epithelial cells. For example, *Pparg*, *Pbx1*, and *Ybx2* were enriched in the pit, neck, and parietal cell lineages, respectively (Fig. 2d and Supplementary Fig. 10c, 12c). However, the precise functions of these factors in gastric epithelial cell differentiation have not yet been elucidated. Previous research demonstrated that *PPAR γ* plays a role in anti-inflammatory responses by inhibiting inflammatory genes, including NF- κ B⁵⁸, suggesting that *PPAR γ* may promote pit cell differentiation by inhibiting NF- κ B. Although *Pbx1* knockout mice show diminished proliferation and accelerated differentiation of chondrocytes, leading to defects in endochondral bone development⁵⁹, the role of *PBX1* in the stomach remains unclear. *Ybx2* is primarily expressed in germ cells⁶⁰ and inactivation of *YBX2* leads to infertility^{61,62}. In addition, *YBX2* has been shown to control the activation of brown adipocyte tissue by stabilizing mRNAs encoding proteins involved in mitochondrial function⁶³, suggesting that *YBX2* may regulate mitochondrial biogenesis during parietal cell differentiation. Further investigation is required to confirm the specific roles of these lineage-enriched transcription factors in gastric epithelial cell differentiation.”

REVIEWERS' COMMENTS

Reviewer #1 (Remarks to the Author):

Takada et al. detail single-cell transcriptomic analyses of the gastric corpus (and antrum) at homeostasis through Quartz-Seq2 to determine preferentially expressed pathways during gastric cell lineage specification. Through a subsequent cell lineage fate trajectory analysis, the authors focused on isthmus to pit cell differentiation. They show signaling pathways important for this process using CellChat analysis. Specifically, they demonstrate that the TGF α -EGFR-ERK pathway is important in driving pit cell differentiation, and the TNFSF12/TNF α -NF- κ B pathway preferentially maintains isthmus progenitor cells. Mechanistic validation is provided with 3D gastric organoid data through manipulation of these pathways using ligand agonists and antagonists to demonstrate morphologic, transcriptomic, and protein changes in pit cell differentiation/isthmal progenitor cell maintenance. In vivo data using an EGFR inhibitor support their in vitro findings.

The novelty of this paper lies in the use of single-cell RNA analysis to uncover differentiation pathways in the gastric corpus. Strong mostly in vitro (and in vivo) data support the bioinformative findings. The authors have addressed previous reviewer comments in a thorough and thoughtful manner, and the revised manuscript has been significantly improved. While, there might be limited clinical applicability of the current study as raised by other reviewers, the novel data workflow proposed here can likely be applied to other gastric cell lineages (as briefly detailed in the paper) including chief and parietal cells. Further elucidation of the signaling pathways involved in these cell lineages will be applicable to atrophic gastritis and gastric carcinogenesis (Correa sequence). Application of this single cell workflow under gastric injury conditions will also be very fruitful as future studies.

The only minor point is the addition of Figure 3C data. These CellChat signaling networks are dense and a bit confusing. Perhaps the authors can better highlight the epi-epi auto/paracrine signaling for the EGF signaling graph (left) and the stromal-epi paracrine signaling for the TNF signaling graph (right).

Reviewer #2 (Remarks to the Author):

The authors have been added additional experiments and try to respond the reviewer's comments and the reviewer satisfied the most of the responses to the comments.

However, the review is still have some concern on the question No3 (3. In page 17 the discussion section, authors claimed that “NF-kB mediated DUSP1 induction might inhibit EGFR-MAPK signaling”, There is less evidence to state this conclusion). Authors did not provide more evidence and only revised the Text.

Reviewer #4 (Remarks to the Author):

I have no further concerns. The authors addressed all my comments.

We thank the reviewers for their careful reading of the manuscript and constructive suggestions.

We addressed the reviewers' concerns and also the editorial requests. Our response to reviewers' comments are as below.

Reviewer #1 (Remarks to the Author):

Takada et al. detail single-cell transcriptomic analyses of the gastric corpus (and antrum) at homeostasis through Quartz-Seq2 to determine preferentially expressed pathways during gastric cell lineage specification. Through a subsequent cell lineage fate trajectory analysis, the authors focused on isthmus to pit cell differentiation. They show signaling pathways important for this process using CellChat analysis. Specifically, they demonstrate that the TGF α -EGFR-ERK pathway is important in driving pit cell differentiation, and the TNFSF12/TNF α -NF-kappaB pathway preferentially maintains isthmus progenitor cells. Mechanistic validation is provided with 3D gastric organoid data through manipulation of these pathways using ligand agonists and antagonists to demonstrate morphologic, transcriptomic, and protein changes in pit cell differentiation/isthmal progenitor cell maintenance. In vivo data using an EGFR inhibitor support their in vitro findings.

The novelty of this paper lies in the use of single-cell RNA analysis to uncover differentiation pathways in the gastric corpus. Strong mostly in vitro (and in vivo) data support the bioinformative findings. The authors have addressed previous reviewer comments in a thorough and thoughtful manner, and the revised manuscript has been significantly improved. While, there might be limited clinical applicability of the current study as raised by other reviewers, the novel data workflow proposed here can likely be applied to other gastric cell lineages (as briefly detailed in the paper) including chief and parietal cells. Further elucidation of the signaling pathways involved in these cell lineages will be applicable to atrophic gastritis and gastric carcinogenesis (Correa sequence). Application of this single cell workflow under gastric injury conditions will also be very fruitful as future studies.

We express our sincere appreciation for the invaluable insights and suggestions that you have provide thus far, which guided us to improve our paper.

We thank the reviewer for recognizing the novelty of our study's use of single-cell RNA sequencing data to uncover gastric progenitor cell differentiation pathways. We are interested in exploring gastric injury conditions such as atrophic gastritis and gastric ulcer in our future studies.

The only minor point is the addition of Figure 3C data. These CellChat signaling networks are dense and a bit confusing. Perhaps the authors can better highlight the epi-epi auto/paracrine signaling for the EGF signaling graph (left) and the stromal-epi paracrine signaling for the TNF signaling graph (right).

We agree that the networks are too small and dense. However, we found it difficult to alter the density of the network. Therefore, we have decided to enlarge these figures to improve their readability and moved them to Supplementary Figure 16. The violin plot showing the expression of ligands and receptor across gastric cell types (new Figure 3d) is the main data to suggest the epithelial to epithelial interactions for EGF signaling and the stromal to epithelial interactions for TNF signaling.

Reviewer #2 (Remarks to the Author):

The authors have been added additional experiments and try to respond the reviewer's comments and the reviewer satisfied the most of the responses to the comments.

We thank the reviewer for taking time to review our manuscript and for providing constructive feedback.

However, the review is still have some concern on the question No3 (3. In page 17 the discussion section, authors claimed that "NF- κ B mediated DUSP1 induction might inhibit EGFR-MAPK signaling", There is less evidence to state this conclusion). Authors did not provide more evidence and only revised the Text.

We have carefully considered your comment and have decided to remove the sentences about the hypothesis how NF- κ B inhibits EGFR signaling from the Discussion section as follows. We acknowledge the lack of evidence to support this hypothesis other than the expression pattern of *Dusp1*, because here we have discussed the possible mechanism based on the findings of previous literatures. Nevertheless, we are confident that the removal of this part does not affect the overall interpretation and conclusion of our paper.

~~"Although the mechanisms of EGFR-ERK inhibition by NF- κ B signaling remain unknown, one possible explanation is that NF- κ B signaling induces the expression of the MAP kinase phosphatase DUSP1. In the intestinal epithelium, Toll-like receptor (TLR) induces p38-mediated inflammatory response to luminal bacteria. Concomitantly, TLR induces DUSP1 through NF- κ B signaling, which counteracts p38, leading to hyporesponsiveness to commensal microbiota⁴⁹. We observed that in the adult mouse stomach, *Dusp1* is expressed in the isthmus progenitor cells but not in the pit cell lineage (Supplementary Fig. 24f). It would be interesting to examine whether NF- κ B-mediated DUSP1 induction inhibits EGFR-MAPK signaling in gastric epithelial-~~

cells. It would be interesting to examine how NF- κ B signaling inhibits EGFR-MAPK signaling to maintain the balance between self-renewal and differentiation of gastric progenitor cells in future studies." (Page 18, Line 10)

Reviewer #4 (Remarks to the Author):

I have no further concerns. The authors addressed all my comments.

We thank the reviewer for your support and your thoughtful comments throughout the review process.